# Provable Sample-Efficient Transfer Learning Conditional Diffusion Models via Representation Learning

**Ziheng Cheng**[1], **Tianyu Xie**[2], **Shiyue Zhang**[2], **Cheng Zhang**[2,3,*]

[1] Department of Industrial Engineering and Operations Research, University of California, Berkeley
[2] School of Mathematical Sciences, Peking University
[3] Center for Statistical Science, Peking University
`ziheng_cheng@berkeley.edu, tianyuxie@pku.edu.cn,`
`zhangshiyue@stu.pku.edu.cn, chengzhang@math.pku.edu.cn`

## Abstract

While conditional diffusion models have achieved remarkable success in various applications, they require abundant data to train from scratch, which is often infeasible in practice. To address this issue, transfer learning has emerged as an essential paradigm in small data regimes. Despite its empirical success, the theoretical underpinnings of transfer learning conditional diffusion models remain unexplored. In this paper, we take the first step towards understanding the sample efficiency of transfer learning conditional diffusion models through the lens of representation learning. Inspired by practical training procedures, we assume that there exists a low-dimensional representation of conditions shared across all tasks. Our analysis shows that with a well-learned representation from source tasks, the sample complexity of target tasks can be reduced substantially. Numerical experiments are also conducted to verify our results.

## 1 Introduction

Conditional diffusion models (CDMs) utilize a user-defined condition to guide the generative process of diffusion models (DMs) to sample from the desired conditional distribution. In recent years, CDMs have achieved groundbreaking success in various generative tasks, including text-to-image generation [Ho et al., 2020, Song et al., 2020, Ho and Salimans, 2022, Rombach et al., 2022], reinforcement learning [Janner et al., 2022, Chi et al., 2023, Wang et al., 2022, Reuss et al., 2023], time series [Tashiro et al., 2021, Rasul et al., 2021], and life science [Song et al., 2021, Watson et al., 2022, Gruver et al., 2024, Guo et al., 2024].

Training a CDM from scratch requires a large amount of data to achieve good generalization. However, in practical scenarios, users often have access to only limited data for the target distribution due to cost or risk concerns, making the model prone to over-fitting. In such small data regime, transfer learning has emerged as a predominant paradigm [Moon et al., 2022, Ruiz et al., 2023, Xie et al., 2023, Han et al., 2023]. By leveraging knowledge acquired during pre-training on large source datasets, transfer learning enhances the performance of fine-tuning on target tasks, facilitating few-shot learning and significantly improving practicality.

Among the successful applications of transfer learning CDMs, the conditions are typically high-dimensional vectors with embedded low-dimensional representations (features) that encapsulate all the information required for inference. In addition, these representations are likely to be task-agnostic,

---

[*]Corresponding Author.

39th Conference on Neural Information Processing Systems (NeurIPS 2025).

| Tasks | Backbone Score Network | Condition Encoder |
|---|---|---|
| Text-to-Image [Esser et al., 2024] | 2-8B | 4.7B |
| Text-to-Audio [Liu et al., 2024] | 350-750M | 750M |
| Robotic Control [Chi et al., 2023] | 9M | 20-45M |

Table 1: Comparing the number of parameters of different parts in CDMs.

enabling effective knowledge transfer. For example, in text-to-image generation, the text input is inherently in high-dimensional space, but contains low-dimensional semantic information such as object attributes, spatial relationships, despite the differences of styles or contents in different image distributions. To take advantage of this structure, condition encoders are often frozen in the fine-tuning stage [Rombach et al., 2022, Esser et al., 2024], which typically constitutes a significant portion of the overall model (see Table 1).

While this paradigm has demonstrated remarkable empirical success, its theoretical underpinnings remain largely unexplored. The following fundamental question is still open:

*Can transfer learning CDMs improve the sample efficiency of target tasks by leveraging the representation of conditions learned from source tasks?*

There are some recent works attempting to study the theoretical underpinnings of CDMs [Fu et al., 2024, Jiao et al., 2024, Hu et al., 2024], but focus on single task training. Notably, Yang et al. [2024] investigates transfer learning DMs under the assumption that the data is a linear transformation of a low-dimensional latent variable following the same distribution across all tasks. However, fine-tuning merely the data encoder is not a widely adopted training approach in practice.

In this paper, we take the first step towards addressing the above question. Our key assumption is that there exists a generic low-dimensional representation of conditions shared across all distributions. Then we show that, with a well-learned representation from source tasks, the sample complexity of target tasks can be reduced substantially by training only the score network. The main contributions are summarized as follows:

- In Section 3, we establish the first generalization guarantee for transferring score matching error in CDMs, showing that transfer learning can reduce the sample complexity for learning condition encoder in the target task. This is aligned with existing transfer learning theory in supervised learning. Specifically, we present two results in Theorem 3.4 and Theorem 3.6, under the settings of task diversity assumption and meta-learning[2], respectively. On the technical side, we develop a novel approach to tackle Lipschitz continuity under weaker assumptions on data distribution in Lemma 3.1, which may be of independent interest for the analysis of even single-task diffusion models.

- In Section 4, we provide an end-to-end distribution estimation error bound in transfer learning CDMs. To obtain an $L^2$ accurate conditional score estimator, we construct a universal approximation theory using deep ReLU neural networks in Theorem 4.1. Then by combining both generalization error and approximation error, Theorem 4.2 and 4.3 provide sample complexity bounds for estimating conditional distribution. Notably, our results are *the state of the art* even when reduced to single-task learning setting.

In Section A, we further utilize our results to establish statistical guarantees in practical applications of CDMs. In particular, we investigate amortized variational inference (Theorem A.1) and behavior cloning (Theorem A.2), and present guarantees in terms of posterior estimation and optimality gap, laying the theoretical foundations of transfer learning CDMs in practice. We also conduct numerical experiments in Section 5 to verify our results.

---

[2]In practice, the terms such as transfer learning, meta-learning, learning-to-learn, *etc.*, often refer to the same training paradigm, *i.e.*, to fine-tune on target tasks with limited data using knowledge from source tasks. However, in the theoretical framework, we use meta-learning to emphasize that target tasks and source tasks are randomly sampled from a meta distribution [Baxter, 2000], whereas in transfer learning, the tasks are fixed.

## 1.1 Related Works

**Score Approximation and Distribution Estimation**   Recently, some works analyze the score approximation theory via deep neural networks and corresponding sample complexity bounds for diffusion models. Oko et al. [2023] considers distributions with density in Besov space and supported on bounded domain. Chen et al. [2023b] assumes the data distribution lies in a low-dimensional linear subspace and obtains improved rates only depending on intrinsic dimension. Fu et al. [2024] studies conditional diffusion models for Hölder densities and Hu et al. [2024] further extends the framework to more advanced neural network architectures, *e.g.*, diffusion transformers. Wibisono et al. [2024] establishes a minimax optimal rate to estimate Lipschitz score by kernel methods. With an $L^2$ accurate score estimator, several works provide the convergence rate of discrete samplers for diffusion models [Chen et al., 2022b, 2023a, Lee et al., 2023, Chen et al., 2024]. Combining score matching error and convergence of samplers, one can obtain an end-to-end distribution estimation error bound.

**Transfer Learning and Meta-learning Theory in Supervised Learning**   The remarkable empirical success of transfer learning, meta-learning, and multi-task learning across a wide range of machine learning applications has been accompanied by gradual progress in their theoretical foundations, especially from the perspective of representation learning. To the best of our knowledge, Baxter [2000] is the first theoretical work on meta-learning. It assumes a universal *environment* to generate tasks with some shared features. Following this setting, Maurer et al. [2016] provides sample complexity bound for general supervised learning problem and Aliakbarpour et al. [2024] studies very few samples per task regime. Another line of research replaces the *environment* assumption and instead establishes connections between source tasks and target tasks through various notions of task diversity [Tripuraneni et al., 2020, Du et al., 2020, Tripuraneni et al., 2021, Watkins et al., 2023, Chua et al., 2021]. However, theoretical understandings of transfer learning for unsupervised learning are much more limited.

**Few-shot Fine-Tuning of Diffusion Models**   Adapting pre-trained conditional diffusion models to specific tasks with limited data remains a challenge in varied application scenarios. Few-shot fine-tuning aims to bridge this gap by leveraging various techniques to adapt those models to a novel task with minimal data requirements [Ruiz et al., 2023, Giannone et al., 2022]. A promising paradigm is to use transfer (meta) learning by constructing a representation for conditions in all the tasks, which has been widely applied in image generation [Rombach et al., 2022, Ramesh et al., 2022, Sinha et al., 2021], reinforcement learning [He et al., 2023, Ni et al., 2023], inverse problem [Tewari et al., 2023, Chung et al., 2023], *etc*. Another work Yang et al. [2024] is closely related to this paper, proving that few-shot diffusion models can escape the curse of dimensionality by fine-tuning a linear encoder.

## 2   Preliminaries and Problem Setup

**Notations**   We use $x$ and $y$ to denote the data and conditions, respectively. The blackboard bold letter $\mathbb{P}$ represents the joint distribution of $(x, y)$, while the lowercase $p$ denotes its density function. The superscript $k$ indicates the task index, and the subscript $i$ means the sample index. The norm $\|\cdot\|$ refers to the $\ell_2$-norm for vectors and the spectral norm for matrices. For the hypothesis class $\mathcal{F}$, we use $\mathcal{F}^{\otimes K}$ to refer its $K$-fold Cartesian product. For any $a, b \in \mathbb{R}$, $a \wedge b = \min\{a, b\}$ and $a \vee b = \max\{a, b\}$. Finally, we use standard $\mathcal{O}(\cdot), \Omega(\cdot)$ to omit constant factors.

### 2.1   Conditional Diffusion Models

Let $\mathbb{R}^{d_x}$ denote the data space and $[0, 1]^{D_y}$ denote the condition space. Let $\mathbb{P}$ be any joint distribution over $\mathbb{R}^{d_x} \times [0, 1]^{D_y}$ with density $p$ and $\mathbb{P}(\cdot|y)$ be the conditional distribution with density $p(\cdot|y)$. As in diffusion models, the forward process is defined as an Ornstein–Uhlenbeck (OU) process,

$$\mathrm{d}X_t = -X_t \mathrm{d}t + \sqrt{2}\mathrm{d}W_t, X_0 \sim \mathbb{P}(\cdot|y). \tag{2.1}$$

where $\{W_t\}_{t \geq 0}$ is a standard Wiener process. We denote the distribution of $X_t$ as $\mathbb{P}_t(\cdot|y)$. Note that the limiting distribution $\mathbb{P}_\infty(\cdot|y)$ is a standard Gaussian $\mathcal{N}(0, I)$.

To generate new samples, we can reverse the forward process (2.1) from any $T > 0$,

$$\mathrm{d}X_t^\leftarrow = (X_t^\leftarrow + 2\nabla \log p_{T-t}(X_t^\leftarrow|y))\mathrm{d}t + \sqrt{2}\mathrm{d}\overline{W}_t, X_0^\leftarrow \sim \mathbb{P}_T(\cdot|y), 0 \leq t \leq T. \tag{2.2}$$

where $\{\overline{W}_t\}_{0 \le t \le T}$ is a time-reversed Wiener process. Unfortunately, we don't have access to the exact conditional score function $\nabla \log p_{T-t}$ and need to estimate it through neural networks. For any $(x, y) \sim \mathbb{P}$ and score estimator $s$, define the individual denoising score matching objective [Vincent, 2011] as

$$\ell(x, y, s) := \frac{1}{T - T_0} \int_{T_0}^{T} \mathbb{E}_{x_t \sim \phi_t(\cdot|x)} \left[ \|s(x_t, y, t) - \nabla \log \phi_t(x_t|x)\|^2 \right] \mathrm{d}t, \tag{2.3}$$

where $\phi_t(x_t|x) = \mathcal{N}(x_t|\alpha_t x, \sigma_t^2 I), \alpha_t = e^{-t}, \sigma_t^2 = 1 - e^{-2t}$, is the transition kernel of $x_t|x_0 = x$. And the population error of score matching is

$$L^{\mathbb{P}}(s) := \mathbb{E}_{(x,y)\sim\mathbb{P}} \mathbb{E}_{t,x_t}[\|s(x_t, y, t) - \nabla \log p_t(x_t|y)\|^2] = \mathbb{E}_{(x,y)\sim\mathbb{P}}[\ell(x, y, s) - \ell(x, y, s_*^{\mathbb{P}})]. \tag{2.4}$$

Here $s_*^{\mathbb{P}}$ denotes the true score function and $t \sim \mathrm{Unif}([T_0, T])$. We also define $\ell^{\mathbb{P}}(x, y, s) := \ell(x, y, s) - \ell(x, y, s_*^{\mathbb{P}})$. In practice, with a score estimator $\widehat{s}$, the generative process is to simulate

$$\mathrm{d}\widehat{X}_t^{\leftarrow} = (\widehat{X}_t^{\leftarrow} + 2\widehat{s}(\widehat{X}_t^{\leftarrow}, y, T - t))\mathrm{d}t + \sqrt{2}\mathrm{d}\overline{W}_t, \widehat{X}_0^{\leftarrow} \sim \mathcal{N}(0, I), 0 \le t \le T - T_0. \tag{2.5}$$

Here $T_0 > 0$ is the early-stopping time. And the distribution of $\widehat{X}_{T-T_0}^{\leftarrow}$ is written as $\widehat{\mathbb{P}}(\cdot|y)$.

Note that we don't apply the commonly used classifier-free guidance [Ho and Salimans, 2022] which has a tunable guidance strength since we mainly concentrate on sampling from conditional distribution instead of optimizing other objectives.

## 2.2 Transfer Diffusion Models via Learning Representation

Consider $K$ source distributions over $\mathbb{R}^{d_x} \times [0, 1]^{D_y}$, $\mathbb{P}^1, \cdots, \mathbb{P}^K$, and a target distribution $\mathbb{P}^0$. Suppose that for each source distribution $\mathbb{P}^k, 1 \le k \le K$, we have $n$ *i.i.d.* samples $\{(x_i^k, y_i^k)\}_{i=1}^n \sim \mathbb{P}^k$, and $m$ *i.i.d.* samples $\{(x_i^0, y_i^0)\}_{i=1}^m \sim \mathbb{P}^0$ are available for the target distribution, where typically $m \ll n$. In transfer (meta) learning setup, we assume there exists a shared nonlinear representation of the condition $y$ for all distributions, *i.e.*, the conditional distribution $\mathbb{P}_{x|y}^k = \mathbb{P}_{x|h_*(y)}^k$ for some $h_* : [0, 1]^{D_y} \to [0, 1]^{d_y}$ (see also Assumption 3.2). Note that due to the shared features, the score of $p_t^k(\cdot|y)$ also has the form of $\nabla \log p_t^k(x_t|y) = f_*^k(x_t, h_*(y), t)$ for some $f_*^k$.

Similar to Tripuraneni et al. [2020], our transfer learning procedures consist of two phases. In the pre-training phase, the goal is to learn a representation map $h_*$ through $nK$ samples from $K$ source distributions. Then during the fine-tuning phase, we learn the target distribution via $m$ new samples and the representation map learned in the pre-training phase.

Formally, let $\mathcal{F}, \mathcal{H}$ be the hypothesis classes of score networks and representation maps, respectively. Further let $\mathcal{F}^0 \subseteq \mathcal{F}$ be the hypothesis class of score network in fine-tuning phase. In the pre-training phase, we solve the following Empirical Risk Minimization (ERM),

$$\widehat{\boldsymbol{f}}, \widehat{h} = \underset{\boldsymbol{f} \in \mathcal{F}^{\otimes K}, h \in \mathcal{H}}{\arg\min} \frac{1}{nK} \sum_{k=1}^{K} \sum_{i=1}^{n} \ell(x_i^k, y_i^k, s_{f^k, h}). \tag{2.6}$$

Then for the fine-tuning task, we solve

$$\widehat{f}^0 := \underset{f \in \mathcal{F}^0}{\arg\min} \frac{1}{m} \sum_{i=1}^{m} \ell(x_i^0, y_i^0, s_{f, \widehat{h}}). \tag{2.7}$$

Here $s_{f,h}(x, y, t) := f(x, h(y), t)$ for $f : \mathbb{R}^{d_x} \times [0, 1]^{d_y} \times [T_0, T] \to \mathbb{R}^{d_x}$ and $h : [0, 1]^{D_y} \to [0, 1]^{d_y}$ and $\ell$ is defined in (2.3).

In the meta-learning setting, we further assume that all the distributions $\{\mathbb{P}^k\}_k$ are *i.i.d.* sampled from a meta distribution $\mathbb{P}_{\text{meta}}$. Here $\mathbb{P}_{\text{meta}}$ can be interpreted as a universal *environment* [Baxter, 2000, Maurer et al., 2016]. In this case, we posit the existence of a shared representation map that holds for all $\mathbb{P} \sim \mathbb{P}_{\text{meta}}$. And the performance benchmark is then defined as the expected error on the target distribution $\mathbb{P}^0 \sim \mathbb{P}_{\text{meta}}$.

## 2.3 Deep ReLU Neural Network Family

We use feedforward neural networks to approximate the score function and representation map. Let $\sigma(x) := \max\{x, 0\}$ be the ReLU activation. Define the score network family $NN_f(L, W, M, S, B, R, \gamma) := \big\{f(x, w, t) = (A_L \sigma(\cdot) + b_L) \circ \cdots \circ (A_1 [x^\top, w^\top, t]^\top + b_1) :$
$A_i \in \mathbb{R}^{d_i \times d_{i+1}}, b_i \in \mathbb{R}^{d_{i+1}}, d_{L+1} = d_x, \max d_i \leq W, \|f\|_{L^\infty} \leq M, \sum_{i=1}^{L} (\|A_i\|_0 + \|b_i\|_0) \leq$
$S, \max \|A_i\|_\infty \vee \|b_i\|_\infty \leq B, \|f(x, w, t) - f(x, w', t)\| \leq \gamma \|w - w'\|_\infty, \forall \|x\|_\infty \leq R, t \leq T\big\}$,
and encoder network $NN_h(L, W, S, B) := \big\{h(y) = (A_L \sigma(\cdot) + b_L) \circ \cdots \circ (A_1 y + b_1) :$
$A_i \in \mathbb{R}^{d_i \times d_{i+1}}, b_i \in \mathbb{R}^{d_{i+1}}, d_{L+1} = d_y, \max d_i \leq W, \|h\|_{L^\infty([0,1]^{D_y})} \leq 1, \sum_{i=1}^{L} (\|A_i\|_0 +$
$\|b_i\|_0) \leq S, \max \|A_i\|_\infty \vee \|b_i\|_\infty \leq B\big\}$. Throughout this paper, we let $\mathcal{F}^0 = \mathcal{F} = NN_f(L_f, W_f, M_f, S_f, B_f, R_f, \gamma_f)$ and $\mathcal{H} = NN_h(L_h, W_h, S_h, B_h)$ unless otherwise specified.

**Remark 1.** *In practice, $\mathcal{F}^0 \subseteq \mathcal{F}$ may (and typically will) depend on $\widehat{f}$ for parameter efficient fine-tuning (PEFT), e.g., LoRA [Hu et al., 2021]. This will substantially reduce the complexity of $\mathcal{F}^0$ and further improve sample efficiency. The analysis of PEFT is beyond the scope of this paper.*

# 3 Statistical Guarantees for Transferring Score Matching Error

In this section, we present our main theoretical results, a statistical theory of transferring the conditional score matching loss. We provide two upper bounds of the score matching loss on target distribution, based on whether task diversity [Tripuraneni et al., 2020] is explicitly assumed. Our analysis introduces novel techniques to address the smoothness properties of the noised data distribution—a challenge that remains nontrivial even in single-task settings. Additionally, we extend the classical theory of local Rademacher complexity to quantify the empirical estimation error.

Throughout this paper, we make the following standard and mild regularity assumptions [Tripuraneni et al., 2020, Chen et al., 2023b] on the initial data distribution $\mathbb{P}$ and the representation map $h_*$.

**Assumption 3.1** (Sub-gaussian tail). *For any source or target distribution $\mathbb{P}$, $\mathbb{P}$ is supported on $\mathbb{R}^{d_x} \times [0,1]^{D_y}$ and admits a continuous density $p(x, y) \in \mathcal{C}^2(\mathbb{R}^{d_x} \times [0,1]^{D_y})$. Moreover, the conditional distribution $p(x|y) \leq C_1 \exp(-C_2 \|x\|^2)$ for some constant $C_1, C_2$.*

**Assumption 3.2** (Shared low-dimensional representation). *There exists an $L$-Lipschitz function $h_* : [0,1]^{D_y} \to [0,1]^{d_y}$ with $d_y \leq D_y$, such that for any source and target distribution $\mathbb{P}$, the conditional density $p(x|y) = g_*^{\mathbb{P}}(x, h_*(y))$ for some $g_*^{\mathbb{P}} \in \mathcal{C}^2(\mathbb{R}^{d_x} \times [0,1]^{d_y})$.*

Equivalently, $h_*(y)$ is a sufficient statistic for $x$, which indicates that $p_t(x|y) = p_t(x|h_*(y))$. Therefore, with a little abuse of notation, for any $w \in [0,1]^{d_y}$, we define $p(x; w) = p(x|h_*(y) = w) = g_*^{\mathbb{P}}(x, w)$. Also note that by definition, for any $x, y$, we have $p(x; h_*(y)) = p(x|h_*(y)) = p(x|y)$.

**Assumption 3.3** (Lipschitz score). *For any source and target distribution $\mathbb{P}$ and its density function $p$, the conditional score $\nabla_x \log p(x|y) = \nabla_x \log g_*^{\mathbb{P}}(x, h_*(y))$. The score function $\nabla_x \log g_*^{\mathbb{P}}(x, w)$ is $L$-Lipschitz in $x$ and $w$. And $\|\nabla_x \log g_*^{\mathbb{P}}(0, w)\| \leq B$ for some constant $B$ and any $w$.*

## 3.1 Tackling Lipschitz Continuity under Weaker Assumptions

Notice that we only impose smoothness assumption on the original data distribution $p(\cdot|y)$, instead of the entire trajectory $p_t(\cdot|y)$ in forward process. This is substantially weaker than the Lipschitzness assumption required in Chen et al. [2023b, 2022b], Yuan et al. [2024], Yang et al. [2024]. However, Lipschitzness of loss function $\ell$ and class $\mathcal{F}$ is a crucial hypothesis in theoretical analysis of transfer learning [Tripuraneni et al., 2020, Chua et al., 2021]. The intuition is that without Lipschitz continuity of the score network $f$, it is generally impossible to characterize the error from an imperfect representation map $h$. Hence it is inevitable to show the smoothness of $p_t(\cdot|y)$ to some extent.

Fortunately, even with assumptions merely on the initial data distribution, we are still able to prove smoothness of the forward process in any bounded region, as shown in the following lemma. The proof can be found in Appendix B.1.

**Lemma 3.1.** *Under Assumption 3.1, 3.2, 3.3, for any $w \in [0,1]^{d_y}$, denote the conditional score of forward process $\nabla_x \log p_t(x; w)$ by $f_*(x, w, t)$. There exist constants $C_X, C_X'$, such that for any $R > 0$, the function $f_*(x, w, t)$ is $(C_X + C_X' R^2)$-Lipschitz in $x$, $(C_X + C_X' R)$-Lipschitz in $w$, in the domain $\mathcal{B}_R \times [0,1]^{d_y} \times [0, T]$. Here $\mathcal{B}_R$ denotes the ball with radius $R$ centered at the origin.*

## 3.2 Results under Task Diversity: Sample-Efficient Transfer Learning

In the literature of transfer learning, task diversity is an important assumption that connects target tasks with source tasks [Tripuraneni et al., 2020, Du et al., 2020, Chua et al., 2021]. In the context of conditional diffusion models, we state the formal definition as follows.

**Definition 3.1** (Task diversity). Given hypothesis classes $\mathcal{F}, \mathcal{H}$, we say the source distributions $\mathbb{P}^1, \cdots, \mathbb{P}^K$ are $(\nu, \Delta)$-diverse over target distribution $\mathbb{P}^0$, if for any representation $h \in \mathcal{H}$,

$$\inf_{f^0 \in \mathcal{F}^0} L^{\mathbb{P}^0}(s_{f^0, h}) \leq \frac{1}{\nu} \inf_{\boldsymbol{f} \in \mathcal{F}^{\otimes K}} \frac{1}{K} \sum_{k=1}^K L^{\mathbb{P}^k}(s_{f^k, h}) + \Delta. \tag{3.1}$$

Here $L^{\mathbb{P}}$ is defined in (2.4). This notion of diversity ensures that the representation error on the target task caused by $\widehat{h}$ can be controlled by the error on the source tasks, thereby establishing certain relationships in between. More detailed discussions are deferred to Appendix B.5.

We first present the generalization guarantee for each phase respectively.

**Proposition 3.2** (Fine-tuning phase generalization). *Under Assumption 3.1, 3.2, 3.3, for any $\widehat{h} \in \mathcal{H}$, the population loss of $\widehat{f}^0$ can be bounded by*

$$\mathbb{E}_{\{(x_i, y_i)\}_{i=1}^m \sim \mathbb{P}^0} \mathbb{E}_{(x,y) \sim \mathbb{P}^0}[\ell^{\mathbb{P}^0}(x, y, s_{\widehat{f}^0, \widehat{h}})] \lesssim \inf_{f \in \mathcal{F}} \mathbb{E}_{(x,y) \sim \mathbb{P}^0}[\ell^{\mathbb{P}^0}(x, y, s_{f, \widehat{h}})] + \log^3(m) r_x, \tag{3.2}$$

*where $r_x = \dfrac{\log \widetilde{\mathcal{N}}_{\mathcal{F}}}{m}$ and $\log \widetilde{\mathcal{N}}_{\mathcal{F}}$ is some complexity measures of $\mathcal{F}$.*

**Proposition 3.3** (Pre-training phase generalization). *Under Assumption 3.1, 3.2, 3.3, if $R_f \gtrsim \log^{\frac{1}{2}}(nKM_f/\delta)$, with probability no less than $1 - \delta$, the population loss can be bounded by*

$$\frac{1}{K} \sum_{k=1}^K \mathbb{E}_{(x,y) \sim \mathbb{P}^k} \ell^{\mathbb{P}^k}(x, y, s_{\widehat{f}^k, \widehat{h}}) \lesssim \inf_{\boldsymbol{f} \in \mathcal{F}^{\otimes K}, h \in \mathcal{H}} \frac{1}{K} \sum_{k=1}^K \mathbb{E}_{(x,y) \sim \mathbb{P}^k}[\ell^{\mathbb{P}}(x, y, s_{f^k, h})] + \log^3(nK/\delta) \left(r_z + \frac{\log(1/\delta)}{nK}\right),$$
$$\tag{3.3}$$

*where $r_z := \dfrac{K \log \widetilde{\mathcal{N}}_{\mathcal{F}} + \log \widetilde{\mathcal{N}}_{\mathcal{H}}}{nK}$ and $\log \widetilde{\mathcal{N}}_{\mathcal{F}}, \log \widetilde{\mathcal{N}}_{\mathcal{H}}$ are some complexity measures of $\mathcal{F}, \mathcal{H}$.*

Combining these two propositions with the notion of task diversity in Definition 3.1, we are able to show the statistical rate of transfer learning as follows.

**Theorem 3.4.** *Under Assumption 3.1, 3.2, 3.3, suppose $\mathbb{P}^1, \cdots, \mathbb{P}^K$ are $(\nu, \Delta)$-diverse over target distribution $\mathbb{P}^0$ given $\mathcal{F}, \mathcal{H}$. If $R_f \gtrsim \log^{\frac{1}{2}}(nKM_f/\delta)$, then with probability no less than $1 - \delta$,*

$$\mathbb{E}_{\{(x_i, y_i)\}_{i=1}^m} \mathbb{E}_{(x,y) \sim \mathbb{P}^0}[\ell^{\mathbb{P}^0}(x, y, s_{\widehat{f}^0, \widehat{h}})] \lesssim \frac{1}{\nu} \inf_{h \in \mathcal{H}} \frac{1}{K} \sum_{k=1}^K \inf_{f \in \mathcal{F}} \mathbb{E}_{(x,y) \sim \mathbb{P}^k}[\ell^{\mathbb{P}^k}(x, y, s_{f, h})] + \Delta$$
$$+ \frac{\log^3(m) \log \mathcal{N}_{\mathcal{F}}}{m} + \frac{\log^3(nK/\delta)(K \log \mathcal{N}_{\mathcal{F}} + \log(\mathcal{N}_{\mathcal{H}}/\delta))}{\nu nK}. \tag{3.4}$$

*where*

$$\log \mathcal{N}_{\mathcal{F}} := M_f^2 S_f L_f \log\left(mn L_f W_f (B_f \vee 1) M_f T \log(1/\delta)\right),$$
$$\log \mathcal{N}_{\mathcal{H}} := S_h L_h \log\left(nK L_h W_h (B_h \vee 1) M_f \gamma_f \log(1/\delta)\right). \tag{3.5}$$

The formal statements and proofs are provided in Appendix B.2.

Let $\varepsilon_{\text{approx}} = \inf_{h \in \mathcal{H}} \frac{1}{K} \sum_{k=1}^{K} \inf_{f \in \mathcal{F}} \mathbb{E}_{(x,y) \sim \mathbb{P}^k}[\ell^{\mathbb{P}^k}(x, y, s_{f,h})]$ be the approximation error. The leading terms can be simplified to $\widetilde{\mathcal{O}}\left(\varepsilon_{\text{approx}} + \frac{K \log \mathcal{N}_{\mathcal{F}} + \log \mathcal{N}_{\mathcal{H}}}{nK} + \frac{\log \mathcal{N}_{\mathcal{F}}}{m}\right)$, where $\log \mathcal{N}_{\mathcal{F}}$ and $\log \mathcal{N}_{\mathcal{H}}$ capture the complexity of the hypothesis classes.

**Improving Sample Efficiency**    Theorem 3.4 demonstrates the sample efficiency of transfer learning. Compared to naively training the full CDM for target distribution, which has an error of $\widetilde{\mathcal{O}}\left(\varepsilon_{\text{approx}} + \frac{\log \mathcal{N}_{\mathcal{F}} + \log \mathcal{N}_{\mathcal{H}}}{m}\right)$, transfer learning saves the complexity of learning $\mathcal{H}$ and thus the performance is much better when $m$ is relatively small to $n, K$ (*i.e.*, in few-shot learning setting).

### 3.3   Results without Task Diversity: Meta-Learning Perspective

The results in previous section heavily depend on the task diversity assumption, which is hard to verify in practice. An alternative is to consider meta-learning setting, where all source and target distributions are sampled from the same *environment*, *i.e.*, a meta distribution.

For any $h \in \mathcal{C}([0, 1]^{D_y}; [0, 1]^{d_y})$ and distribution $\mathbb{P}$ over $\mathbb{R}^{d_x} \times [0, 1]^{D_y}$, define the representation error as

$$\mathcal{L}(\mathbb{P}, h) := \inf_{f \in \mathcal{F}} \mathbb{E}_{(x,y) \sim \mathbb{P}}[\ell^{\mathbb{P}}(x, y, s_{f,h})] \geq 0. \tag{3.6}$$

We characterize the generalization bound of source tasks on the entire meta distribution as follows.

**Proposition 3.5** (Generalization on meta distribution). *Under Assumption 3.1, 3.2, 3.3, there exists constant $C_P$ such that for $\{\mathbb{P}^k\}_{k=1}^{K} \overset{i.i.d.}{\sim} \mathbb{P}_{meta}$, with probability no less than $1 - \delta$,*

$$\mathbb{E}_{\mathbb{P} \sim \mathbb{P}_{meta}} \mathcal{L}(\mathbb{P}, h) \leq \frac{2}{K} \sum_{k=1}^{K} \mathcal{L}(\mathbb{P}^k, h) + C_P\left(r_P + \frac{\log(1/\delta)}{K}\right), \tag{3.7}$$

$$\frac{1}{K} \sum_{k=1}^{K} \mathcal{L}(\mathbb{P}^k, h) \leq 2\mathbb{E}_{\mathbb{P} \sim \mathbb{P}_{meta}} \mathcal{L}(\mathbb{P}, h) + C_P\left(r_P + \frac{\log(1/\delta)}{K}\right), \tag{3.8}$$

*holds for any $h \in \mathcal{H}$, where $r_P = M_f^2 \exp(-\Omega(R_f^2)) + \frac{S_h L_h \log\left(K L_h W_h (B_h \vee 1) M_f \gamma_f\right)}{K}$.*

**Theorem 3.6.** *Under Assumption 3.1, 3.2, 3.3, if $R_f \gtrsim \log^{\frac{1}{2}}(nK M_f / \delta)$, then with probability no less than $1 - \delta$, the expected population loss of new task can be bounded by*

$$\mathbb{E}_{\mathbb{P}^0 \sim \mathbb{P}_{meta}} \mathbb{E}_{\{(x_i, y_i)\}_{i=1}^{m} \sim \mathbb{P}^0} \mathbb{E}_{(x,y) \sim \mathbb{P}^0}[\ell^{\mathbb{P}}(x, y, s_{\widehat{f}^0, \widehat{h}})]$$

$$\lesssim \inf_{h \in \mathcal{H}} \mathbb{E}_{\mathbb{P} \sim \mathbb{P}_{meta}} \inf_{f \in \mathcal{F}} \mathbb{E}_{(x,y) \sim \mathbb{P}}[\ell^{\mathbb{P}}(x, y, s_{f,h})] + \frac{\log^3(m) \log \mathcal{N}_{\mathcal{F}}}{m} + \frac{\log^3(nK/\delta) \log \mathcal{N}_{\mathcal{F}}}{n} + \frac{\log(\mathcal{N}_{\mathcal{H}}/\delta)}{K}, \tag{3.9}$$

*where $\log \mathcal{N}_{\mathcal{F}}, \log \mathcal{N}_{\mathcal{H}}$ are defined in (3.5).*

The formal statements and proofs are provided in Appendix B.3.

Let $\widetilde{\varepsilon}_{\text{approx}} = \inf_{h \in \mathcal{H}} \mathbb{E}_{\mathbb{P} \sim \mathbb{P}_{meta}} \inf_{f \in \mathcal{F}} \mathbb{E}_{(x,y) \sim \mathbb{P}}[\ell^{\mathbb{P}}(x, y, s_{f,h})]$ be the approximation error in meta-learning.

The results above can be further simplified to $\widetilde{\mathcal{O}}\left(\widetilde{\varepsilon}_{\text{approx}} + \frac{\log \mathcal{N}_{\mathcal{F}}}{m \wedge n} + \frac{\log \mathcal{N}_{\mathcal{H}}}{K}\right)$. Different from transfer learning bound in Theorem 3.4, the leading term decays only in $K$ and not in $n$. This is because that without task diversity assumption, the connection between source distributions and target distributions can only be constructed through meta distribution. And according to Proposition 3.5, the source distributions $\mathbb{P}^1, \cdots, \mathbb{P}^K$ collectively form a $K$-shot empirical estimation of $\mathbb{P}_{meta}$, leading to an estimation error of $\mathcal{O}(1/K)$. Despite this, Theorem 3.6 still demonstrates the sample efficiency of meta-learning compared to naive training method when $m$ is small and $n, K$ are sufficient large.

# 4 End-to-End Distribution Estimation via Deep Neural Network

Section 3 provides a statistical guarantee for transferring score matching. In this section, we establish an approximation theory using deep neural network to quantify the misspecification error. Combining both results we are able to obtain an end-to-end distribution estimation error bound for transfer learning diffusion models.

## 4.1 Score Neural Network Approximation

The following theorem provides a guarantee for the ability of deep ReLU neural networks to approximate score and representation. The proof is provided in Appendix C.1.

**Theorem 4.1.** *Under Assumption 3.1, 3.2, 3.3, to achieve $R_f \gtrsim \log^{\frac{1}{2}}(nKM_f/\delta)$ and*

$$\inf_{h \in \mathcal{H}} \frac{1}{K} \sum_{k=1}^{K} \inf_{f \in \mathcal{F}} \mathbb{E}_{(x,y) \sim \mathbb{P}^k}[\ell^{\mathbb{P}^k}(x,y,s_{f,h})] = \mathcal{O}\left(\log^2(nK/(\varepsilon\delta))\varepsilon^2\right), \quad \textit{(transfer learning)} \quad (4.1)$$

$$\inf_{h \in \mathcal{H}} \mathbb{E}_{\mathbb{P} \sim \mathbb{P}_{meta}} \inf_{f \in \mathcal{F}} \mathbb{E}_{(x,y) \sim \mathbb{P}}[\ell^{\mathbb{P}}(x,y,s_{f,h})] = \mathcal{O}\left(\log^2(nK/(\varepsilon\delta))\varepsilon^2\right), \quad \textit{(meta-learning)} \quad (4.2)$$

*the configuration of $\mathcal{F}$ and $\mathcal{H}$ should satisfy*

$$L_f = \mathcal{O}\left(\log\left(\frac{\log(nK/(\varepsilon\delta))}{\varepsilon}\right)\right), W_f = \mathcal{O}\left(\frac{\log^{3(d_x+d_y)/2}(nK/(\varepsilon\delta))}{\varepsilon^{d_x+d_y+1}T_0^3}\right),$$

$$S_f = \mathcal{O}\left(\frac{\log^{3(d_x+d_y)/2+1}(nK/(\varepsilon\delta))}{\varepsilon^{d_x+d_y+1}T_0^3}\right), B_f = \mathcal{O}\left(\frac{T\log^{\frac{3}{2}}(nK/(\varepsilon\delta))}{\varepsilon}\right), \quad (4.3)$$

$$R_f = \mathcal{O}\left(\log^{\frac{1}{2}}(nK/(\varepsilon\delta))\right), M_f = \mathcal{O}\left(\log^3(nK/(\varepsilon\delta))\right), \gamma_f = \mathcal{O}\left(\log(nK/(\varepsilon\delta))\right),$$

$$L_h = \mathcal{O}\left(\log(1/\varepsilon)\right), W_h = \mathcal{O}\left(\varepsilon^{-D_y}\log(1/\varepsilon)\right), S_h = \mathcal{O}\left(\varepsilon^{-D_y}\log^2(1/\varepsilon)\right), B_h = \mathcal{O}(1). \quad (4.4)$$

*Here $\mathcal{O}(\cdot)$ hides all the polynomial factors of $d_x, d_y, D_y, C_1, C_2, L, B$.*

Universal approximation of deep ReLU neural networks in a bounded region has been widely studied [Yarotsky, 2017, Schmidt-Hieber, 2020]. However, we have to deal with an unbounded domain here, hence more refined analysis is required, e.g. truncation arguments.

In addition, traditional approximation theories typically cannot provide Lipschitz continuity guarantees, which is crucial in transfer learning analysis. Following the constructions in Chen et al. [2023b], the Lipschitzness restriction doesn't compromise the approximation ability of neural networks, while ensuring validity of the generalization analysis in Section 3.

## 4.2 Distribution Estimation Error Bound

Given the approximation and generalization results, we are in the position of bounding the distribution estimation error of our transfer (meta) learning procedures. The formal statements and proofs can be found in Appendix C.2.

**Theorem 4.2** (Transfer learning). *Under Assumption 3.1, 3.2, 3.3 and $(\nu, \Delta)$-diversity with proper configuration of neural network family and $T, T_0$, it holds that with probability at least $1 - \delta$,*

$$\mathbb{E}_{\{(x_i,y_i)\}_{i=1}^m \sim \mathbb{P}^0}\mathbb{E}_{y \sim \mathbb{P}^0_y}[\mathrm{TV}(\widehat{\mathbb{P}}^0_{x|y}, \mathbb{P}^0_{x|y})] \lesssim \frac{\log^{\frac{5}{2}}(nK/\delta)\log^3((m/\nu) \wedge n)}{\nu^{\frac{1}{2}}((m/\nu) \wedge n)^{\frac{1}{d_x+d_y+9}}} + \frac{\log^2(nK/\delta)}{\nu^{\frac{1}{2}}(nK)^{\frac{1}{D_y+2}}} + \sqrt{\Delta}. \quad (4.5)$$

**Theorem 4.3** (Meta-learning). *Under Assumption 3.1, 3.2, 3.3 and meta-learning setting, with proper configuration of neural network family and $T, T_0$, it holds that with probability at least $1 - \delta$,*

$$\mathbb{E}_{\mathbb{P}^0 \sim \mathbb{P}_{meta}}\mathbb{E}_{\{(x_i,y_i)\}_{i=1}^m \sim \mathbb{P}^0}\mathbb{E}_{y \sim \mathbb{P}^0_y}[\mathrm{TV}(\widehat{\mathbb{P}}^0_{x|y}, \mathbb{P}^0_{x|y})] \lesssim \frac{\log^{\frac{5}{2}}(nK/\delta)\log^3(m \wedge n)}{(m \wedge n)^{\frac{1}{d_x+d_y+9}}} + \frac{\log^2(nK/\delta)}{K^{\frac{1}{D_y+2}}}. \quad (4.6)$$

| $m$ | 10 | 20 | 30 | 40 | 50 | 100 |
|---|---|---|---|---|---|---|
| fine-tuning | 14.47 | 3.68 | 2.45 | 1.82 | 1.9 | 0.91 |
| train-from-scratch | 21.99 | 10.61 | 5.71 | 2.38 | 1.77 | 1.04 |

Table 2: MSEs for $\beta_0 = 5.5$.

| $m$ | 10 | 20 | 30 | 40 | 50 | 100 |
|---|---|---|---|---|---|---|
| fine-tuning | 6.14 | 2.65 | 1.61 | 1.08 | 0.96 | 0.45 |
| train-from-scratch | 24.41 | 20.62 | 18.67 | 13.49 | 7.03 | 1.23 |

Table 3: MSEs for $\beta_0 = 15$.

Theorem 4.2 and 4.3 again unveil the benefits of transfer (meta) learning for conditional diffusion models, with a rate of $\widetilde{\mathcal{O}}((m \wedge n)^{-\frac{1}{d_x+d_y+9}} + (nK)^{-\frac{1}{D_y+2}})$ or $\widetilde{\mathcal{O}}((m \wedge n)^{-\frac{1}{d_x+d_y+9}} + K^{-\frac{1}{D_y+2}})$. To compare, naively learning the target distribution in isolation will yield $\widetilde{\mathcal{O}}(m^{-\frac{1}{d_x+D_y+9}})$. When the condition dimension $D_y$ is much larger than feature dimension $d_y$, transfer (meta) learning can substantially improve sample efficiency on target tasks, thanks to representation learning.

**Comparison with Existing Complexity Bounds of CDMs** Fu et al. [2024] studies conditional diffusion model for sub-gaussian distributions with $\beta$-Hölder density. Since the Lipschitzness of score is analogous to the requirement of twice differentiability of density [Wibisono et al., 2024], it is reasonable to let $\beta = 2$ for a fair comparison. In this case, the TV distance is bounded by $\widetilde{\mathcal{O}}(m^{-\frac{1}{2(d_x+D_y+2)}})$ with sample size $m$ according to Fu et al. [2024], which is worse than our naive bound $\widetilde{\mathcal{O}}(m^{-\frac{1}{d_x+D_y+9}})$ due to the inefficiency of score approximation. We are also aware of another work [Jiao et al., 2024] that assumes Lipschitz density and score, obtaining a rate of $\widetilde{\mathcal{O}}(m^{-\frac{1}{2(d_x+3)(d_x+D_y+3)}})$.

**Relation to Yang et al. [2024]** Unlike our setup, Yang et al. [2024] considers transfer learning unconditional diffusion models with only one source task, i.e., $D_y = d_y = 0, K = 1$. The unconditional distribution is assumed to be supported in a low-dimensional linear subspace, where the source task and the target task have the same latent variable distribution. Hence, *only a linear encoder* is trained for fine-tuning instead of the full score network. In this case, Yang et al. [2024] is able to bound the TV distance by $\widetilde{\mathcal{O}}(m^{-\frac{1}{4}} + n^{-\frac{1-\alpha(n)}{d_x+5}})$, escaping the curse of dimensionality for target task. However, the assumption on shared latent variable distribution is stringent and we believe our analysis methods can be extended to this setting as well.

## 5  Experiments

Our theoretical results can be readily applied in various real world settings. In Appendix A, we investigate amortized variational inference and behavior cloning utilizing our theories, providing statistical guarantees of practical applications of CDMs. In addition, we conduct experiments on both synthetic and real world data to numerically verify the sample efficiency of transfer learning.

**Conditioned Diffusion** The first numerical example is the high-dimensional conditioned diffusion [Cui et al., 2016, Yu et al., 2023] arising from the following Langevin SDE

$$\mathrm{d}u_s = \beta u_s(1 - u_s^2)\mathrm{d}s + \mathrm{d}w_s, \; u_0 = 0, \tag{5.1}$$

where $\beta > 0$ and $w_s$ is a one-dimensional standard Brownian motion. The SDE (5.1) is discretized by the Euler-Maruyama scheme with a step size of $0.02$, which defines the prior distribution $p_\beta(x)$ for the (discretized) trajectory $x = (u_{0.02}, u_{0.04}, \ldots, u_{1.00})^\top \in \mathbb{R}^{50}$. We consider a conditional Gaussian likelihood function, $p(y|x) = \mathcal{N}(Mx, I_{100}/4)$, where $M \in \mathbb{R}^{100 \times 50}$ is a pre-defined projection matrix. Given a set of pre-selected $\{\beta_k; 1 \le k \le K\}$ with $\beta_k = k$ and $K = 10$, the $k$-th joint source distribution is given by $\mathbb{P}^k(x, y) = p_{\beta_k}(x)p(y|x)$. The target distribution $\mathbb{P}^0(x, y)$ is given by $\beta_0 = 5.5$ (in-domain) or $\beta_0 = 15$ (out-of-domain). More details are found in Appendix E.1.

We report the MSEs of the estimated posterior mean of $\mathbb{P}^0(x|y)$ on the test samples in Table 2 and 3. We see that across different values of $\beta$ and $m$, the fine-tuned models can provide significantly more accurate posterior mean estimations in most cases, suggesting the effectiveness of the representation map $\widehat{h}$ learned in the pre-training phase. Notably, as the number of fine-tuning samples $m$ increases, the performance gaps between fine-tuned models and train-from-scratch models get smaller, since more training samples yield more generalization benefits and thus less dependence on the pre-trained

| $m$ | 10 | 20 | 30 | 40 | 50 | 100 |
|---|---|---|---|---|---|---|
| fine-tuning | 0.3799 | 0.2846 | 0.2544 | 0.2406 | 0.2404 | 0.2268 |
| train-from-scratch | 0.4409 | 0.3180 | 0.2746 | 0.2551 | 0.2501 | 0.2344 |

Table 4: MSEs on the image restoration task.

model. This is aligned with our theoretical results. We also notice a large variance among the results of different replicates, and attribute the slightly worse performance of fine-tuned models at $m = 50, \beta_0 = 5.5$ to the potential randomness.

**Image Restoration** For a real data experiment, we consider the image restoration task on MNIST. We have $K = 9$ source tasks with $\mathbb{P}^k(x, y) = p_k(x)p(y|x)$, where the prior $p_k(x)$ is the data distribution of the digit $k$ in the MNIST data set ($1 \leq k \leq K$) and $p(y|x) = \mathcal{N}(x, I_{784}/4)$. The target task is $\mathbb{P}^0(x, y) = p_0(x)p(y|x)$, where $p_0(x)$ is the data distribution of the digit 0. We use the full MNIST 1-9 data for pre-training which corresponds to $n = 5000$. For the finetuning phase, we consider $m = 10, 20, 30, 40, 50, 100$ training samples and 100 test samples from $\mathbb{P}^0(x, y)$. More details can be found in Appendix E.2.

We report the MSEs between estimated posterior mean of $\mathbb{P}^0(x, y) = p_0(x)p(y|x)$ and the ground truth sample $x$ on the test samples in Table 4. We see that for all fine-tuning sample sizes $m$, the results obtained by fine-tuning consistently outperform those obtained by training from scratch, indicating the benefits of transfer learning. Similarly to the experiment on conditioned diffusion, we also observe a reduced performance gap as $m$ increases.

## 6    Conclusion and Discussion

In this paper, we take the first step towards understanding the sample efficiency of transfer learning conditional diffusion models from the perspective of representation learning. We provide a generalization guarantee for transferring score matching in CDMs in different settings. We further establish an end-to-end distribution estimation error bound using deep neural networks. Two practical applications are investigated based on our theoretical results. We hope this work can motivate future theoretical study on the popular transfer learning paradigm in generative AIs.

Although this work provides the first statistical guarantee for transfer learning in CDMs, it has several limitations that we plan to address in future research. First, our theoretical results heavily rely on the task diversity notion introduced in Section 3.1, which can be challenging to verify in practice. While we provide some preliminary empirical evidence in Appendix B.5, a more fine-grained theoretical and empirical analysis will be essential for a deeper understanding of CDMs. Second, our analysis focuses on the ERM estimator, whereas in practice, fine-tuning typically starts from a pre-trained model and may employ techniques such as LoRA. Incorporating these settings would allow for an optimization-based perspective on the sample efficiency of transfer learning. Finally, in our current formulation, the sample efficiency gain arises from reducing the complexity associated with learning the conditional encoder. Consequently, our results primarily apply to CDMs in which the conditional encoder constitutes a substantial part of the overall model. Extending the theory to settings where this assumption does not hold is an important direction for future work.

## Acknowledgements

This work was supported by National Natural Science Foundation of China (grant no. 12201014, grant no. 12292980 and grant no. 12292983). The research of Cheng Zhang was support in part by National Engineering Laboratory for Big Data Analysis and Applications, the Key Laboratory of Mathematics and Its Applications (LMAM) and the Key Laboratory of Mathematical Economics and Quantitative Finance (LMEQF) of Peking University. The authors are grateful for the computational resources provided by the High-performance Computing Platform of Peking University. The authors appreciate the anonymous NeurIPS reviewers for their constructive feedback.

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

# A Applications

We explore two applications of transfer learning for conditional diffusion models, supported by theoretical guarantees derived from our earlier results. In particular, we study amortized variational inference and behavior cloning. These real-world use cases not only validate the applicability of our theoretical findings but also lay the foundations of transferring diffusion models in practice.

## A.1 Amortized Variational Inference

Diffusion models have exhibited groundbreaking success in probabilistic inference, especially latent variable models. We study a simple amortized variational inference model, where the observation $y$ given latent variable $x$ is distributed according to an exponential family $\mathcal{F}_\Psi$ with density

$$p_\psi(y|x) = \psi(y)\exp(\langle x, h_*(y)\rangle - A_\psi(x)), \tag{A.1}$$

where $\psi \in \Psi$ is non-negative and supported on $[0,1]^D$ and $h_*(y) \in [0,1]^d$. Note that we also have $d_x = d$ in this case. The prior distribution of variable $x$ is denoted as $p_\phi$ for some $\phi \in \Phi$. Let $\theta = (\psi, \phi)$ and we aim to sample from the posterior distribution of $p_\theta(x|y) \propto p_\phi(x)p_\psi(y|x) \propto p_\phi(x)\exp(\langle x, h_*(y)\rangle - A_\psi(x))$. Due to the special structure, the posterior $p_\theta(x|y)$ only depends on the low-dimensional feature $h_*(y)$, shared across all $\theta \in \Theta := \Psi \times \Phi$. This formulation encompasses various applications including independent component analysis [Comon, 1994], inverse problem [Song et al., 2021, Ajay et al., 2022] and variational Bayesian inference [Kingma, 2013].

Consider source tasks consisting of $\theta^1, \cdots, \theta^K \in \Theta$, and for each $\theta^k$ we have $n$ *i.i.d.* samples $\{(x_i^k, y_i^k)\}_{i=1}^n$. For the target task $\theta^0$, we only have $m$ samples $\{(x_i^0, y_i^0)\}_{i=1}^m$. We conduct our transfer learning procedures to train a conditional diffusion models $\widehat{\mathbb{P}}_{\theta^0}(\cdot|y)$. For theoretical analysis, we further impose some assumptions on the probabilistic model as follows.

**Assumption A.1.** *The prior distribution satisfies $p_\phi(x) \leq C_1\exp(-C_2\|x\|^2)$ and $\nabla_x \log p_\phi(x)$ is $L$-Lipschitz in $x$, $\|\nabla_x \log p_\phi(0)\| \leq B$ for any $\phi \in \Phi$. The representation $h_*$ is $L$-Lipschitz. The integral $\int \psi(y)\mathrm{d}y \in [1/C, C]$ for any $\psi \in \Psi$.*

**Theorem A.1.** *Suppose Assumption A.1 holds. Then under meta-learning setting, we have with probability no less than $1 - \delta$,*

$$\mathbb{E}_{\theta^0}\mathbb{E}_{\{(x_i^0,y_i^0)\}_{i=1}^m}\mathbb{E}_{y\sim\mathbb{P}_{\theta^0}}[\mathrm{TV}(\widehat{\mathbb{P}}_{\theta^0}(\cdot|y), \mathbb{P}_{\theta^0}(\cdot|y))] \lesssim \frac{\log^{\frac{5}{2}}(nK/\delta)\log^3(m\wedge n)}{(m\wedge n)^{\frac{1}{2d+9}}} + \frac{\log^2(nK/\delta)}{K^{\frac{1}{D+2}}}. \tag{A.2}$$

*If $(\nu, \Delta)$-diversity holds, then we have with probability no less than $1 - \delta$,*

$$\mathbb{E}_{\{(x_i^0,y_i^0)\}_{i=1}^m}\mathbb{E}_{y\sim\mathbb{P}_{\theta^0}}[\mathrm{TV}(\widehat{\mathbb{P}}_{\theta^0}(\cdot|y), \mathbb{P}_{\theta^0}(\cdot|y))] \lesssim \frac{\log^{\frac{5}{2}}(nK/\delta)\log^3((m/\nu)\wedge n)}{\nu^{\frac{1}{2}}((m/\nu)\wedge n)^{\frac{1}{2d+9}}} + \frac{\log^2(nK/\delta)}{\nu^{\frac{1}{2}}(nK)^{\frac{1}{D+2}}} + \sqrt{\Delta}. \tag{A.3}$$

The proof is deferred to Appendix D.1. We show that under mild assumptions, transfer (meta) learning diffusion models can improve the sample efficiency for target task in the context of amortized variational inference. This error bound can be further extended to establish guarantees for statistical inference such as moment prediction, uncertainty assessment, *etc.*

## A.2 Behavior Cloning via Meta-Diffusion Policy

Although originally developed for image generation tasks, diffusion models have recently been extended to reinforcement learning (RL) [Janner et al., 2022, Chi et al., 2023, Wang et al., 2022], enabling the modeling of complex distributions of dynamics and policies. In the context of meta-RL, some works have further utilized diffusion models for planning and synthesis tasks [Ni et al., 2023, He et al., 2023]. In this application, we focus on a popular framework of behavior cloning, *diffusion policy* [Chi et al., 2023], which uses conditional diffusion models to learn multi-modal expert policies in high-dimensional state spaces. In such settings, the state often corresponds to visual observations of the robot's surroundings, such as high resolution images, and thus typically share a low-dimensional underlying representation.

Let $\mathcal{M}$ be the space of decision-making environments, where each $M \in \mathcal{M}$ is an infinite horizon Markov Decision Process (MDP) sharing the same state space $\mathcal{S}$, action space $\mathcal{A}$, discount factor $\gamma$ and initial distribution $\rho \in \Delta(\mathcal{S})$. And each $M \in \mathcal{M}$ has its own transition kernel $\mathcal{T}_M : \mathcal{S} \times \mathcal{A} \to \Delta(\mathcal{S})$, and reward function $r_M : \mathcal{S} \times \mathcal{A} \to [0, 1]$. The policy is defined as a map $\pi : \mathcal{S} \to \Delta(\mathcal{A})$. The value function of MDP $M$ under policy $\pi$ is

$$V_M(\pi, s_0) := \mathbb{E}\Big[\sum_{t=0}^{\infty} \gamma^t r_M(s_t, a_t)\Big], a_t \sim \pi(\cdot|s_t), s_{t+1} \sim \mathcal{T}_M(\cdot|s_t, a_t),$$

$$V_M(\pi) := \mathbb{E}_{s_0 \sim \rho}[V_M(\pi, s_0)]. \tag{A.4}$$

Denote the visitation measure as $d_M^\pi(s, a) := (1 - \gamma)\mathbb{E}_{s_0 \sim \rho}\sum_{t=0}^{\infty}\gamma^t \mathbb{P}(s_t = s|\pi, s_0)\pi(a|s)$.

Suppose there are $K$ source tasks $M^1, \cdots, M^K \in \mathcal{M}$, and the expert policy of each task is denoted as $\pi_*^k$. In behavior cloning, for each source task $M^k$, we have $n$ pairs of $\{(s_i^k, a_i^k)\}_{i=1}^n \overset{i.i.d.}{\sim} d_*^k := d_{M^k}^{\pi_*^k}$. The goal is to imitate the expert policy of target task $M^0 \in \mathcal{M}$, of which the sample size is only $m \ll n$.

To unify the notation, let $x = a, y = s$ and assume $\mathcal{A} = \mathbb{R}^{d_a}, \mathcal{S} = [0, 1]^{D_s}$ and representation space $[0, 1]^{d_s}$. Our meta diffusion-policy framework aims to learn a state encoder $h : \mathcal{S} \to [0, 1]^{d_s}$ during pre-training, which acts as a shared representation map in different MDPs and consequently enhances sample efficiency on fine-tuning tasks. Let $\widehat{\pi}^0$ be the learned policy in fine-tuning phase. The following theorem shows the optimality gap between the learned policy and the expert policy.

**Theorem A.2.** *Suppose the expert policy $\pi_*^k$ satisfies Assumption 3.1, 3.2, 3.3. Then under meta-learning setting, it holds that with probability no less than $1 - \delta$,*

$$\mathbb{E}_{M^0}\mathbb{E}_{\{(s_i^0, a_i^0)\}_{i=1}^m \sim d_*^0}[V_{M^0}(\pi_*^0) - V_{M^0}(\widehat{\pi}^0)] \lesssim \frac{1}{(1-\gamma)^2}\left[\frac{\log^{\frac{5}{2}}(nK/\delta)\log^3(m \wedge n)}{(m \wedge n)^{\frac{1}{d_a+d_s+9}}} + \frac{\log^2(nK/\delta)}{K^{\frac{1}{D_s+2}}}\right]. \tag{A.5}$$

*If we further assume $\pi_*^1, \cdots, \pi_*^K$ are $(\nu, \Delta)$-diverse over $\pi_*^0$, then the gap can be improved by*

$$\mathbb{E}_{\{(s_i^0, a_i^0)\}_{i=1}^m \sim d_*^0}[V_{M^0}(\pi_*^0) - V_{M^0}(\widehat{\pi}^0)] \lesssim \frac{1}{(1-\gamma)^2}\left[\frac{\log^{\frac{5}{2}}(nK/\delta)\log^3((m/\nu) \wedge n)}{\nu^{\frac{1}{2}}((m/\nu) \wedge n)^{\frac{1}{d_a+d_s+9}}} + \frac{\log^2(nK/\delta)}{\nu^{\frac{1}{2}}(nK)^{\frac{1}{D_s+2}}} + \sqrt{\Delta}\right]. \tag{A.6}$$

The proof can be found in Appendix D.2. This provides the first statistical guarantee of diffusion policy in behavior cloning. Notably, in both cases, the number of source tasks $K$ has an exponential dependence on $D_s$, further suggesting the importance of data coverage when tackling distribution shift in offline meta-RL [Pong et al., 2022].

# B    Proofs in Section 3

## B.1    Preliminaries

**Lemma B.1.** *If $x_0 \sim p(x_0|y)$, the density of forward process $p_t(x|y)$ can be written as*

$$p_t(x|y) = \int \phi_t(x|x_0)p(x_0|y)\mathrm{d}x_0, \quad \phi_t(x|x_0) = \frac{1}{(2\pi\sigma_t^2)^{\frac{d_x}{2}}}\exp\left(-\frac{\|x - \alpha_t x_0\|^2}{2\sigma_t^2}\right). \tag{B.1}$$

*Besides, the score function has the form of*

$$\nabla_x \log p_t(x|y) = \int \nabla_x \log \phi_t(x|x_0)\frac{\phi_t(x|x_0)p(x_0|y)}{\int \phi_t(x|z)p(z|y)\mathrm{d}z}\mathrm{d}x_0 \tag{B.2}$$

$$= \frac{1}{\alpha_t}\int \nabla_x \log p(x_0|y)\frac{\phi_t(x|x_0)p(x_0|y)}{\int \phi_t(x|z)p(z|y)\mathrm{d}z}\mathrm{d}x_0. \tag{B.3}$$

*Proof.* (B.1) can be directly implied by the definition of forward process. And it yields

$$
\begin{aligned}
\nabla_x \log p_t(x|y) &= \frac{\nabla_x p_t(x|y)}{p_t(x|y)} \\
&= \frac{\int \nabla_x \phi_t(x|x_0) p(x_0|y) \mathrm{d}x_0}{\int \phi_t(x|x_0) p(x_0|y) \mathrm{d}x_0} \\
&= \int \nabla_x \log \phi_t(x|x_0) \frac{\phi_t(x|x_0) p(x_0|y)}{\int \phi_t(x|z) p(z|y) \mathrm{d}z} \mathrm{d}x_0,
\end{aligned} \tag{B.4}
$$

which is (B.2). Moreover, noticing that $\nabla_x \phi_t(x|x_0) = -\frac{1}{\alpha_t} \nabla_{x_0} \phi_t(x|x_0)$, then by integration by parts,

$$
\begin{aligned}
\frac{\int \nabla_x \phi_t(x|x_0) p(x_0|y) \mathrm{d}x_0}{\int \phi_t(x|x_0) p(x_0|y) \mathrm{d}x_0} &= -\frac{1}{\alpha_t} \frac{\int \nabla_{x_0} \phi_t(x|x_0) p(x_0|y) \mathrm{d}x_0}{\int \phi_t(x|x_0) p(x_0|y) \mathrm{d}x_0} \\
&= \frac{1}{\alpha_t} \frac{\int \phi_t(x|x_0) \nabla_{x_0} p(x_0|y) \mathrm{d}x_0}{\int \phi_t(x|x_0) p(x_0|y) \mathrm{d}x_0} \\
&= \frac{1}{\alpha_t} \int \nabla_x \log p(x_0|y) \frac{\phi_t(x|x_0) p(x_0|y)}{\int \phi_t(x|z) p(z|y) \mathrm{d}z} \mathrm{d}x_0.
\end{aligned} \tag{B.5}
$$

Hence (B.3) is proved. $\qquad\square$

**Lemma B.2.** *[Lem. 3.1] For any $w \in [0,1]^{d_y}$, denote the conditional score of forward process $\nabla_x \log p_t(x; w)$ by $f_*(x, w, t)$. Then there exist constants $C_X, C_X'$, such that for any $R > 0$, the function $f_*(x, w, t)$ is $(C_X + C_X' R^2)$-Lipschitz in $x$, $(C_X + C_X' R)$-Lipschitz in $w$, in the domain $\mathcal{B}_R \times [0,1]^{d_y} \times [0, T]$. Here $\mathcal{B}_R$ denotes the ball with radius $R$ centered at the origin.*

*Proof.* Define density function $q_t(x_0|x, w) \propto \phi_t(x|x_0) p(x_0; w)$. Our proof strategy will depend on whether $t \geq \frac{1}{2(L+1)}$.

When $t \geq \frac{1}{2(L+1)}$, according to (B.2), we have

$$
\begin{aligned}
\nabla_x f_*(x, w, t) &= \nabla_x^2 \log p_t(x; w) \\
&= \mathbb{E}_{q_t(x_0|x, w)} \left[ \nabla_x^2 \log \phi_t(x|x_0) \right] + \mathrm{Var}_{q_t(x_0|x, w)}(\nabla_x \log \phi_t(x|x_0)) \\
&= -\frac{I}{\sigma_t^2} + \mathrm{Var}_{q_t(x_0|x, w)} \left( \frac{\alpha_t x_0 - x}{\sigma_t^2} \right)
\end{aligned} \tag{B.6}
$$

For any $R > 0$, we have

$$
\begin{aligned}
\mathrm{Var}_{q_t(x_0|x, w)} \left( \frac{\alpha_t x_0 - x}{\sigma_t^2} \right) &\preceq \frac{1}{\sigma_t^2} \int \left\| \frac{\alpha_t x_0 - x}{\sigma_t} \right\|^2 \frac{\phi_t(x|x_0) p(x_0|y)}{\int \phi_t(x|z) p(z|y) \mathrm{d}z} \mathrm{d}x_0 \\
&\leq \frac{R^2}{\sigma_t^2} + \frac{\int_{\|\frac{\alpha_t x_0 - x}{\sigma_t}\| \geq R} \left\| \frac{\alpha_t x_0 - x}{\sigma_t} \right\|^2 \exp\left( -\frac{\|\alpha_t x_0 - x\|^2}{2\sigma_t^2} \right) p(x_0; w) \mathrm{d}x_0}{\sigma_t^2 \int \exp\left( -\frac{\|\alpha_t x_0 - x\|^2}{2\sigma_t^2} \right) p(x_0; w) \mathrm{d}x_0} \\
&\leq \frac{R^2}{\sigma_t^2} + \frac{\int_{\|\frac{\alpha_t x_0 - x}{\sigma_t}\| \geq R} \exp(-\frac{R^2}{4}) p(x_0; w) \mathrm{d}x_0}{\sigma_t^2 \int_{\|\frac{\alpha_t x_0 - x}{\sigma_t}\| \leq R/2} \exp(-\frac{R^2}{8}) p(x_0; w) \mathrm{d}x_0}.
\end{aligned} \tag{B.7}
$$

Let $R = \frac{2\|x\| + 2C_0}{\sigma_t}$, then the domain $\left\{ x_0 : \|\frac{\alpha_t x_0 - x}{\sigma_t}\| \leq R/2 \right\}$ includes $\left\{ x_0 : \|x_0\| \leq C_0 \right\}$, indicating

$$
\begin{aligned}
\int_{\|\frac{\alpha_t x_0 - x}{\sigma_t}\| \leq R/2} p(x_0; w) \mathrm{d}x_0 &\geq \int_{\|x_0\| \leq C_0} p(x_0; w) \mathrm{d}x_0 \geq 1 - 2\exp(-C_1' C_0^2) \geq \frac{1}{2}, \\
\int_{\|\frac{\alpha_t x_0 - x}{\sigma_t}\| \geq R} p(x_0; w) \mathrm{d}x_0 &\leq \int_{\|x_0\| \geq C_0} p(x_0; w) \mathrm{d}x_0 \leq \frac{1}{2}.
\end{aligned} \tag{B.8}
$$

and

$$\|\nabla_x f_*(x, w, t)\| \leq \frac{1}{\sigma_t^2} + \left\|\text{Var}_{q_t(x_0|x,w)}\left(\frac{\alpha_t x_0 - x}{\sigma_t^2}\right)\right\| \leq \frac{R^2}{\sigma_t^2} + \frac{2}{\sigma_t^2} \leq \frac{8\|x\|^2 + 8C_0^2 + 2\sigma_t^2}{\sigma_t^4}. \quad \text{(B.9)}$$

Similarly, for $w$ we have

$$\nabla_w f_*(x, w, t) = \text{Cov}_{q_t(x_0|x,w)}\left(\nabla_x \log \phi_t(x|x_0), \nabla_w \log p(x_0; w)\right)$$
$$= \text{Cov}_{q_t(x_0|x,y)}\left(\frac{\alpha_t x_0}{\sigma_t^2}, \nabla_w \log p(x_0; w)\right) \quad \text{(B.10)}$$

which implies

$$\|\nabla_w f_*(x, w, t)\| \leq B\sqrt{\left\|\text{Var}_{q_t(x_0|x,w)}\left(\frac{\alpha_t x_0 - x}{\sigma_t^2}\right)\right\|}$$
$$\leq \frac{B(2\|x\| + 2C_0 + 1)}{\sigma_t} \quad \text{(B.11)}$$

When $t \leq \frac{1}{2(L+1)}$, we have $\sigma_t^2 \leq \frac{\alpha_t^2}{2L}$ and

$$\nabla_x f_*(x, w, t) = \nabla_x^2 \log p_t(x; w)$$
$$= \frac{\nabla_x^2 p_t(x; w)}{p_t(x; w)} - \nabla_x \log p_t(x; w)(\nabla_x \log p_t(x; w))^\top$$
$$= \frac{1}{\alpha_t^2} \frac{\int \phi_t(x|x_0)\nabla_x^2 p(x_0; w)\mathrm{d}x_0}{p_t(x; w)} - \nabla_x \log p_t(x; w)(\nabla_x \log p_t(x; w))^\top$$
$$= \frac{1}{\alpha_t^2}\mathbb{E}_{q_t(x_0|x,w)}\left[\frac{\nabla_x^2 p(x_0; w)}{p(x_0; w)}\right] - \nabla_x \log p_t(x; w)(\nabla_x \log p_t(x; w))^\top$$
$$= \frac{1}{\alpha_t^2}\mathbb{E}_{q_t(x_0|x,w)}\left[\nabla_x^2 \log p(x_0; w) + \nabla_x \log p(x_0; w)(\nabla_x \log p(x_0; w))^\top\right]$$
$$\quad - \nabla_x \log p_t(x; w)(\nabla_x \log p_t(x; w))^\top$$
$$\overset{(B.3)}{=} \frac{1}{\alpha_t^2}\mathbb{E}_{q_t(x_0|x,y)}\left[\nabla_x^2 \log p(x_0; w)\right] + \frac{1}{\alpha_t^2}\text{Var}_{q_t(x_0|x,w)}\left(\nabla_x \log p(x_0; w)\right). \quad \text{(B.12)}$$

Note that when $\sigma_t^2 \leq \frac{\alpha_t^2}{2L}$, the distribution $q_t(x_0|x, w) \propto \exp\left(-\frac{\|\alpha_t x_0 - x\|^2}{2\sigma_t^2}\right)p(x_0; w)$ is $L$-strongly log-concave, and thus satisfies the Poincare inequality with a constant $L^{-1}$ [Chen et al., 2023a],

$$\text{Var}_{q_t(x_0|x,w)}\left(\nabla_x \log p(x_0; w)\right) \preceq L^{-1}\mathbb{E}\left[\nabla_x^2 \log p(x_0; w)(\nabla_x^2 \log p(x_0; w))^\top\right] \leq L. \quad \text{(B.13)}$$

And thus

$$\|\nabla_x f_*(x, w, t)\| \leq \frac{2L}{\alpha_t^2}. \quad \text{(B.14)}$$

Analogously,

$$\nabla_w f_*(x, w, t) = \frac{1}{\alpha_t}\mathbb{E}_{q_t(x_0|x,w)}\left[\nabla_w \nabla_x \log p(x_0; w)\right] + \frac{1}{\alpha_t}\text{Cov}_{q_t(x_0|x,w)}\left(\nabla_x \log p(x_0; w), \nabla_w \log p(x_0; w)\right)$$
$$\leq \frac{L}{\alpha_t} + \frac{B}{\alpha_t}\sqrt{\text{Var}_{q_t(x_0|x,w)}\left(\nabla_x \log p(x_0; w)\right)}$$
$$\leq \frac{L + B\sqrt{L}}{\alpha_t} \quad \text{(B.15)}$$

Combine all the arguments in (B.9),(B.11),(B.14),(B.15) and we complete the proof. $\square$

**Lemma B.3** (Lemma 7, Chen et al. [2022a])**.** *The covering number of* $\mathcal{F} = NN_f(L_f, W_f, M_f, S_f, B_f, R_f, \gamma_f)$ *can be bounded by*

$$\log \mathcal{N}(\mathcal{F}, \|\cdot\|_{L^\infty([-R,R]^{d_x+d_y+1})}, \varepsilon) \lesssim S_f L_f \log\left(\frac{L_f W_f (B_f \vee 1)R}{\varepsilon}\right). \quad \text{(B.16)}$$

*The covering number of $\mathcal{H} = NN_h(L_h, W_h, S_h, B_h)$ can be bounded by*

$$\log \mathcal{N}(\mathcal{H}, \|\cdot\|_{L^\infty([0,1]^{D_y})}, \varepsilon) \lesssim S_h L_h \log\left(\frac{L_h W_h(B_h \vee 1)}{\varepsilon}\right). \tag{B.17}$$

## B.2  Proofs of Transfer Learning

**Proposition B.4** (Prop. 3.2)**.** *Under Assumption 3.1, 3.2, 3.3, there exists some constant $C_{xy}$ such that the following holds. For any $h \in \mathcal{H}$ and $(x_1, y_1), \cdots, (x_m, y_m) \overset{i.i.d.}{\sim} \mathbb{P}$, define the empirical minimizer*

$$\widehat{f} := \arg\min_{f \in \mathcal{F}} \frac{1}{m} \sum_{i=1}^m \ell(x_i, y_i, s_{f,h}). \tag{B.18}$$

*The population loss of $\widehat{f}$ can be bounded by*

$$\mathbb{E}_{\{(x_i, y_i)\}_{i=1}^m \sim \mathbb{P}} \mathbb{E}_{(x,y) \sim \mathbb{P}}[\ell^{\mathbb{P}}(x, y, s_{\widehat{f}, h})] \leq 4 \inf_{f \in \mathcal{F}} \mathbb{E}_{(x,y) \sim \mathbb{P}}[\ell^{\mathbb{P}}(x, y, s_{f,h})] + C_{xy} \log^3(m) r_x, \tag{B.19}$$

*where* $r_x = \dfrac{M_f^2 S_f L_f \log\left(m L_f W_f (B_f \vee 1) M_f T\right)}{m}$.

*Proof.* Consider the truncated function class defined on $\mathbb{R}^{d_x} \times [0,1]^{D_y}$,

$$\Phi = \{(x,y) \mapsto \widetilde{\ell}(x, y, f) := (\ell(x, y, s_{f,h}) - \ell(x, y, s_*^{\mathbb{P}})) \cdot \mathbb{1}_{\|x\|_\infty \leq R} : f \in \mathcal{F}\}, \tag{B.20}$$

where the truncation radius $R \geq 1$ will be defined later. It is easy to show that with probability no less than $1 - 2m \exp(-C_1' R^2)$, it holds that $\|x_i\|_\infty \leq R$ for all $1 \leq i \leq m$. Hence by definition, the empirical minimizer also satisfies $\widehat{f} = \arg\min_{f \in \mathcal{F}} \dfrac{1}{m} \sum_{i=1}^m \widetilde{\ell}(x_i, y_i, f)$. Below we reason conditioned on this event and verify the conditions required in Lemma B.11.

**Step 1.** To bound the individual loss,

$$\widetilde{\ell}(x, y, f) \leq \mathbb{E}_{t, x_t | x} \|s_{f,h}(x_t, y, t) - \nabla_x \log \phi_t(x_t | x)\|^2 \lesssim M_f^2 + d_x\left(\frac{\log(1/T_0)}{T - T_0} + 1\right). \tag{B.21}$$

And by Lemma B.10,

$$-\widetilde{\ell}(x, y, f) \leq \mathbb{E}_{t, x_t | x} \|s_*^{\mathbb{P}}(x_t, y, t) - \nabla_x \log \phi_t(x_t | x)\|^2 \cdot \mathbb{1}_{\|x\|_\infty \leq R} \lesssim C_X'' R^6 + d_x\left(\frac{\log(1/T_0)}{T} + 1\right). \tag{B.22}$$

Let $M := C\left(C_X'' R^6 + M_f^2 + d_x\left(\frac{\log(1/T_0)}{T} + 1\right)\right)$ and thus $|\widetilde{\ell}(x, y, f)| \leq M$.

**Step 2.** To bound the second order moment, we have

$$\mathbb{E}_{(x,y) \sim \mathbb{P}}\left[\mathbb{1}_{\|x\|_\infty \leq R}\left(\ell(x, y, s_{f,h}) - \ell(x, y, s_*^{\mathbb{P}})\right)^2\right]$$

$$= \mathbb{E}_{(x,y) \sim \mathbb{P}}\left[\mathbb{1}_{\|x\|_\infty \leq R}\left(\mathbb{E}_{t, x_t | x}\|s_{f,h}(x_t, y, t) - \nabla_x \log \phi_t(x_t | x)\|^2 - \|s_*^{\mathbb{P}}(x_t, y, t) - \nabla_x \log \phi_t(x_t | x)\|^2\right)^2\right]$$

$$\leq \mathbb{E}_{(x,y) \sim \mathbb{P}}\left[\mathbb{1}_{\|x\|_\infty \leq R}\left(\mathbb{E}_{t, x_t | x}\|s_{f,h}(x_t, y, t) - s_*^{\mathbb{P}}(x_t, y, t)\|^2\right)\right.$$
$$\left. \cdot \left(\mathbb{E}_{t, x_t | x}\|s_{f,h}(x_t, y, t) + s_*^{\mathbb{P}}(x_t, y, t) - 2\nabla_x \log \phi_t(x_t | x)\|^2\right)\right]$$

$$\leq 4M \mathbb{E}_{(x,y) \sim \mathbb{P}}\left[\mathbb{1}_{\|x\|_\infty \leq R}\left(\mathbb{E}_{t, x_t | x}\|s_{f,h}(x_t, y, t) - s_*^{\mathbb{P}}(x_t, y, t)\|^2\right)\right]$$

$$\leq 4M \mathbb{E}_{(x,y) \sim \mathbb{P}}\left(\ell(x, y, s_{f,h}) - \ell(x, y, s_*^{\mathbb{P}})\right)$$

$$\leq 4M \mathbb{E}_{(x,y) \sim \mathbb{P}}[\widetilde{\ell}(x, y, f)] + 8M^2 \exp(-C_1' R^2). \tag{B.23}$$

**Step 3.** To bound the local Rademacher complexity, note that

$$\left\|\frac{1}{\sqrt{m}}\sum_{i=1}^{m}\sigma_i\widetilde{\ell}(x_i,y_i,f_1)-\frac{1}{\sqrt{m}}\sum_{i=1}^{m}\sigma_i\widetilde{\ell}(x_i,y_i,f_2)\right\|_{\psi_2}\leq 4\|\widetilde{\ell}(\cdot,\cdot,f_1)-\widetilde{\ell}(\cdot,\cdot,f_2)\|_{L^2(\widehat{\mathbb{P}}_m)},$$
(B.24)

where $\widehat{\mathbb{P}}_m:=\frac{1}{m}\sum_{i=1}^{m}\delta_{(x_i,y_i)}$. Define $\Phi_r:=\{\varphi\in\Phi:\frac{1}{m}\sum_{i=1}^{m}\varphi(x_i,y_i)^2\leq r\}$ and it is easy to show that $\mathbf{diam}(\Phi_r,\|\cdot\|_{L^2(\widehat{\mathbb{P}}_m)})\leq 2\sqrt{r}$. By Dudley's bound [Van Handel, 2014, Wainwright, 2019], there exists an absolute constant $C_0$ such that for any $\theta>0$,

$$\mathcal{R}_m(\Phi_r)\leq C_0\left(\theta+\int_\theta^{2\sqrt{r}}\sqrt{\frac{\log\mathcal{N}(\Phi_r,\|\cdot\|_{L^2(\widehat{\mathbb{P}}_m)},\varepsilon)}{m}}\,\mathrm{d}\varepsilon\right).$$
(B.25)

Since $\|x_i\|\leq R$,

$$\frac{1}{m}\sum_{i=1}^{m}(\widetilde{\ell}(x_i,y_i,f_1)-\widetilde{\ell}(x_i,y_i,f_2))^2=\frac{1}{m}\sum_{i=1}^{m}(\ell(x_i,y_i,s_{f_1,h})-\ell(x_i,y_i,s_{f_2,h}))^2$$

$$\leq\frac{1}{m}\sum_{i=1}^{m}\left[\mathbb{E}_{t,x_t|x_i}\|f_1-f_2\|^2\right]\cdot\left[\mathbb{E}_{t,x_t|x_i}\|f_1+f_2-2\nabla_x\log\phi_t\|^2\right]$$

$$\leq\frac{4M}{m}\sum_{i=1}^{m}\mathbb{E}_{t,x_t|x_i}\|f_1(x_t,h(y_i),t)-f_2(x_t,h(y_i),t)\|^2.$$
(B.26)

Let $R_1=2R$. Since $x_t|x_i\sim\mathcal{N}(x_t;\alpha_t x_i,\sigma_t^2 I)$, we have $\mathbb{P}(\|x_t\|_\infty\geq R_1)\leq d_x\mathbb{P}(|\mathcal{N}(0,1)|\leq R)\leq 2d_x\exp(-C_0'R^2)$ for some absolute constant $C_0'$. Therefore,

$$\mathbb{E}_{t,x_t|x_i}\|f_1(x_t,h(y_i),t)-f_2(x_t,h(y_i),t)\|^2$$

$$\leq\mathbb{E}_{t,x_t|x_i}[\mathbb{1}_{\|x_t\|\leq R_1}][\|f_1(x_t,h(y_i),t)-f_2(x_t,h(y_i),t)\|^2]+8d_xM_f^2\exp(-C_0'R^2)$$

$$\leq\|f_1-f_2\|_{L^\infty(\Omega_{R_1})}^2+8d_xM_f^2\exp(-C_0'R^2)$$
(B.27)

where $\Omega_{R_1}:=[-R_1,R_1]^{d_x}\times[0,1]^{d_y}\times[T_0,T]$. Plug in the bound above,

$$\sqrt{\frac{1}{m}\sum_{i=1}^{m}(\widetilde{\ell}(x_i,y_i,f_1)-\widetilde{\ell}(x_i,y_i,f_2))^2}\leq 4M^{\frac{1}{2}}\|f_1-f_2\|_{L^\infty(\Omega_{R_1})}+8d_x^{\frac{1}{2}}M\exp(-C_0'R^2/2).$$
(B.28)

For any $\varepsilon\geq 16d_x^{\frac{1}{2}}M\exp(-C_0'R^2/2)$, according to B.3,

$$\log\mathcal{N}(\Phi_r,\|\cdot\|_{L^2(\widehat{\mathbb{P}}_m)},\varepsilon)\leq\log\mathcal{N}(\mathcal{F},\|\cdot\|_{L^\infty(\Omega_{R_1})},\varepsilon/(8M^{\frac{1}{2}}))$$

$$\leq C_4S_fL_f\log\left(\frac{L_fW_f(B_f\vee 1)(R\vee T)M}{\varepsilon}\right).$$
(B.29)

Plug in (B.25) and let $\theta=16d_x^{\frac{1}{2}}M\exp(-C_0'R^2/2)$,

$$\mathcal{R}_m(\Phi_r)\leq C_0\left(\theta+\int_\theta^{2\sqrt{r}}\sqrt{\frac{C_4S_fL_f\log\left(\frac{L_fW_f(B_f\vee 1)(R\vee T)M}{\varepsilon}\right)}{m}}\mathrm{d}\varepsilon\right)$$

$$\leq C_0\left(16d_x^{\frac{1}{2}}M\exp(-C_0'R^2/2)+\sqrt{\frac{C_4'S_fL_f\log\left(\frac{L_fW_f(B_f\vee 1)(R\vee T)M}{r}\right)\cdot r}{m}}\right)$$

$$=:\widetilde{\mathcal{R}}_m(r)$$
(B.30)

Combine the three steps above, by Lemma B.11 with $B_0 = 8M^2 \exp(-C_1'R^2), B = 4M, b = M$, it holds that with probability no less than $1 - 2m \exp(-C_1'R^2) - \delta/2$, for any $f \in \mathcal{F}$,

$$\mathbb{E}_{(x,y)\sim\mathbb{P}}[\widetilde{\ell}(x,y,f)] \leq \frac{2}{m}\sum_{i=1}^{m}\widetilde{\ell}(x_i,y_i,f) + C_5 M\left(r_m^* + \frac{\log(\log(m)/\delta)}{m}\right)$$
$$+ C_5\sqrt{\frac{M^2\log(\log(m)/\delta)}{m}}\exp(-C_1'R^2), \tag{B.31}$$

$$\frac{1}{m}\sum_{i=1}^{m}\widetilde{\ell}(x_i,y_i,f) \leq 2\mathbb{E}_{(x,y)\sim\mathbb{P}}[\widetilde{\ell}(x,y,f)] + C_5 M\left(r_m^* + \frac{\log(\log(m)/\delta)}{m}\right)$$
$$+ C_5\sqrt{\frac{M^2\log(\log(m)/\delta)}{m}}\exp(-C_1'R^2). \tag{B.32}$$

where $r_m^*$ is the largest fixed point of $\widetilde{\mathcal{R}}_m$, and it can be bounded as

$$r_m^* \leq C_6\left(d_x^{\frac{1}{2}}M\exp(-C_0'R^2/2) + \frac{S_f L_f \log\left(mL_fW_f(B_f\vee 1)(R\vee T)M\right)}{m}\right), \tag{B.33}$$

for some absolute constant $C_6$. Moreover, we have

$$\left|\mathbb{E}_{(x,y)\sim\mathbb{P}}[\ell(x,y,s_{f,h}) - \ell(x,y,s_*^{\mathbb{P}})] - \mathbb{E}_{(x,y)\sim\mathbb{P}}[\widetilde{\ell}(x,y,f)]\right| \leq 2M\exp(-C_1'R^2). \tag{B.34}$$

Combine this with (B.31),(B.32),

$$\mathbb{E}_{(x,y)\sim\mathbb{P}}[\ell(x,y,s_{f,h}) - \ell(x,y,s_*^{\mathbb{P}})] \leq \frac{2}{m}\sum_{i=1}^{m}[\ell(x_i,y_i,s_{f,h}) - \ell(x_i,y_i,s_*^{\mathbb{P}})]$$
$$+ C_5 M\left(r_m^* + \frac{\log(\log(m)/\delta)}{m} + \exp(-C_1'R^2)\right), \tag{B.35}$$

$$\frac{1}{m}\sum_{i=1}^{m}[\ell(x_i,y_i,s_{f,h}) - \ell(x_i,y_i,s_*^{\mathbb{P}})] \leq 2\mathbb{E}_{(x,y)\sim\mathbb{P}}[\ell(x,y,s_{f,h}) - \ell(x,y,s_*^{\mathbb{P}})]$$
$$+ C_5 M\left(r_m^* + \frac{\log(\log(m)/\delta)}{m} + \exp(-C_1'R^2)\right), \tag{B.36}$$

Plug in the definition of $M = C\left(C_X''R^6 + M_f^2 + d_x\left(\frac{\log(1/T_0)}{T} + 1\right)\right)$ and let $R = C\log^{\frac{1}{2}}(md_xM_f/\delta)$ for some large constant $C$. Hence (B.35) and (B.36) reduce to

$$\mathbb{E}_{(x,y)\sim\mathbb{P}}[\ell^{\mathbb{P}}(x,y,s_{f,h})] \leq \frac{2}{m}\sum_{i=1}^{m}[\ell(x_i,y_i,s_{f,h}) - \ell(x_i,y_i,s_*^{\mathbb{P}})] + C_7 M_f^2\log^3(m/\delta)\left(r_m^\dagger + \frac{\log(\log(m)/\delta)}{m}\right), \tag{B.37}$$

$$\frac{1}{m}\sum_{i=1}^{m}[\ell(x_i,y_i,s_{f,h}) - \ell(x_i,y_i,s_*^{\mathbb{P}})] \leq 2\mathbb{E}_{(x,y)\sim\mathbb{P}}[\ell^{\mathbb{P}}(x,y,s_{f,h})] + C_7 M_f^2\log^3(m/\delta)\left(r_m^\dagger + \frac{\log(\log(m)/\delta)}{m}\right), \tag{B.38}$$

where $r_m^\dagger := \frac{S_f L_f \log\left(mL_fW_f(B_f\vee 1)TM_f\log(1/\delta)\right)}{m}$.

Therefore, we obtain that with probability no less than $1 - \delta$, the population loss of the empirical minimizer $\widehat{f}$ can be bounded by

$$\mathbb{E}_{(x,y)\sim\mathbb{P}}[\ell^\mathbb{P}(x,y,s_{\widehat{f},h})] \leq \frac{2}{m}\sum_{i=1}^{m}[\ell(x_i,y_i,s_{\widehat{f},h}) - \ell(x_i,y_i,s_*^\mathbb{P})] + 2C_7 M_f^2 \log^3(m/\delta)\left(r_m^\dagger + \frac{\log(1/\delta)}{m}\right)$$

$$\leq \inf_{f\in\mathcal{F}} \frac{2}{m}\sum_{i=1}^{m}[\ell(x_i,y_i,s_{f,h}) - \ell(x_i,y_i,s_*^\mathbb{P})] + 2C_7 M_f^2 \log^3(m/\delta)\left(r_m^\dagger + \frac{\log(1/\delta)}{m}\right)$$

$$\leq 4\inf_{f\in\mathcal{F}}\mathbb{E}_{(x,y)\sim\mathbb{P}}[\ell^\mathbb{P}(x,y,s_{f,h})] + 6C_7 M_f^2 \log^3(m/\delta)\left(r_m^\dagger + \frac{\log(1/\delta)}{m}\right),$$
$$(\text{B.39})$$

We conclude the proof by noticing that $\mathbb{E}[X] = \int_0^\infty \mathbb{P}(X \geq x)\mathrm{d}x$ and plugging in the bound above. $\qquad\square$

**Proposition B.5** (Prop. 3.3)**.** *There exists some constant $C_Z, C_R$ such that the following holds. For any $\mathbb{P}^1, \cdots, \mathbb{P}^K$, let $x_1^k, \cdots, x_n^k \overset{i.i.d.}{\sim} \mathbb{P}^k$ for any $k$ and $(x_i^k)_{i,k}$ are all independent. Consider the empirical minimizer*

$$\widehat{\boldsymbol{f}},\widehat{h} = \operatorname*{arg\,min}_{\boldsymbol{f}\in\mathcal{F}^{\otimes K}, h\in\mathcal{H}} \frac{1}{nK}\sum_{k=1}^{K}\sum_{i=1}^{n}\ell(x_i^k, y_i^k, s_{f^k,h}). \qquad (\text{B.40})$$

*For any $\delta \in (0,1)$, if the configuration of $\mathcal{F}$ satisfies $R_f \geq C_R \log^{\frac{1}{2}}(nKM_f/\delta)$, then with probability no less than $1 - \delta$, the population loss of $\widehat{\boldsymbol{f}}, \widehat{h}$ can be bounded by*

$$\frac{1}{K}\sum_{k=1}^{K}\mathbb{E}_{(x,y)\sim\mathbb{P}^k}\ell^{\mathbb{P}^k}(x,y,s_{\widehat{f}^k,\widehat{h}}) \leq \inf_{\boldsymbol{f}\in\mathcal{F}^{\otimes K}, h\in\mathcal{H}}\frac{4}{K}\sum_{k=1}^{K}\mathbb{E}_{(x,y)\sim\mathbb{P}^k}[\ell^\mathbb{P}(x,y,s_{f^k,h})] + C_Z\log^3(nK/\delta)\left(r_z + \frac{\log(1/\delta)}{nK}\right),$$
$$(\text{B.41})$$

*where $r_z := \dfrac{M_f^2\left[KS_fL_f\log\left(nL_fW_f(B_f\vee 1)M_fT\log(1/\delta)\right) + S_hL_h\log\left(nKL_hW_h(B_h\vee 1)M_f\gamma_f\log(1/\delta)\right)\right]}{nK}.$*

*Proof.* Throughout the proof, we will use $z = (k,x,y)$ to denote the tuple of task index $k$ and data $(x,y)$. With a little abuse of notation, we will also let $s_*^k = s_*^{\mathbb{P}^k}$. Consider the function class defined on $[K] \times \mathbb{R}^{d_x} \times [0,1]^{D_y}$,

$$\Phi = \left\{z = (k,x,y) \mapsto \widetilde{\ell}(z,\boldsymbol{f},h) := (\ell(x,y,s_{f^k,h}) - \ell(x,y,s_*^k)) \cdot \mathbb{1}_{\|x\|_\infty \leq R} : \boldsymbol{f}\in\mathcal{F}^{\otimes K}, h\in\mathcal{H}\right\},$$
$$(\text{B.42})$$

where $1 \leq R \leq \dfrac{R_f}{2}$ will be specified later. It is easy to show that with probability no less than $1 - 2nK\exp(-C_1'R^2)$, it holds that $\|x_i^k\|_\infty \leq R$ for all $i,k$. Hence by definition, the empirical minimizer also satisfies

$$\widehat{\boldsymbol{f}},\widehat{h} = \operatorname*{arg\,min}_{\boldsymbol{f}\in\mathcal{F}^{\otimes K}, h\in\mathcal{H}} \frac{1}{nK}\sum_{k=1}^{K}\sum_{i=1}^{n}\widetilde{\ell}(z_i^k, \boldsymbol{f}, h). \qquad (\text{B.43})$$

where $z_i^k = (k, x_i^k, y_i^k)$. Below we reason conditioned on this event and verify the conditions in Lemma B.12.

Following Step 1 and 2 in Proposition B.4, we have for any $\boldsymbol{f}\in\mathcal{F}^{\otimes K}, h\in\mathcal{H}$,

$$|\widetilde{\ell}(z,\boldsymbol{f},h)| \leq M := C\left(C_X''R^6 + M_f^2 + d_x\Big(\frac{\log(1/T_0)}{T} + 1\Big)\right). \qquad (\text{B.44})$$

$$\frac{1}{K}\sum_{k=1}^{K}\mathbb{E}_{(x,y)\sim\mathbb{P}^k}[\widetilde{\ell}(z^k,\boldsymbol{f},h)^2] \leq \frac{4M}{K}\sum_{k=1}^{K}\mathbb{E}_{(x,y)\sim\mathbb{P}^k}[\widetilde{\ell}(z^k,\boldsymbol{f},h)] + 8M^2\exp(-C_1'R^2). \quad (\text{B.45})$$

For the local Rademacher complexity bound, note that

$$\left\| \frac{1}{\sqrt{nK}} \sum_{k=1}^{K} \sum_{i=1}^{n} \sigma_i^k \widetilde{\ell}(z_i^k, \boldsymbol{f}_1, h_1) - \frac{1}{\sqrt{nK}} \sum_{k=1}^{K} \sum_{i=1}^{n} \sigma_i^k \widetilde{\ell}(z_i^k, \boldsymbol{f}_2, h_2) \right\|_{\psi_2} \leq 4 \| \widetilde{\ell}(\cdot, \boldsymbol{f}_1, h_1) - \widetilde{\ell}(\cdot, \boldsymbol{f}_2, h_2) \|_{L^2(\widehat{\mathbb{P}}_n^{(K)})},$$
(B.46)

where $\widehat{\mathbb{P}}_n^{(K)} := \frac{1}{nK} \sum_{k=1}^{K} \sum_{i=1}^{n} \delta_{z_i^k}$ and $\mathbf{diam}\big(\Phi_r, \| \cdot \|_{L^2(\widehat{\mathbb{P}}_n^{(K)})}\big) \leq 2\sqrt{r}$. By Dudley's bound [Van Handel, 2014, Wainwright, 2019], there exists an absolute constant $C_0$ such that for any $\theta > 0$,

$$\mathcal{R}_{K,n}(\Phi_r) \leq C_0 \left( \theta + \int_{\theta}^{2\sqrt{r}} \sqrt{\frac{\log \mathcal{N}(\Phi_r, \| \cdot \|_{L^2(\widehat{\mathbb{P}}_n^{(K)})}, \varepsilon)}{nK}} \, d\varepsilon \right).$$
(B.47)

Since $\|x_i^k\|_\infty \leq R$,

$$\frac{1}{nK} \sum_{k=1}^{K} \sum_{i=1}^{n} (\widetilde{\ell}(z_i^k, \boldsymbol{f}_1, h_1) - \widetilde{\ell}(z_i^k, \boldsymbol{f}_2, h_2))^2$$

$$= \frac{1}{nK} \sum_{k=1}^{K} \sum_{i=1}^{n} (\ell(x_i^k, y_i^k, s_{f_1^k, h_1}) - \ell(x_i^k, y_i^k, s_{f_2^k, h_2}))^2$$

$$\leq \frac{1}{nK} \sum_{k=1}^{K} \sum_{i=1}^{n} \left[ \mathbb{E}_{t, x_t | x_i^k} \| f_1^k - f_2^k \|^2 \right] \cdot \left[ \mathbb{E}_{t, x_t | x_i^k} \| f_1^k + f_2^k - 2\nabla_x \log \phi_t \|^2 \right]$$
(B.48)

$$\leq \frac{4M}{nK} \sum_{k=1}^{K} \sum_{i=1}^{n} \mathbb{E}_{t, x_t | x_i^k} \| f_1^k(x_t, h_1(y_i^k), t) - f_2^k(x_t, h_2(y_i^k), t) \|^2$$

$$\leq \frac{8M}{nK} \sum_{k=1}^{K} \sum_{i=1}^{n} \mathbb{E}_{t, x_t | x_i^k} \| f_1^k(x_t, h_1(y_i^k), t) - f_2^k(x_t, h_1(y_i^k), t) \|^2$$

$$+ \frac{8M}{nK} \sum_{k=1}^{K} \sum_{i=1}^{n} \mathbb{E}_{t, x_t | x_i^k} \| f_2^k(x_t, h_1(y_i^k), t) - f_2^k(x_t, h_2(y_i^k), t) \|^2.$$

Let $R_1 = 2R$. Since $x_t | x_i^k \sim \mathcal{N}(x_t; \alpha_t x_i^k, \sigma_t^2 I)$, we have $\mathbb{P}(\|x_t\|_\infty \geq R_1) \leq d_x \mathbb{P}(|\mathcal{N}(0,1)| \leq R) \leq 2d_x \exp(-C_0' R^2)$ for some absolute constant $C_0'$. Therefore,

$$\mathbb{E}_{t, x_t | x_i^k} \| f_1^k(x_t, h_1(y_i^k), t) - f_2^k(x_t, h_1(y_i^k), t) \|^2$$

$$\leq \mathbb{E}_{t, x_t | x_i^k} [\mathbb{1}_{\|x_t\| \leq R_1}] [\| f_1^k(x_t, h_1(y_i^k), t) - f_2^k(x_t, h_1(y_i^k), t) \|^2] + 8d_x M_f^2 \exp(-C_0' R^2)$$

$$\leq \| f_1^k - f_2^k \|_{L^\infty(\Omega_{R_1})}^2 + 8d_x M_f^2 \exp(-C_0' R^2),$$
(B.49)

where $\Omega_{R_1} := [-R_1, R_1]^{d_x} \times [0,1]^{d_y} \times [T_0, T]$. Moreover, notice that $R_f \geq 2R = R_1$,

$$\mathbb{E}_{t, x_t | x_i^k} \| f_2^k(x_t, h_1(y_i^k), t) - f_2^k(x_t, h_2(y_i^k), t) \|^2$$

$$\leq \mathbb{E}_{t, x_t | x_i^k} [\mathbb{1}_{\|x_t\| \leq R_f}] [\| f_2^k(x_t, h_1(y_i^k), t) - f_2^k(x_t, h_2(y_i^k), t) \|^2] + 8d_x M_f^2 \exp(-C_0' R^2)$$

$$\leq \gamma_f^2 \| h_1 - h_2 \|_{L^\infty([0,1]^{D_y})}^2 + 8d_x M_f^2 \exp(-C_0' R^2).$$
(B.50)

Plug in the bound above,

$$\sqrt{\frac{1}{nK} \sum_{k=1}^{K} \sum_{i=1}^{n} (\widetilde{\ell}(z_i^k, \boldsymbol{f}_1, h_1) - \widetilde{\ell}(z_i^k, \boldsymbol{f}_2, h_2))^2}$$

$$\leq 8M^{\frac{1}{2}} \left( \max_k \| f_1^k - f_2^k \|_{L^\infty(\Omega_{R_1})} + \gamma_f \| h_1 - h_2 \|_{L^\infty([0,1]^{D_y})} \right) + 16d_x^{\frac{1}{2}} M \exp(-C_0' R^2 / 2).$$
(B.51)

For any $\varepsilon \geq 32 d_x^{\frac{1}{2}} M \exp(-C_0' R^2/2)$, according to Lemma B.3,

$$
\begin{aligned}
\log \mathcal{N}&(\Phi_r, \|\cdot\|_{L^2(\widehat{\mathbb{P}}_n^{(K)})}, \varepsilon) \\
&\leq K \log \mathcal{N}(\mathcal{F}, \|\cdot\|_{L^\infty(\Omega_{R_1})}, \varepsilon/(16M^{\frac{1}{2}})) + \log \mathcal{N}(\mathcal{H}, \|\cdot\|_{L^\infty([0,1]^{D_y})}, \varepsilon/(16\gamma_f M^{\frac{1}{2}})) \\
&\leq C_4 K S_f L_f \log\left(\frac{L_f W_f (B_f \vee 1)(R \vee T)M}{\varepsilon}\right) + C_4 S_h L_h \log\left(\frac{L_h W_h (B_h \vee 1)M\gamma_f}{\varepsilon}\right).
\end{aligned}
\tag{B.52}
$$

Plug in (B.25) and let $\theta = 32 d_x^{\frac{1}{2}} M \exp(-C_0' R^2/2)$,

$$
\begin{aligned}
\mathcal{R}_{K,n}(\Phi_r) &\leq C_0 \left( \theta + \int_\theta^{2\sqrt{r}} \sqrt{\frac{C_4 K S_f L_f \log\left(\frac{L_f W_f (B_f \vee 1)(R \vee T)M}{\varepsilon}\right) + C_4 S_h L_h \log\left(\frac{L_h W_h (B_h \vee 1)M\gamma_f}{\varepsilon}\right)}{nK}} \, d\varepsilon \right) \\
&\leq C_0 \sqrt{\frac{C_4'\left[ K S_f L_f \log\left(\frac{L_f W_f (B_f \vee 1)(R \vee T)M}{r}\right) + S_h L_h \log\left(\frac{L_h W_h (B_h \vee 1)M\gamma_f}{r}\right)\right] \cdot r}{nK}} \\
&\quad + C_0 32 d_x^{\frac{1}{2}} M \exp(-C_0' R^2/2) \\
&=: \widetilde{\mathcal{R}}_{K,n}(r).
\end{aligned}
\tag{B.53}
$$

Combine the arguments above, by Lemma B.12 with $B_0 = 8M^2 \exp(-C_1' R^2)$, $B = 4M$, $b = M$, it holds that with probability no less than $1 - 2nK \exp(-C_1' R^2) - \delta/2$, for any $\boldsymbol{f} \in \mathcal{F}^{\otimes K}$, $h \in \mathcal{H}$,

$$
\begin{aligned}
\mathbb{E}_{z \sim \widehat{\mathbb{P}}^{(K)}}[\widetilde{\ell}(z, \boldsymbol{f}, h)] &\leq \frac{2}{nK} \sum_{k=1}^K \sum_{i=1}^n \widetilde{\ell}(z_i^k, \boldsymbol{f}, h) + C_5 M \left( r_{K,n}^* + \frac{\log(\log(nK)/\delta)}{nK} \right) \\
&\quad + C_5 \sqrt{\frac{M^2 \log(\log(nK)/\delta)}{nK}} \exp(-C_1' R^2),
\end{aligned}
\tag{B.54}
$$

$$
\begin{aligned}
\frac{1}{nK} \sum_{k=1}^K \sum_{i=1}^n \widetilde{\ell}(z_i^k, \boldsymbol{f}, h) &\leq 2\mathbb{E}_{z \sim \widehat{\mathbb{P}}^{(K)}}[\widetilde{\ell}(z, \boldsymbol{f}, h)] + C_5 M \left( r_{K,n}^* + \frac{\log(\log(nK)/\delta)}{nK} \right) \\
&\quad + C_5 \sqrt{\frac{M^2 \log(\log(nK)/\delta)}{nK}} \exp(-C_1' R^2).
\end{aligned}
\tag{B.55}
$$

where $r_{K,n}^*$ is the largest fixed point of $\widetilde{\mathcal{R}}_{K,n}$, and it can be bounded by

$$
r_{K,n}^* \leq C_6 \left( d_x^{\frac{1}{2}} M_f \exp(-C_0' R^2/2) + \frac{K S_f L_f \log(n L_f W_f (B_f \vee 1)(R \vee T)M) + S_h L_h \log(nK L_h W_h (B_h \vee 1)M\gamma_f)}{nK} \right),
\tag{B.56}
$$

for some absolute constant $C_6$. Moreover, we have

$$
\left| \frac{1}{K} \sum_{k=1}^K \mathbb{E}_{(x,y) \sim \mathbb{P}^k}[\ell(x, y, s_{f^k, h}) - \ell(x, y, s_*^k)] - \mathbb{E}_{z \sim \widehat{\mathbb{P}}^{(K)}}[\widetilde{\ell}(z, \boldsymbol{f}, h)] \right| \leq 2M \exp(-C_1' R^2).
\tag{B.57}
$$

Combine this with (B.54),(B.55),

$$
\begin{aligned}
\frac{1}{K} \sum_{k=1}^K \mathbb{E}_{(x,y) \sim \mathbb{P}^k}[\ell(x, y, s_{f^k, h}) - \ell(x, y, s_*^{\mathbb{P}})] &\leq \frac{2}{nK} \sum_{k=1}^K \sum_{i=1}^n [\ell(x_i^k, y_i^k, s_{f^k, h}) - \ell(x_i^k, y_i^k, s_*^k)] \\
&\quad + C_5 M \left( r_{K,n}^* + \frac{\log(\log(nK)/\delta)}{nK} + \exp(-C_1' R^2) \right),
\end{aligned}
\tag{B.58}
$$

$$\frac{1}{nK}\sum_{k=1}^{K}\sum_{i=1}^{n}[\ell(x_i^k,y_i^k,s_{f^k,h})-\ell(x_i^k,y_i^k,s_*^k)] \leq \frac{2}{K}\sum_{k=1}^{K}\mathbb{E}_{(x,y)\sim\mathbb{P}^k}[\ell(x,y,s_{f^k,h})-\ell(x,y,s_*^{\mathbb{P}})]$$
$$+C_5 M\left(r_{K,n}^* + \frac{\log(\log(nK)/\delta)}{nK}+\exp(-C_1'R^2)\right),$$
$$\text{(B.59)}$$

Plug in the definition of $M = C\left(C_X''R^6 + M_f^2 + d_x\left(\frac{\log(1/T_0)}{T}+1\right)\right)$ and define $R = C'\log^{\frac{1}{2}}(nKd_xM_f/\delta)$ for some large constant $C'$. Hence (B.58) and (B.59) reduce to

$$\frac{1}{K}\sum_{k=1}^{K}\mathbb{E}_{(x,y)\sim\mathbb{P}^k}[\ell(x,y,s_{f^k,h})-\ell(x,y,s_*^{\mathbb{P}})] \leq \frac{2}{nK}\sum_{k=1}^{K}\sum_{i=1}^{n}[\ell(x_i^k,y_i^k,s_{f^k,h})-\ell(x_i^k,y_i^k,s_*^k)]$$
$$+C_7 M_f^2 \log^3(nK/\delta)\left(r_{K,n}^{\dagger}+\frac{\log(\log(nK)/\delta)}{nK}\right),$$
$$\text{(B.60)}$$

$$\frac{1}{nK}\sum_{k=1}^{K}\sum_{i=1}^{n}[\ell(x_i^k,y_i^k,s_{f^k,h})-\ell(x_i^k,y_i^k,s_*^k)] \leq \frac{2}{K}\sum_{k=1}^{K}\mathbb{E}_{(x,y)\sim\mathbb{P}^k}[\ell(x,y,s_{f^k,h})-\ell(x,y,s_*^{\mathbb{P}})]$$
$$+C_7 M_f^2 \log^3(nK/\delta)\left(r_{K,n}^{\dagger}+\frac{\log(\log(nK)/\delta)}{nK}\right),$$
$$\text{(B.61)}$$

where $r_{K,n}^{\dagger} := \dfrac{KS_fL_f\log\left(nL_fW_f(B_f\vee 1)M_fT\log(1/\delta)\right)+S_hL_h\log\left(nKL_hW_h(B_h\vee 1)M_f\gamma_f\log(1/\delta)\right)}{nK}.$

Therefore, we obtain that with probability no less than $1-\delta$, the population loss of the empirical minimizer $\widehat{f},\widehat{h}$ can be bounded by

$$\frac{1}{K}\sum_{k=1}^{K}\mathbb{E}_{(x,y)\sim\mathbb{P}^k}[\ell^{\mathbb{P}^k}(x,y,s_{\widehat{f}^k,\widehat{h}})]$$
$$\leq \frac{2}{nK}\sum_{k=1}^{K}\sum_{i=1}^{m}[\ell(x_i^k,y_i^k,s_{\widehat{f},h})-\ell(x_i^k,y_i^k,s_*^k)]+2C_7M_f^2\log^3(nK/\delta)\left(r_{K,n}^{\dagger}+\frac{\log(1/\delta)}{nK}\right)$$
$$\leq \inf_{\boldsymbol{f}\in\mathcal{F}^{\otimes K},h\in\mathcal{H}}\frac{2}{nK}\sum_{k=1}^{K}\sum_{i=1}^{n}[\ell(x_i^k,y_i^k,s_{f^k,h})-\ell(x_i^k,y_i^k,s_*^k)]+2C_7M_f^2\log^3(nK/\delta)\left(r_{K,n}^{\dagger}+\frac{\log(1/\delta)}{nK}\right)$$
$$\leq \inf_{\boldsymbol{f}\in\mathcal{F}^{\otimes K},h\in\mathcal{H}}\frac{4}{K}\sum_{k=1}^{K}\mathbb{E}_{(x,y)\sim\mathbb{P}^k}[\ell^{\mathbb{P}}(x,y,s_{f^k,h})]+6C_7M_f^2\log^3(nK/\delta)\left(r_{K,n}^{\dagger}+\frac{\log(1/\delta)}{nK}\right),$$
$$\text{(B.62)}$$

which concludes the proof. $\qquad\square$

**Theorem B.6** (Thm. 3.4). *Under Assumption 3.1, 3.2, 3.3, suppose $\mathbb{P}^1,\cdots,\mathbb{P}^K$ are $(\nu,\Delta)$-diverse over target distribution $\mathbb{P}^0$ given $\mathcal{F},\mathcal{H}$. There exists some constant $C,C_R$ such that the following holds. Define the empirical minimizer of training task and new task as*

$$\widehat{\boldsymbol{f}},\widehat{h} = \operatorname*{arg\,min}_{\boldsymbol{f}\in\mathcal{F}^{\otimes K},h\in\mathcal{H}}\frac{1}{nK}\sum_{k=1}^{K}\sum_{i=1}^{n}\ell(x_i^k,y_i^k,s_{f^k,h}),\tag{B.63}$$

$$\widehat{f}^{\mathbb{P}^0} := \operatorname*{arg\,min}_{f\in\mathcal{F}}\frac{1}{m}\sum_{i=1}^{m}\ell(x_i^0,y_i^0,s_{f,\widehat{h}}).\tag{B.64}$$

*If $R_f \geq C_R \log^{\frac{1}{2}}(nKM_f/\delta)$, then with probability no less than $1 - \delta$, the expected population loss of new task can be bounded by*

$$\mathbb{E}_{\{(x_i,y_i)\}_{i=1}^m}\mathbb{E}_{(x,y)\sim\mathbb{P}^0}[\ell^{\mathbb{P}^0}(x,y,s_{\widehat{f}^{\mathbb{P}^0},\widehat{h}})] \lesssim \frac{1}{\nu}\inf_{h\in\mathcal{H}}\frac{1}{K}\sum_{k=1}^K\inf_{f\in\mathcal{F}}\mathbb{E}_{(x,y)\sim\mathbb{P}^k}[\ell^{\mathbb{P}^k}(x,y,s_{f,h})] + \Delta$$
$$+ C\left(\frac{\log^3(m)\log\mathcal{N}_\mathcal{F}}{m} + \frac{\log^3(nK/\delta)(K\log\mathcal{N}_\mathcal{F} + \log(\mathcal{N}_\mathcal{H}/\delta))}{\nu nK}\right).$$
$$\text{(B.65)}$$

*where*

$$\log\mathcal{N}_\mathcal{F} := M_f^2 S_f L_f \log\left(mnL_f W_f(B_f \vee 1)M_f T\log(1/\delta)\right), \tag{B.66}$$

$$\log\mathcal{N}_\mathcal{H} := S_h L_h \log\left(nKL_h W_h(B_h \vee 1)M_f\gamma_f\log(1/\delta)\right). \tag{B.67}$$

*Proof.*

$$\mathbb{E}_{\{(x_i,y_i)\}_{i=1}^m}\mathbb{E}_{(x,y)\sim\mathbb{P}^0}[\ell^{\mathbb{P}^0}(x,y,s_{\widehat{f}^{\mathbb{P}^0},\widehat{h}})]$$
$$\lesssim \inf_{f\in\mathcal{F}}\mathbb{E}_{(x,y)\sim\mathbb{P}^0}[\ell^{\mathbb{P}}(x,y,s_{f,\widehat{h}})] + C_{xy}\log^3(m)r_x$$
$$\lesssim \frac{1}{\nu K}\sum_{k=1}^K\inf_{f\in\mathcal{F}}\mathbb{E}_{(x,y)\sim\mathbb{P}^k}[\ell^{\mathbb{P}^k}(x,y,s_{f,\widehat{h}})] + \Delta + C_{xy}\log^3(m)r_x$$
$$\lesssim \frac{1}{\nu K}\sum_{k=1}^K\mathbb{E}_{(x,y)\sim\mathbb{P}^k}[\ell^{\mathbb{P}^k}(x,y,s_{\widehat{f}^k,\widehat{h}})] + \Delta + C_{xy}\log^3(m)r_x$$
$$\lesssim \frac{1}{\nu}\inf_{\mathbf{f}\in\mathcal{F}^{\otimes K},h\in\mathcal{H}}\frac{1}{K}\sum_{k=1}^K\mathbb{E}_{(x,y)\sim\mathbb{P}^k}[\ell^{\mathbb{P}^k}(x,y,s_{f^k,h})] + \frac{1}{\nu}C_Z\log^3(nK/\delta)\left(r_z + \frac{\log(1/\delta)}{nK}\right)$$
$$+ \Delta + C_{xy}\log^3(m)r_x.$$
$$\text{(B.68)}$$

Here we apply Proposition B.4 in the first inequality, task diversity in the second inequality, and Proposition B.5 in the fourth. Plug in the definition of $r_z, r_x$ and $\log\mathcal{N}_\mathcal{F}, \log\mathcal{N}_\mathcal{H}$ and we complete the proof. $\qquad\square$

## B.3 Proofs of Meta-Learning

**Proposition B.7** (Prop. 3.5). *There exists some constants $C_1', C_P$, such that for $\mathbb{P}^1,\cdots,\mathbb{P}^K \overset{i.i.d.}{\sim} \mathbb{P}_{meta}$, with probability no less than $1 - \delta$, we have for any $h \in \mathcal{H}$,*

$$\mathbb{E}_{\mathbb{P}\sim\mathbb{P}_{meta}}\mathcal{L}(\mathbb{P},h) \leq \frac{2}{K}\sum_{k=1}^K\mathcal{L}(\mathbb{P}^k,h) + C_P\left(r_P + \frac{\log(1/\delta)}{K}\right), \tag{B.69}$$

$$\frac{1}{K}\sum_{k=1}^K\mathcal{L}(\mathbb{P}^k,h) \leq 2\mathbb{E}_{\mathbb{P}\sim\mathbb{P}_{meta}}\mathcal{L}(\mathbb{P},h) + C_P\left(r_P + \frac{\log(1/\delta)}{K}\right), \tag{B.70}$$

*where $r_P = M_f^2\exp(-C_1'R_f^2) + \dfrac{S_h L_h\log\left(KL_h W_h(B_h \vee 1)M_f\gamma_f\right)}{K}$.*

*Proof.* Given $\mathbb{P}^1,\cdots,\mathbb{P}^K \overset{i.i.d.}{\sim} \mathbb{P}_{\text{meta}}$, we define the empirical Rademacher complexity of a function class $\Phi$ defined on the set of distribution $\mathcal{P}(\mathbb{R}^{d_x} \times [0,1]^{D_y})$ as

$$\mathcal{R}_K(\Phi) := \mathbb{E}_{\boldsymbol{\sigma}}\sup_{\varphi\in\Phi}\left|\frac{1}{K}\sum_{k=1}^K\sigma_k\varphi(\mathbb{P}^k)\right|, \quad \boldsymbol{\sigma}\sim\text{Unif}(\{-1,1\}^K). \tag{B.71}$$

For any $r > 0$, let $\mathcal{H}_r := \left\{ h \in \mathcal{H} : \frac{1}{K} \sum_{k=1}^{K} (\mathcal{L}(\mathbb{P}^k, h))^2 \leq r \right\}$ and $\Phi_r := \{\mathcal{L}(\cdot, h) : h \in \mathcal{H}_r\}$. Note that for any $\varphi_1, \varphi_2 \in \Phi_r$,

$$
\left\| \frac{1}{\sqrt{K}} \sum_{k=1}^{K} \sigma_k \varphi_1(\mathbb{P}^k) - \frac{1}{\sqrt{K}} \sum_{k=1}^{K} \sigma_k \varphi_2(\mathbb{P}^k) \right\|_{\psi_2} \leq 4\sqrt{\frac{1}{K} \sum_{k=1}^{K} \|\varphi_1(\mathbb{P}^k) - \varphi_2(\mathbb{P}^k)\|^2}
$$
$$
= 4\|\varphi_1 - \varphi_2\|_{L^2(\mathbb{P}_{\mathrm{meta}}^{(K)})}, \tag{B.72}
$$

where $\mathbb{P}_{\mathrm{meta}}^{(K)} := \frac{1}{K} \sum_{k=1}^{K} \delta_{\mathbb{P}^k}$ and $\mathbf{diam}\left(\Phi_r, \|\cdot\|_{L^2(\mathbb{P}_{\mathrm{meta}}^{(K)})}\right) \leq 2\sqrt{r}$. Then by Dudley's bound [Van Handel, 2014, Wainwright, 2019], there exists an absolute constant $C_0$ such that for any $\theta \geq 0$,

$$
\mathcal{R}_K(\Phi_r) \leq C_0 \left( \theta + \int_\theta^{2\sqrt{r}} \sqrt{\frac{\log \mathcal{N}(\Phi_r, \|\cdot\|_{L^2(\mathbb{P}_{\mathrm{meta}}^{(K)})}, \varepsilon)}{K}} \, d\varepsilon \right). \tag{B.73}
$$

For any $\mathbb{P}$ and $h_1, h_2 \in \mathcal{H}_r$, denote the minimizer of (3.6) in $\mathcal{F}$ as $f_1, f_2$, respectively. Without loss of generality, suppose $\mathcal{L}(\mathbb{P}, h_1) \geq \mathcal{L}(P, h_2)$. Then

$$
\mathcal{L}(\mathbb{P}, h_1) - \mathcal{L}(P, h_2) \leq \mathbb{E}_{t,x_t,y} \left[ \left| \|f_2(x_t, h_1(y), t) - \nabla_x \log p_t(x_t|y)\|^2 - \|f_2(x_t, h_2(y), t) - \nabla_x \log p_t(x_t|y)\|^2 \right| \right]
$$
$$
\leq \mathbb{E}_{t,x_t,y} \left[ \|f_2(x_t, h_1(y), t) - f_2(x_t, h_2(y), t)\| \right.
$$
$$
\left. \times \|f_2(x_t, h_1(y), t) + f_2(x_t, h_2(y), t) - 2\nabla_x \log p_t(x_t|y)\| \right]
$$
$$
\leq \sqrt{\mathbb{E}_{t,x_t,y} \left[ \|f_2(x_t, h_1(y), t) - f_2(x_t, h_2(y), t)\|^2 \right]} \cdot 8(M_f + C_L^{1/2})
$$
$$
\tag{B.74}
$$

In the last inequality we apply $\|f_i\| \leq M_f$ and $\mathbb{E}_{t,x_t,y} \|\nabla_x \log p_t(x_t|y)\|^2 \leq C_L$ by Lemma B.9. Moreover,

$$
\mathbb{E}_{(t,x_t,y)} \left[ \|f_2(x_t, h_1(y), t) - f_2(x_t, h_2(y), t)\|^2 \right]
$$
$$
\leq \mathbb{E}_{t,y} \left[ \int \|f_2(x_t, h_1(y), t) - f_2(x_t, h_2(y), t)\|^2 p_t(x_t|y) dx_t \right]
$$
$$
\leq \mathbb{E}_{t,y} \left[ \int_{\|x_t\|_\infty \leq R_f} \|f_2(x_t, h_1(y), t) - f_2(x_t, h_2(y), t)\|^2 p_t(x_t|y) dx_t + 4M_f^2 \mathbb{P}(\|x_t\|_\infty > R_f|y) \right]
$$
$$
\leq \gamma_f^2 \mathbb{E}_y[\|h_1(y) - h_2(y)\|^2] + 8M_f^2 \exp(-C_1' R_f^2)
$$
$$
\leq \gamma_f^2 \|h_1 - h_2\|_{L^\infty([0,1]^{D_y})}^2 + 8M_f^2 \exp(-C_1' R_f^2)
$$
$$
\tag{B.75}
$$

Therefore, let $C_3 = 32(M_f + C_L^{1/2})M_f \leq 64M_f^2$ and we have

$$
|\mathcal{L}(\mathbb{P}, h_1) - \mathcal{L}(P, h_2)| \leq C_3 \left( \gamma_f \|h_1 - h_2\|_{L^\infty([0,1]^{D_y})} + \exp(-C_1' R_f^2) \right), \tag{B.76}
$$

which implies that when $\varepsilon \geq 2C_3 \exp(-C_1' R_f^2)$, by Lemma B.3,

$$
\log \mathcal{N}(\Phi_r, \|\cdot\|_{L^2(\mathbb{P}_{\mathrm{meta}}^{(K)})}, \varepsilon) \leq \log \mathcal{N}(\mathcal{H}_r, \|\cdot\|_{L^\infty([0,1]^{D_y})}, \varepsilon/(2C_3\gamma_f))
$$
$$
\leq C_4 S_h L_h \log \left( \frac{L_h W_h (B_h \vee 1) C_3 \gamma_f}{\varepsilon} \right). \tag{B.77}
$$

Plug in (B.73) and let $\theta = 2C_3 \exp(-C_1' R_f^2)$,

$$
\begin{aligned}
\mathcal{R}_K(\Phi_r) &\leq C_0 \left( \theta + \int_\theta^{2\sqrt{r}} \sqrt{\frac{C_4 S_h L_h \log\left( \frac{L_h W_h (B_h \vee 1) C_3 \gamma_f}{\varepsilon} \right)}{K}} \, d\varepsilon \right) \\
&\leq C_0 \left( 2C_3 \exp(-C_1' R_f^2) + \sqrt{\frac{C_4' S_h L_h \log\left( \frac{L_h W_h (B_h \vee 1) M_f \gamma_f}{r} \right) \cdot r}{K}} \right) \\
&=: \widetilde{\mathcal{R}}_K(r).
\end{aligned}
\tag{B.78}
$$

According to Lemma B.11 (by setting $B_0 = 0, B = b = C_L$), for some absolute constant $C_5$, with probability no less than $1 - \delta$, we have for any $h \in \mathcal{H}$,

$$
\mathbb{E}_{\mathbb{P} \sim \mathbb{P}_{\text{meta}}} \mathcal{L}(\mathbb{P}, h) \leq \frac{2}{K} \sum_{k=1}^K \mathcal{L}(\mathbb{P}^k, h) + C_5 C_L \left( r_K^* + \frac{\log(\log(K)/\delta)}{K} \right),
\tag{B.79}
$$

$$
\frac{1}{K} \sum_{k=1}^K \mathcal{L}(\mathbb{P}^k, h) \leq 2 \mathbb{E}_{\mathbb{P} \sim \mathbb{P}_{\text{meta}}} \mathcal{L}(\mathbb{P}, h) + C_5 C_L \left( r_K^* + \frac{\log(\log(K)/\delta)}{K} \right),
\tag{B.80}
$$

where $r_K^*$ is the unique fixed point of $\widetilde{\mathcal{R}}_K$. And it is easy to show that for some absolute constant $C_6$,

$$
r_K^* \leq C_6 \left( C_3 \exp(-C_1' R_f^2) + \frac{S_h L_h \log\left( K L_h W_h (B_h \vee 1) M_f \gamma_f \right)}{K} \right).
\tag{B.81}
$$

which concludes the proof. $\qquad\square$

**Theorem B.8** (Thm. 3.6). *Under Assumption 3.1, 3.2, 3.3, there exists some constant $C, C_R$ such that the following holds. Define the empirical minimizer of training task and new task as*

$$
\widehat{\boldsymbol{f}}, \widehat{h} = \operatorname*{arg\,min}_{\boldsymbol{f} \in \mathcal{F}^{\otimes K}, h \in \mathcal{H}} \frac{1}{nK} \sum_{k=1}^K \sum_{i=1}^n \ell(x_i^k, y_i^k, s_{f^k, h}),
\tag{B.82}
$$

$$
\widehat{f}^{\mathbb{P}} := \operatorname*{arg\,min}_{f \in \mathcal{F}} \frac{1}{m} \sum_{i=1}^m \ell(x_i, y_i, s_{f, \widehat{h}}).
\tag{B.83}
$$

*If $R_f \geq C_R \log^{\frac{1}{2}}(nK M_f/\delta)$, then with probability no less than $1 - \delta$, the expected population loss of new task can be bounded by*

$$
\mathbb{E}_{\mathbb{P} \sim \mathbb{P}_{\text{meta}}} \mathbb{E}_{\{(x_i, y_i)\}_{i=1}^m \sim \mathbb{P}} \mathbb{E}_{(x,y) \sim \mathbb{P}} [\ell^{\mathbb{P}}(x, y, s_{\widehat{f}^{\mathbb{P}}, \widehat{h}})]
$$
$$
\lesssim \inf_{h \in \mathcal{H}} \mathbb{E}_{\mathbb{P} \sim \mathbb{P}_{\text{meta}}} \inf_{f \in \mathcal{F}} \mathbb{E}_{(x,y) \sim \mathbb{P}} [\ell^{\mathbb{P}}(x, y, s_{f, h})] + C \left( \frac{\log^3(m) \log \mathcal{N}_{\mathcal{F}}}{m} + \frac{\log^3(nK/\delta) \log \mathcal{N}_{\mathcal{F}}}{n} + \frac{\log(\mathcal{N}_{\mathcal{H}}/\delta)}{K} \right),
\tag{B.84}
$$

*where*

$$
\log \mathcal{N}_{\mathcal{F}} := M_f^2 S_f L_f \log\left( mn L_f W_f (B_f \vee 1) M_f T \log(1/\delta) \right),
\tag{B.85}
$$

$$
\log \mathcal{N}_{\mathcal{H}} := S_h L_h \log\left( nK L_h W_h (B_h \vee 1) M_f \gamma_f \log(1/\delta) \right).
\tag{B.86}
$$

*Proof.*

$$\mathbb{E}_{\mathbb{P}\sim\mathbb{P}_{\text{meta}}}\mathbb{E}_{\{(x_i,y_i)\}_{i=1}^m\sim\mathbb{P}}\mathbb{E}_{(x,y)\sim\mathbb{P}}[\ell^{\mathbb{P}}(x,y,s_{\widehat{f}^{\mathbb{P}},\widehat{h}})]$$

$$\lesssim \mathbb{E}_{\mathbb{P}\sim\mathbb{P}_{\text{meta}}}\inf_{f\in\mathcal{F}}\mathbb{E}_{(x,y)\sim\mathbb{P}}[\ell^{\mathbb{P}}(x,y,s_{f,\widehat{h}})] + C_{xy}\log^3(m)r_x$$

$$\lesssim \frac{1}{K}\sum_{k=1}^{K}\inf_{f\in\mathcal{F}}\mathbb{E}_{(x,y)\sim\mathbb{P}^k}[\ell^{\mathbb{P}^k}(x,y,s_{f,\widehat{h}})] + C_P\left(r_P + \frac{\log(1/\delta)}{K}\right) + C_{xy}\log^3(m)r_x$$

$$\lesssim \frac{1}{K}\sum_{k=1}^{K}\mathbb{E}_{(x,y)\sim\mathbb{P}^k}[\ell^{\mathbb{P}^k}(x,y,s_{\widehat{f}^k,\widehat{h}})] + C_P\left(r_P + \frac{\log(1/\delta)}{K}\right) + C_{xy}\log^3(m)r_x$$

$$\lesssim \inf_{\boldsymbol{f}\in\mathcal{F}^{\otimes K},h\in\mathcal{H}}\frac{1}{K}\sum_{k=1}^{K}\mathbb{E}_{(x,y)\sim\mathbb{P}^k}[\ell^{\mathbb{P}^k}(x,y,s_{f^k,h})] + C_Z\log^3(nK/\delta)\left(r_z + \frac{\log(1/\delta)}{nK}\right)$$

$$\qquad + C_P\left(r_P + \frac{\log(1/\delta)}{K}\right) + C_{xy}\log^3(m)r_x$$

$$\lesssim \inf_{h\in\mathcal{H}}\mathbb{E}_{\mathbb{P}\sim\mathbb{P}_{\text{meta}}}\inf_{f\in\mathcal{F}}\mathbb{E}_{(x,y)\sim\mathbb{P}}[\ell^{\mathbb{P}}(x,y,s_{f,h})] + C_Z\log^3(nK/\delta)\left(r_z + \frac{\log(1/\delta)}{nK}\right)$$

$$\qquad + C_P\left(r_P + \frac{\log(1/\delta)}{K}\right) + C_{xy}\log^3(m)r_x.$$

$$(B.87)$$

Here we apply Proposition B.4 in the first inequality, Proposition B.7 in the second and last inequality, Proposition B.5 in the fourth. Plugging in the definition of $r_z, r_P, r_x$ and $\log\mathcal{N}_\mathcal{F}, \log\mathcal{N}_\mathcal{H}$ and noticing that $R_f \geq C_R\log^{\frac{1}{2}}(nKd_xM_f/\delta) \geq C_R'\log^{\frac{1}{2}}\left(\frac{M_fK}{\log\mathcal{N}_\mathcal{H}}\right)$, we have with probability no less than $1-\delta$,

$$\mathbb{E}_{\mathbb{P}\sim\mathbb{P}_{\text{meta}}}\mathbb{E}_{\{(x_i,y_i)\}_{i=1}^m\sim\mathbb{P}}\mathbb{E}_{(x,y)\sim\mathbb{P}}[\ell^{\mathbb{P}}(x,y,s_{\widehat{f}^{\mathbb{P}},\widehat{h}})]$$

$$\lesssim \inf_{h\in\mathcal{H}}\mathbb{E}_{\mathbb{P}\sim\mathbb{P}_{\text{meta}}}\inf_{f\in\mathcal{F}}\mathbb{E}_{(x,y)\sim\mathbb{P}}[\ell^{\mathbb{P}}(x,y,s_{f,h})] + C\left(\frac{\log^3(m)\log\mathcal{N}_\mathcal{F}}{m} + \frac{\log^3(nK/\delta)\log\mathcal{N}_\mathcal{F}}{n} + \frac{\log(\mathcal{N}_\mathcal{H}/\delta)}{K}\right).$$

$$(B.88)$$

$\square$

## B.4 Auxiliary Lemmas

**Lemma B.9.** *There exists some constant $C_L$ such that for any $h, \mathbb{P}$,*

$$\mathcal{L}(\mathbb{P},h) \leq \mathbb{E}_{t,x_t,y}\|\nabla_x\log p_t(x_t|y)\|^2 \leq C_L. \tag{B.89}$$

*Proof.* Note that

$$\mathbb{E}_{(x,y)\sim\mathbb{P}}[\ell^{\mathbb{P}}(x,y,s_{f,h})] = \mathbb{E}_{(x,y)\sim\mathbb{P}}\mathbb{E}_{t,x_t|x}[\|f(x_t,h(y),t) - \nabla_x\log p_t(x_t|y)\|^2]$$

$$= \mathbb{E}_{t,x_t,y}[\|f(x_t,h(y),t) - \nabla_x\log p_t(x_t|y)\|^2] \tag{B.90}$$

and $0\in\mathcal{F}$, it suffices to show that $\mathbb{E}_{t,x_t,y}[\|\nabla_x\log p_t(x_t|y)\|^2]$ is uniformly bounded for any $\mathbb{P}, h$. According to (B.2),

$$\mathbb{E}_{x_t,y}[\|\nabla_x\log p_t(x_t|y)\|^2] \leq \mathbb{E}_{x_t,y}\mathbb{E}_{x_0|(x_t,y)}[\|\nabla_x\log\phi_t(x_t|x_0)\|^2]$$

$$= \mathbb{E}_{x_0,y}\mathbb{E}_{x_t|x_0}[\|\nabla_x\log\phi_t(x_t|x_0)\|^2] \tag{B.91}$$

$$= \frac{d_x}{\sigma_t^2} = \frac{d_x}{1-e^{-2t}}.$$

On the other hand, by (B.3) and Assumption 3.3,

$$
\begin{aligned}
\mathbb{E}_{x_t,y}[\|\nabla_x \log p_t(x_t|y)\|^2] &\leq \mathbb{E}_{x_t,y}\mathbb{E}_{x_0|(x_t,y)}[\|\nabla_x \log p(x_0|y)\|^2 \cdot e^{2t}] \\
&= \mathbb{E}_{x_0,y}\mathbb{E}_{x_t|x_0}[\|\nabla_x \log p(x_0|y)\|^2 \cdot e^{2t}] \\
&= \mathbb{E}_{x_0,y}[\|\nabla_x \log p(x_0|y)\|^2/\alpha_t^2] \\
&\leq \mathbb{E}_{x_0,y}[(B + L\|x_0\|)^2 \cdot e^{2t}] \\
&\leq C_2' e^{2t}
\end{aligned}
\tag{B.92}
$$

Therefore, we have

$$
\begin{aligned}
\mathcal{L}(\mathbb{P}, h) &\leq \mathbb{E}_{t,x_t,y}[\|\nabla_x \log p_t(x_t|y)\|^2] \\
&\leq \mathbb{E}_t\big[\frac{d_x}{1 - e^{-2t}} \wedge C_2' e^{2t}\big] \\
&\leq 2(C_2' + d_x) =: C_L.
\end{aligned}
\tag{B.93}
$$

$\square$

**Lemma B.10.** *There exists some constant $C_X''$ such that for any $t \in [0, T]$ and $x \in \mathbb{R}^{d_x}, y \in [0, 1]^{D_y}$,*

$$
\mathbb{E}_{x_t|x}\|\nabla_x \log p_t(x_t|y)\|^2 \leq C_X''(\|x\|^6 + 1).
\tag{B.94}
$$

*Proof.* Note that $x_t|x \sim \mathcal{N}(x_t|\alpha_t x, \sigma_t^2 I)$ and by Lemma B.2,

$$
\mathbb{E}_{x_t|x}\|\nabla_x \log p_t(x_t|y)\|^2 \leq \mathbb{E}_{x_t|x} 2\Big[\|\nabla_x \log p_t(0|y)\|^2 + (C_X + C_X'\|x_t\|^2)^2\|x_t\|^2\Big]
\tag{B.95}
$$

Let $q_t(x_0|x_t, y) \propto \phi_t(x_t|x_0)p(x_0|y)$. Since $\phi_t(0|x_0) \propto \exp\left(-\dfrac{\alpha_t^2\|x\|^2}{2\sigma_t^2}\right)$ is decreasing in $\|x\|$, by Fortuin–Kasteleyn–Ginibre inequality,

$$
\mathbb{E}_{q_t(x_0|0,y)}\|x_0\|^2 \leq \mathbb{E}_{p(x_0|y)}\|x_0\|^2 \leq C_0.
\tag{B.96}
$$

According to (B.2),

$$
\|\nabla_x \log p_t(0|y)\|^2 \leq \frac{\alpha_t^2}{\sigma_t^4}\mathbb{E}_{q_t(x_0|0,y)}\|x_0\|^2 \leq \frac{C_0\alpha_t^2}{\sigma_t^4}.
\tag{B.97}
$$

By (B.3), we also have

$$
\|\nabla_x \log p_t(0|y)\|^2 \leq \frac{1}{\alpha_t^2}\mathbb{E}_{q_t(x_0|0,y)}\|\nabla_x \log p(x_0|y)\|^2 \leq \frac{1}{\alpha_t^2}\mathbb{E}_{q_t(x_0|0,y)}[(B+L\|x_0\|)^2] \leq \frac{2(B^2 + L^2 C_0)}{\alpha_t^2}.
\tag{B.98}
$$

Combine the two inequalities,

$$
\|\nabla_x \log p_t(0|y)\|^2 \leq (B^2 + (L^2 + 1)C_0) \cdot (\frac{\alpha_t^2}{\sigma_t^4} \wedge \frac{1}{\alpha_t^2}) \leq 2(B^2 + (L^2 + 1)C_0).
\tag{B.99}
$$

Plug in (B.95) and we obtain for some constant $C_X''$,

$$
\begin{aligned}
\mathbb{E}_{x_t|x}\|\nabla_x \log p_t(x_t|y)\|^2 &\leq \mathbb{E}_{x_t|x} 2\Big[(C_X + C_X'\|x_t\|^2)^2\|x_t\|^2\Big] + 2(B^2 + (L^2 + 1)C_0) \\
&\leq C_X''(\|x\|^6 + 1).
\end{aligned}
\tag{B.100}
$$

$\square$

**Lemma B.11.** *Let $\Phi$ be a class of functions on domain $\Omega$ and $\mathbb{P}$ be a probability distribution over $\Omega$. Suppose that for any $\varphi \in \Phi$, $\|\varphi\|_{L^\infty(\Omega)} \leq b$, $\mathbb{E}_{\mathbb{P}}[\varphi] \geq 0$, and $\mathbb{E}_{\mathbb{P}}[\varphi^2] \leq B\mathbb{E}_{\mathbb{P}}[\varphi] + B_0$ for some $b, B, B_0 \geq 0$. Let $x_1, \cdots, x_n \overset{i.i.d.}{\sim} \mathbb{P}$ and $\phi_n$ be a positive, non-decreasing and sub-root function such that*

$$
\mathcal{R}_n(\Phi_r) := \mathbb{E}_{\boldsymbol{\sigma}} \sup_{\varphi \in \Phi_r} \Big| \frac{1}{n}\sum_{i=1}^n \sigma_i \varphi(x_i) \Big| \leq \phi_n(r).
\tag{B.101}
$$

where $\Phi_r := \left\{ \varphi \in \Phi : \frac{1}{n} \sum_{i=1}^{n} (\varphi(x_i))^2 \leq r \right\}$. *Define the largest fixed point of $\phi_n$ as $r_n^*$. Then for some absolute constant $C'$, with probability no less than $1 - \delta$, it holds that for any $\varphi \in \Phi$,*

$$\mathbb{E}_{\mathbb{P}}[\varphi] \leq \frac{2}{n} \sum_{i=1}^{n} \varphi(x_i) + C'(B \vee b) \left( r_n^* + \frac{\log\left((\log n)/\delta\right)}{n} \right) + C' \sqrt{\frac{B_0 \log\left((\log n)/\delta\right)}{n}}, \tag{B.102}$$

$$\frac{1}{n} \sum_{i=1}^{n} \varphi(x_i) \leq 2\mathbb{E}_{\mathbb{P}}[\varphi] + C'(B \vee b) \left( r_n^* + \frac{\log\left((\log n)/\delta\right)}{n} \right) + C' \sqrt{\frac{B_0 \log\left((\log n)/\delta\right)}{n}}. \tag{B.103}$$

*Proof.* We follow the procedures in Bousquet [2002]. Let $\epsilon_j = b2^{-j}$ and consider a sequence of classes

$$\Phi^{(j)} := \{ \varphi \in \Phi : \epsilon_{j+1} < \mathbb{E}_{\mathbb{P}}[\varphi] \leq \epsilon_j \}. \tag{B.104}$$

Note that $\Phi = \cup_{j \geq 0} \Phi^{(j)}$ and for $\varphi \in \Phi^{(j)}$, $\mathbb{E}_{\mathbb{P}}[\varphi^2] \leq B\epsilon_k + B_0$. Let $j_0 = \lfloor \log_2 n \rfloor$. Then by Bousquet [2002, Lemma 6.1], it holds that with probability no less than $1 - \delta$, for any $j \leq j_0$ and $\varphi \in \Phi^{(j)}$,

$$\left| \frac{1}{n} \sum_{i=1}^{n} \varphi(x_i) - \mathbb{E}_{\mathbb{P}}[\varphi] \right| \lesssim \mathcal{R}_n(\Phi^{(j)}) + \sqrt{\frac{(B\epsilon_j + B_0) \log\left(\log(b/\epsilon_j)/\delta\right)}{n}} + \frac{b \log\left(\log(b/\epsilon_j)/\delta\right)}{n}, \tag{B.105}$$

$$\left| \frac{1}{n} \sum_{i=1}^{n} (\varphi(x_i))^2 - \mathbb{E}_{\mathbb{P}}[\varphi^2] \right| \lesssim b\mathcal{R}_n(\Phi^{(j)}) + \sqrt{\frac{b^2(B\epsilon_j + B_0) \log\left(\log(b/\epsilon_j)/\delta\right)}{n}} + \frac{b^2 \log\left(\log(b/\epsilon_j)/\delta\right)}{n}. \tag{B.106}$$

Besides, for $\varphi \in \cup_{k > k_0} \Phi^{(j)} =: \Phi^{(j_0:)}$,

$$\left| \frac{1}{n} \sum_{i=1}^{n} \varphi(x_i) - \mathbb{E}_{\mathbb{P}}[\varphi] \right| \lesssim \mathcal{R}_n(\Phi^{(j_0:)}) + \sqrt{\frac{(B\epsilon_{j_0} + B_0) \log\left(\log(n)/\delta\right)}{n}} + \frac{b \log\left((\log n)/\delta\right)}{n} \tag{B.107}$$

From now on we reason on the conjunction of (B.105), (B.106) and (B.107). Define

$$U_j = B\epsilon_j + B_0 + b\mathcal{R}_n(\Phi^{(k)}) + \sqrt{\frac{b^2(B\epsilon_j + B_0) \log\left(\log(b/\epsilon_j)/\delta\right)}{n}} + \frac{b^2 \log\left(\log(b/\epsilon_j)/\delta\right)}{n}. \tag{B.108}$$

and thus for any $\varphi \in \Phi^{(j)}$, we have $\frac{1}{n} \sum_{i=1}^{n} (\varphi(x_i))^2 \leq CU_j$ for some absolute constant $C$ by (B.106), indicating that $\mathcal{R}_n(\Phi^{(j)}) \leq \phi_n(CU_j) \leq \sqrt{C}\phi_n(U_j)$. For any $j \leq j_0$,

$$U_j \leq 2(B\epsilon_j + B_0) + b\sqrt{C}\phi_n(U_j) + \frac{2b^2 \log\left((\log n)/\delta\right)}{n}. \tag{B.109}$$

Since $\phi_n$ is non-decreasing and sub-root, the inequality above implies that

$$U_j \lesssim b^2 r_n^* + B\epsilon_j + B_0 + \frac{b^2 \log\left((\log n)/\delta\right)}{n} =: r_n(\epsilon_j). \tag{B.110}$$

Therefore, for any $\varphi \in \Phi^{(j)}, j \leq j_0$, by (B.105),

$$\left| \frac{1}{n} \sum_{i=1}^{n} \varphi(x_i) - \mathbb{E}_{\mathbb{P}}[\varphi] \right| \lesssim \phi_n(r_n(\epsilon_j)) + \sqrt{\frac{(B\epsilon_j + B_0) \log\left((\log n)/\delta\right)}{n}} + \frac{b \log\left((\log n)/\delta\right)}{n}$$

$$=: F_n(\epsilon_j). \tag{B.111}$$

Noticing that $\mathbb{E}_{\mathbb{P}}[\varphi] \leq \epsilon_j \leq 2\mathbb{E}_{\mathbb{P}}[\varphi]$, it reduces to

$$\left| \frac{1}{n} \sum_{i=1}^{n} \varphi(x_i) - \mathbb{E}_{\mathbb{P}}[\varphi] \right| \lesssim F_n(\mathbb{E}_{\mathbb{P}}[\varphi]). \tag{B.112}$$

Hence we have by noting that $F_n$ is also a non-decreasing sub-root function,

$$\mathbb{E}_{\mathbb{P}}[\varphi] \leq \frac{2}{n} \sum_{i=1}^{n} \varphi(x_i) + C'(B \vee b) \left( r_n^* + \frac{\log\left((\log n)/\delta\right)}{n} \right) + C'\sqrt{\frac{B_0 \log\left((\log n)/\delta\right)}{n}}, \tag{B.113}$$

$$\frac{1}{n} \sum_{i=1}^{n} \varphi(x_i) \leq 2\mathbb{E}_{\mathbb{P}}[\varphi] + C'(B \vee b) \left( r_n^* + \frac{\log\left((\log n)/\delta\right)}{n} \right) + C'\sqrt{\frac{B_0 \log\left((\log n)/\delta\right)}{n}}. \tag{B.114}$$

Here $C'$ is an absolute constant. Moreover, when $\varphi \in \Phi^{(j)}$ for $j > j_0$, we have $\mathbb{E}_{\mathbb{P}}[\varphi] \leq \frac{b}{n}$, and according to (B.107),

$$\left| \frac{1}{n} \sum_{i=1}^{n} \varphi(x_i) - \mathbb{E}_{\mathbb{P}}[\varphi] \right| \lesssim F_n(\varepsilon_{j_0}). \tag{B.115}$$

Hence the same bounds apply, which completes the proof. $\qquad\square$

**Lemma B.12.** *Let $\Phi$ be a class of functions on domain $\Omega$, $\mathbb{P}^1, \cdots, \mathbb{P}^K$ be probability distributions over $\Omega$, and $\widehat{\mathbb{P}}^{(K)} = \frac{1}{K} \sum_{k=1}^{K} \delta_{\mathbb{P}^k}$. Suppose that for any $\varphi \in \Phi$, $\|\varphi\|_{L^\infty(\Omega)} \leq b$, $\mathbb{E}_{\widehat{\mathbb{P}}^{(K)}}[\varphi] \geq 0$, and $\mathbb{E}_{\widehat{\mathbb{P}}^{(K)}}[\varphi^2] \leq B\mathbb{E}_{\widehat{\mathbb{P}}^{(K)}}[\varphi] + B_0$ for some $b, B, B_0 \geq 0$. Let $x_1^k, \cdots, x_n^k \overset{i.i.d.}{\sim} \mathbb{P}^k$ for any $k$ and all $(x_i^k)_{i,k}$ are independent. Let $\phi_{K,n}$ be a positive, non-decreasing and sub-root function such that*

$$\mathcal{R}_{K,n}(\Phi_r) := \mathbb{E}_{\boldsymbol{\sigma}} \sup_{\varphi \in \Phi_r} \left| \frac{1}{nK} \sum_{k=1}^{K} \sum_{i=1}^{n} \sigma_i^k \varphi(x_i^k) \right| \leq \phi_{K,n}(r). \tag{B.116}$$

*where $\Phi_r := \left\{ \varphi \in \Phi : \frac{1}{nK} \sum_{k=1}^{K} \sum_{i=1}^{n} (\varphi(x_i^k))^2 \leq r \right\}$. Define the largest fixed point of $\phi_{K,n}$ as $r_{K,n}^*$. Then for some absolute constant $C'$, with probability no less than $1 - \delta$, it holds that for any $\varphi \in \Phi$,*

$$\mathbb{E}_{\widehat{\mathbb{P}}^{(K)}}[\varphi] \leq \frac{2}{nK} \sum_{k=1}^{K} \sum_{i=1}^{n} \varphi(x_i) + C'(B \vee b) \left( r_{K,n}^* + \frac{\log\left((\log nK)/\delta\right)}{nK} \right) + C'\sqrt{\frac{B_0 \log\left((\log nK)/\delta\right)}{nK}}, \tag{B.117}$$

$$\frac{1}{nK} \sum_{k=1}^{K} \sum_{i=1}^{n} \varphi(x_i^k) \leq 2\mathbb{E}_{\widehat{\mathbb{P}}^{(K)}}[\varphi] + C'(B \vee b) \left( r_{K,n}^* + \frac{\log\left((\log nK)/\delta\right)}{nK} \right) + C'\sqrt{\frac{B_0 \log\left((\log nK)/\delta\right)}{nK}}. \tag{B.118}$$

*Proof.* We follow the procedures in Bousquet [2002]. Let $\epsilon_k = b2^{-k}$ and consider a sequence of classes

$$\Phi^{(j)} := \{\varphi \in \Phi : \epsilon_{j+1} < \mathbb{E}_{\widehat{\mathbb{P}}^{(K)}}[\varphi] \leq \epsilon_j\}. \tag{B.119}$$

Note that $\Phi = \cup_{j\geq 0}\Phi^{(j)}$ and for $\varphi \in \Phi^{(j)}$, $\mathbb{E}_{\widehat{\mathbb{P}}(K)}[\varphi^2] \leq B\epsilon_j + B_0$. Let $j_0 = \lfloor \log_2(nK) \rfloor$. Then by Massart [2000, Theorem 3], with probability no less than $1 - \delta$, for any $j \leq j_0$ and $\varphi \in \Phi^{(j)}$,

$$\left| \frac{1}{nK} \sum_{k=1}^{K} \sum_{i=1}^{n} \varphi(x_i^k) - \mathbb{E}_{\widehat{\mathbb{P}}(K)}[\varphi] \right| \lesssim \mathcal{R}_{K,n}(\Phi^{(j)}) + \sqrt{\frac{(B\epsilon_j + B_0)\log\big(\log(b/\epsilon_j)/\delta\big)}{nK}} + \frac{b\log\big(\log(b/\epsilon_j)/\delta\big)}{nK},$$

$$(B.120)$$

$$\left| \frac{1}{nK} \sum_{k=1}^{K} \sum_{i=1}^{n} (\varphi(x_i^k))^2 - \mathbb{E}_{\widehat{\mathbb{P}}(K)}[\varphi^2] \right| \lesssim b\mathcal{R}_{K,n}(\Phi^{(j)}) + \sqrt{\frac{b^2(B\epsilon_j + B_0)\log\big(\log(b/\epsilon_j)/\delta\big)}{nK}} + \frac{b^2\log\big(\log(b/\epsilon_j)/\delta\big)}{nK}.$$

$$(B.121)$$

Besides, for any $\varphi \in \cup_{j > j_0}\Phi^{(j)} =: \Phi^{(j_0:)}$,

$$\left| \frac{1}{nK} \sum_{k=1}^{K} \sum_{i=1}^{n} \varphi(x_i^k) - \mathbb{E}_{\widehat{\mathbb{P}}(K)}[\varphi] \right| \lesssim \mathcal{R}_{K,n}(\Phi^{(j_0:)}) + \sqrt{\frac{(B\epsilon_{j_0} + B_0)\log\big((\log nK)/\delta\big)}{nK}} + \frac{b\log\big((\log nK)/\delta\big)}{nK}.$$

$$(B.122)$$

From now on we reason on the conjunction of (B.120), (B.121) and (B.122). Define

$$U_j = B\epsilon_j + B_0 + b\mathcal{R}_{K,n}(\Phi^{(j)}) + \sqrt{\frac{b^2(B\epsilon_j + B_0)\log\big(\log(b/\epsilon_j)/\delta\big)}{nK}} + \frac{b^2\log\big(\log(b/\epsilon_j)/\delta\big)}{nK}.$$

$$(B.123)$$

and thus for any $\varphi \in \Phi^{(j)}$, we have $\frac{1}{nK}\sum_{k=1}^{K}\sum_{i=1}^{n}(\varphi(x_i^k))^2 \leq CU_j$ for some absolute constant $C$ by (B.121), indicating that $\mathcal{R}_{K,n}(\Phi^{(j)}) \leq \phi_{K,n}(CU_j) \leq \sqrt{C}\phi_{K,n}(U_j)$. For any $j \leq j_0$,

$$U_j \leq 2(B\epsilon_j + B_0) + b\sqrt{C}\phi_{K,n}(U_j) + \frac{2b^2\log\big((\log nK)/\delta\big)}{nK}. \quad (B.124)$$

Since $\phi_{K,n}$ is non-decreasing and sub-root, the inequality above implies that

$$U_j \lesssim b^2 r_{K,n}^* + B\epsilon_j + B_0 + \frac{b^2\log\big((\log nK)/\delta\big)}{nK} =: r_{K,n}(\epsilon_j). \quad (B.125)$$

Therefore, for any $\varphi \in \Phi^{(j)}, j \leq j_0$, by (B.120),

$$\left| \frac{1}{nK} \sum_{k=1}^{K} \sum_{i=1}^{n} \varphi(x_i^k) - \mathbb{E}_{\widehat{\mathbb{P}}(K)}[\varphi] \right| \lesssim \phi_{K,n}(r_{K,n}(\epsilon_j)) + \sqrt{\frac{(B\epsilon_j + B_0)\log\big((\log nK)/\delta\big)}{nK}} + \frac{b\log\big((\log nK)/\delta\big)}{nK}$$

$$=: F_{K,n}(\epsilon_j).$$

$$(B.126)$$

Noticing that $\mathbb{E}_{\widehat{\mathbb{P}}(K)}[\varphi] \leq \epsilon_j \leq 2\mathbb{E}_{\widehat{\mathbb{P}}(K)}[\varphi]$, it reduces to

$$\left| \frac{1}{nK} \sum_{k=1}^{K} \sum_{i=1}^{n} \varphi(x_i^k) - \mathbb{E}_{\widehat{\mathbb{P}}(K)}[\varphi] \right| \lesssim F_{K,n}(\mathbb{E}_{\widehat{\mathbb{P}}(K)}[\varphi]). \quad (B.127)$$

Hence we have by noting that $F_{K,n}$ is also a non-decreasing sub-root function,

$$\mathbb{E}_{\widehat{\mathbb{P}}(K)}[\varphi] \leq \frac{2}{nK} \sum_{k=1}^{K} \sum_{i=1}^{n} \varphi(x_i^k) + C'(B \vee b)\left( r_{K,n}^* + \frac{\log\big((\log nK)/\delta\big)}{nK} \right) + C'\sqrt{\frac{B_0\log\big((\log nK)/\delta\big)}{nK}},$$

$$(B.128)$$

$$\frac{1}{nK} \sum_{k=1}^{K} \sum_{i=1}^{n} \varphi(x_i^k) \leq 2\mathbb{E}_{\widehat{\mathbb{P}}(K)}[\varphi] + C'(B \vee b)\left( r_{K,n}^* + \frac{\log\big((\log nK)/\delta\big)}{nK} \right) + C'\sqrt{\frac{B_0\log\big((\log nK)/\delta\big)}{nK}}.$$

$$(B.129)$$

Here $C'$ is an absolute constant. Moreover, when $\varphi \in \Phi^{(j)}$ for $j > j_0$, we have $\mathbb{E}_{\widehat{\mathbb{P}}(K)}[\varphi] \leq \dfrac{b}{nK}$, and according to (B.122),

$$\left| \frac{1}{nK} \sum_{k=1}^{K} \sum_{i=1}^{n} \varphi(x_i^k) - \mathbb{E}_{\widehat{\mathbb{P}}(K)}[\varphi] \right| \lesssim F_{K,n}(\varepsilon_{j_0}). \tag{B.130}$$

Hence the same bounds apply, which completes the proof. $\qquad\square$

## B.5 Verifying Task Diversity Assumption

When $\mathcal{F}$ is linear function class, Tripuraneni et al. [2020] provides an explicit bound on $(\nu, \Delta)$. However, in general, performing a fine-grained analysis is challenging, especially for complex function classes such as neural networks. In the following proposition, we present a very pessimistic bound for $(\nu, \Delta)$ based on density ratio, which is independent of the specific choice of hypothesis classes $\mathcal{F}$ and $\mathcal{H}$.

**Proposition B.13.** *Suppose $\mathcal{F} = \mathbf{conv}(\mathcal{F})$, and $\inf\limits_{x,y} \dfrac{p^k(x,y)}{p^0(x,y)} \geq \lambda_k$ for any $1 \leq k \leq K$. Let $\lambda = \sum\limits_{k=1}^{K} \lambda_k$. Then $\mathbb{P}^1, \cdots, \mathbb{P}^K$ are $(\widetilde{\nu}, \widetilde{\Delta})$-diverse over $\mathbb{P}^0$ with $\widetilde{\nu} = \lambda/(2K)$,*

$$\widetilde{\Delta} = 2\mathbb{E}_{(x,y)\sim\mathbb{P}^0} \mathbb{E}_{t,x_t} \left[ \left\| \frac{1}{\lambda} \sum_{k=1}^{K} \lambda_k \nabla \log p_t^k(x_t|y) - \nabla \log p_t^0(x_t|y) \right\|^2 \right]. \tag{B.131}$$

We mention that the only requirement is $\mathcal{F}$ is a convex hull of itself, which can be easily satisfied by most hypothesis classes such as neural networks. More refined analysis on specific neural network class is an interesting future work.

*Proof.* For any $h \in \mathcal{H}$, let $f^k \in \mathcal{F}$ be the corresponding minimizer for $1 \leq k \leq K$. Further define $\lambda = \sum\limits_{k=1}^{K} \lambda_k$ and $\widetilde{f}^0 = \dfrac{1}{\lambda} \sum\limits_{k=1}^{K} \lambda_k f^k \in \mathbf{conv}(\mathcal{F}) \in \mathcal{F}$. Then we have

$$
\begin{aligned}
L^{\mathbb{P}^0}(s_{\widetilde{f}^0, h}) &= \mathbb{E}_{\mathbb{P}^0} \left[ \|\widetilde{f}^0(x_t, h(y), t) - \nabla \log p_t^0(x_t|y)\|^2 \right] \\
&\leq 2\mathbb{E}_{\mathbb{P}^0} \left[ \|\widetilde{f}^0(x_t, h(y), t) - \sum_{k=1}^{K} \frac{\lambda_k}{\lambda} \nabla \log p_t^k(x_t|y)\|^2 + \| \sum_{k=1}^{K} \frac{\lambda_k}{\lambda} \nabla \log p_t^k(x_t|y) - \nabla \log p_t^0(x_t|y)\|^2 \right] \\
&\leq \frac{2}{\lambda} \sum_{k=1}^{K} \mathbb{E}_{\mathbb{P}^0} \lambda_k \left[ \|f^k(x_t, h(y), t) - \nabla \log p_t^k(x_t|y)\|^2 \right] + \widetilde{\Delta} \\
&\leq \frac{2}{\lambda} \sum_{k=1}^{K} \mathbb{E}_{\mathbb{P}^k} \left[ \|f^k(x_t, h(y), t) - \nabla \log p_t^k(x_t|y)\|^2 \right] + \widetilde{\Delta} \\
&= \frac{1}{\widetilde{\nu}} \inf_{\boldsymbol{f} \in \mathcal{F}^{\otimes K}} \frac{1}{K} \sum_{k=1}^{K} L^{\mathbb{P}^k}(s_{f^k, h}) + \widetilde{\Delta}.
\end{aligned}
\tag{B.132}
$$

We conclude the proof by noticing that $\inf\limits_{f\in\mathcal{F}} L^{\mathbb{P}^0}(s_{f,h}) \leq L^{\mathbb{P}^0}(s_{\widetilde{f}^0, h})$. $\qquad\square$

# C   Proofs in Section 4

## C.1   Proofs of Score Network Approximation

**Theorem C.1** (Thm. 4.1). *Under Assumption 3.1, 3.2, 3.3, to achieve $R_f \geq C_R \log^{\frac{1}{2}}(nKM_f/\delta)$ and*

$$\inf_{h \in \mathcal{H}} \frac{1}{K} \sum_{k=1}^{K} \inf_{f \in \mathcal{F}} \mathbb{E}_{(x,y) \sim \mathbb{P}^k}[\ell^{\mathbb{P}^k}(x, y, s_{f,h})] = \mathcal{O}\left(\log^2(nK/(\varepsilon\delta))\varepsilon^2\right), \text{ (transfer learning)} \quad \text{(C.1)}$$

$$\inf_{h \in \mathcal{H}} \mathbb{E}_{\mathbb{P} \sim \mathbb{P}_{meta}} \inf_{f \in \mathcal{F}} \mathbb{E}_{(x,y) \sim \mathbb{P}}[\ell^{\mathbb{P}}(x, y, s_{f,h})] = \mathcal{O}\left(\log^2(nK/(\varepsilon\delta))\varepsilon^2\right), \text{ (meta-learning)} \quad \text{(C.2)}$$

*the configuration of $\mathcal{F} = NN_f(L_f, W_f, M_f, S_f, B_f, R_f, \gamma_f), \mathcal{H} = NN_h(L_h, W_h, S_h, B_h)$ should satisfy*

$$
\begin{aligned}
L_f &= \mathcal{O}\left(\log\left(\frac{\log(nK/(\varepsilon\delta))}{\varepsilon}\right)\right), W_f = \mathcal{O}\left(\frac{\log^{3(d_x+d_y)/2}(nK/(\varepsilon\delta))}{\varepsilon^{d_x+d_y+1}T_0^3}\right), \\
S_f &= \mathcal{O}\left(\frac{\log^{3(d_x+d_y)/2+1}(nK/(\varepsilon\delta))}{\varepsilon^{d_x+d_y+1}T_0^3}\right), B_f = \mathcal{O}\left(\frac{T\log^{\frac{3}{2}}(nK/(\varepsilon\delta))}{\varepsilon}\right), \\
R_f &= \mathcal{O}\left(\log^{\frac{1}{2}}(nK/(\varepsilon\delta))\right), M_f = \mathcal{O}\left(\log^3(nK/(\varepsilon\delta))\right), \gamma_f = \mathcal{O}\left(\log(nK/(\varepsilon\delta))\right),
\end{aligned}
\quad \text{(C.3)}
$$

$$
\begin{aligned}
L_h &= \mathcal{O}\left(\log(1/\varepsilon)\right), W_h = \mathcal{O}\left(\varepsilon^{-D_y}\log(1/\varepsilon)\right), \\
S_h &= \mathcal{O}\left(\varepsilon^{-D_y}\log^2(1/\varepsilon)\right), B_h = \mathcal{O}(1).
\end{aligned}
\quad \text{(C.4)}
$$

*Here $\mathcal{O}(\cdot)$ hides all the polynomial factors of $d_x, d_y, D_y, C_1, C_2, L, B$.*

*Proof.* With a little abuse of notation, in transfer learning setting, we define $\mathbb{P}_{\text{meta}} := \frac{1}{K} \sum_{k=1}^{K} \delta_{\mathbb{P}^k}$ and it directly reduces to meta-learning case. Therefore, we only focus on the proof in meta-learning.

We first decompose the misspecification error into two components: representation error and score approximation error.

$$
\begin{aligned}
&\inf_{h \in \mathcal{H}} \mathbb{E}_{\mathbb{P} \sim \mathbb{P}_{\text{meta}}} \inf_{f \in \mathcal{F}} \mathbb{E}_{(x,y) \sim \mathbb{P}}[\ell^{\mathbb{P}}(x, y, s_{f,h})] \\
&= \inf_{h \in \mathcal{H}} \mathbb{E}_{\mathbb{P} \sim \mathbb{P}_{\text{meta}}} \inf_{f \in \mathcal{F}} \mathbb{E}_{(x,y) \sim \mathbb{P}} \mathbb{E}_{t,x_t|x}[\|f(x_t, h(y), t) - f_*^{\mathbb{P}}(x_t, h_*(y), t)\|^2] \\
&\leq \inf_{h \in \mathcal{H}} \mathbb{E}_{\mathbb{P} \sim \mathbb{P}_{\text{meta}}} \inf_{f \in \mathcal{F}} \mathbb{E}_{(x,y) \sim \mathbb{P}} \mathbb{E}_{t,x_t|x} 2\left[\|f(x_t, h(y), t) - f(x_t, h_*(y), t)\|^2 + \|f(x_t, h_*(y), t) - f_*^{\mathbb{P}}(x_t, h_*(y), t)\|^2\right].
\end{aligned}
\quad \text{(C.5)}
$$

Further note that for any $f \in \mathcal{F}$,

$$
\begin{aligned}
\mathbb{E}_{(x,y) \sim \mathbb{P}} \mathbb{E}_{t,x_t|x}[\|f(x_t, h(y), t) - f(x_t, h_*(y), t)\|^2] &\leq \mathbb{E}_{t,x_t,y}\|f(x_t, h(y), t) - f(x_t, h_*(y), t)\|^2 \cdot \mathbb{1}_{\|x_t\| \leq R_f} \\
&\quad + 8M_f^2 \exp(-C_1' R_f^2) \\
&\leq \mathbb{E}_{y \sim \mathbb{P}}[\gamma_f^2 \|h(y) - h_*(y)\|^2] + 8M_f^2 \exp(-C_1' R_f^2),
\end{aligned}
\quad \text{(C.6)}
$$

where $\Omega_{R_f} = [-R_f, R_f]^{d_x} \times [0,1]^{d_y} \times [T_0, T]$. By Proposition C.2, C.3,

$$
\begin{aligned}
&\inf_{h \in \mathcal{H}} \mathbb{E}_{\mathbb{P} \sim \mathbb{P}_{\text{meta}}} \inf_{f \in \mathcal{F}} \mathbb{E}_{(x,y) \sim \mathbb{P}} [\ell^{\mathbb{P}}(x, y, s_{f,h})] \\
&\leq \inf_{h \in \mathcal{H}} \mathbb{E}_{\mathbb{P} \sim \mathbb{P}_{\text{meta}}} \mathbb{E}_{y \sim \mathbb{P}} [2\gamma_f^2 \|h(y) - h_*(y)\|^2] + 16 M_f^2 \exp(-C_1' R_f^2) \\
&\qquad + \mathbb{E}_{\mathbb{P} \sim \mathbb{P}_{\text{meta}}} \inf_{f \in \mathcal{F}} 2\|f(x_t, h_*(y), t) - f_*^{\mathbb{P}}(x_t, h_*(y), t)\|^2 \\
&\leq 2 \inf_{h \in \mathcal{H}} \gamma_f^2 \|h - h_*\|_{L^\infty([0,1]^{D_y})}^2 + 16 M_f^2 \exp(-C_1' R_f^2) \\
&\qquad + 2\mathbb{E}_{\mathbb{P} \sim \mathbb{P}_{\text{meta}}} \inf_{f \in \mathcal{F}} \|f(x_t, h_*(y), t) - f_*^{\mathbb{P}}(x_t, h_*(y), t)\|^2 \\
&\lesssim \left( \log^2(nK/(\varepsilon\delta)) d_y + d_x \right) \varepsilon^2 \\
&= \mathcal{O}\left( \log^2(nK/(\varepsilon\delta))\varepsilon^2 \right).
\end{aligned}
\tag{C.7}
$$

$\square$

**Proposition C.2.** *To achieve $R_f \geq C_R \log^{\frac{1}{2}}(nKM_f/\delta)$ and*

$$
\inf_{f \in \mathcal{F}} \mathbb{E}_{(x,y) \sim \mathbb{P}} \mathbb{E}_{t, x_t|x} [\|f(x_t, h_*(y), t) - f^{\mathbb{P}}(x_t, h_*(y), t)\|^2] \leq d_x \varepsilon^2,
\tag{C.8}
$$

*the configuration of $\mathcal{F} = NN_f(L_f, W_f, M_f, S_f, B_f, R_f, \gamma_f)$ should satisfy*

$$
\begin{aligned}
&L_f = \mathcal{O}\left( \log\left( \frac{\log(nK/(\varepsilon\delta))}{\varepsilon} \right) \right), W_f = \mathcal{O}\left( \frac{\log^{3(d_x+d_y)/2}(nK/(\varepsilon\delta))}{\varepsilon^{d_x+d_y+1} T_0^3} \right), \\
&S_f = \mathcal{O}\left( \frac{\log^{3(d_x+d_y)/2+1}(nK/(\varepsilon\delta))}{\varepsilon^{d_x+d_y+1} T_0^3} \right), B_f = \mathcal{O}\left( \frac{T \log^{\frac{3}{2}}(nK/(\varepsilon\delta))}{\varepsilon} \right), \\
&R_f = \mathcal{O}\left( \log^{\frac{1}{2}}(nK/(\varepsilon\delta)) \right), M_f = \mathcal{O}\left( \log^3(nK/(\varepsilon\delta)) \right), \gamma_f = \mathcal{O}(\log(nK/(\varepsilon\delta))).
\end{aligned}
\tag{C.9}
$$

*Here $\mathcal{O}(\cdot)$ hides all the polynomial factors of $d_x, d_y, D_y, C_1, C_2, L, B$.*

*Proof.* For notation simplicity, we will $f_* = f_*^{\mathbb{P}}$ throughout the proof. Our procedures consist of two main steps. The first is to clip the whole input space to a bounded set $\Omega_{R_f} := [-R_f, R_f]^{d_x} \times [0,1]^{d_y} \times [T_0, T]$ thanks to the light tail property of $\mathbb{P}$. Then we approximate $f_*^{\mathbb{P}}$ on $\Omega_{R_f}$.

By Lemma B.2 and C.6, $f_*$ is $\gamma_1$-Lipschitz in $x$, $\gamma_2$-Lipschitz in $w$, and $\gamma_3$-Lipshcitz in $t$ in a bounded domain $\Omega_{R_f}$, where $\gamma_1 = C_X + C_X' R_f^2, \gamma_2 = C_X + C_X' R_f, \gamma_3 = \dfrac{C_s R_f^3}{T_0^3}$.

We first rescale the input domain by $x' = \dfrac{x}{2R_f} + \dfrac{1}{2}, w' = w, t' = t/T$, which can be implemented by a single ReLU layer. Denote $v = (x', w', t')$. We only need to approximate $g(v) := f_*(R_f(2x' - 1), w', Tt')$ defined on $\Omega := [0,1]^{d_x+d_y} \times [T_0/T, 1]$. And $g$ is $\gamma_x := 2\gamma_1 R_f$-Lipschitz in $x'$, $\gamma_w := \gamma_2$-Lipschitz in $w'$ and $\gamma_t := \gamma_2 T$-Lipschitz in $t'$. We will approximate each coordinate of $g = [g_1, \cdots, g_{d_x}]^\top$ separately and then concatenate them together.

Now we partition the domain $\Omega$ into non-overlapping regions. For the first $d_x + d_y$ dimensions, the space $[0,1]^{d_x+d_y}$ is uniformly divided into hypercubes with an edge length of $e_1$. For the last dimension, the interval $[T_0/T, 1]$ is divided into subintervals of length $e_2$, where the values of $e_1$ and $e_2$ will be specified later. Let the number of intervals in each partition be $N_1 = \lceil 1/e_1 \rceil$ and $N_2 = \lceil 1/e_2 \rceil$, respectively.

Let $u = [u_1, \cdots, u_{d_x+d_y}] \in \{0, \cdots, N_1 - 1\}^{d_x+d_y}$ be a multi-index. Define

$$
\bar{g}_i(x', w', t') = \sum_{u, j} g_i(u/N_1, j/N_2) \Psi_{u,j}(x', w', t'),
\tag{C.10}
$$

where $\Psi$ is the coordinate-wise product of trapezoid function:

$$\Psi_{u,j}(x', w', t') := \psi\big(3N_2(t' - j/N_2)\big) \prod_{r=1}^{d_x} \psi\big(3N_1(x'_r - u_r/N_1)\big) \prod_{r=1}^{d_y} \psi\big(3N_1(w'_r - u_{r+d_x}/N_1)\big),$$

$$\tag{C.11}$$

$$\psi(a) := \left\{ \begin{array}{ll} 1, & |a| < 1 \\ 2 - |a|, & 1 \le |a| < 2 \\ 0, & |a| >\ge 2 \end{array} \right. \tag{C.12}$$

We claim that $\bar{g}_i$ is an approximation to $g_i$ since for any $o' = (x', w') \in [0,1]^{d_x + d_y}, t' \in [T_0/T, 1]$,

$$\sup_{o',t'} \left| \bar{g}_i(o', t') - g_i(o', t') \right| \le \sup_{o',t'} \left| \sum_{u,j} (g_i(\frac{u}{N_1}, \frac{j}{N_2}) - g_i(o', t')) \Psi_{u,j}(o', t') \right|$$

$$\le \sup_{o',t'} \sum_{u:|\frac{u_i}{N_1} - o'_i| \le \frac{2N_1}{3}, j:|\frac{j}{N_2} - t'| \le \frac{2N_2}{3}} \left| g_i(\frac{u}{N_1}, \frac{j}{N_2}) - g_i(o', t') \right| \Psi_{u,j}(o', t')$$

$$\le \frac{2\gamma_x}{3N_1} + \frac{2\gamma_t}{3N_2}. \tag{C.13}$$

Below we construct a ReLU neural network to approximate $\bar{g}_i$. Let $\sigma$ be ReLU activation and $r(a) = 2\sigma(a) - 4\sigma(a - 0.5) + 2\sigma(a - 1)$ for any scalar $a \in [0,1]$. Define

$$\phi_{\text{square}}^l(a) = a - \sum_{m=1}^{l} 2^{-2m} r_m(a), \ r_m = \underbrace{r \circ \cdots \circ r}_{m \text{ compositions}} \tag{C.14}$$

$$\phi_{\text{mul}}^l(a, b) = \phi_{\text{square}}^l(\frac{a+b}{2}) - \phi_{\text{square}}^l(\frac{a-b}{2}) \tag{C.15}$$

According to Yarotsky [2017],

$$|\phi_{\text{mul}}^l(a, b) - ab| \le 2^{-2l-2}, \ \forall a, b \in [0,1]. \tag{C.16}$$

Then we approximate $\Psi_{u,j}$ by recursively apply $\phi_{\text{mul}}^l$:

$$\widehat{\Psi}_{u,j}(x', w', t') := \phi_{\text{mul}}^l \big( \psi\big(3N_2(t' - j/N_2)\big), \phi_{\text{mul}}^l \big( \psi\big(3N_1(x'_1 - u_1/N_2)\big), \cdots \big) \big) \tag{C.17}$$

And we construct the final neural network approximation as

$$\widehat{g}_i(x', w', t') := \sum_{u,j} g_i(u/N_1, j/N_2) \widehat{\Psi}_{u,j}(x', w', t'). \tag{C.18}$$

The approximation error of $\widehat{g}_i$ can be bounded by

$$\|\widehat{g}_i - g_i\|_{L^\infty(\Omega)} \le \|\widehat{g}_i - \bar{g}_i\|_{L^\infty(\Omega)} + |\bar{g}_i - g_i\|_{L^\infty(\Omega)}$$

$$\le 2^{d_x + d_y + 1} \|g_i\|_{L^\infty(\Omega)} \sup_{u,j} \|\widehat{\Psi}_{u,j} - \Psi_{u,j}\|_{L^\infty(\Omega)} + \frac{2\gamma_x(d_x + d_y)^{\frac{1}{2}}}{3N_1} + \frac{2\gamma_t}{3N_2}$$

$$\le (d_x + d_y + 1)2^{d_x + d_y + 1} \|g_i\|_{L^\infty(\Omega)} 2^{-(2l+2)} + \frac{2\gamma_x(d_x + d_y)^{\frac{1}{2}}}{3N_1} + \frac{2\gamma_t}{3N_2}. \tag{C.19}$$

Besides, by Chen et al. [2020, Lemma 15], for $l \gtrsim d_x + d_y$ and $\forall x', w', w'', t'$,

$$|\widehat{g}_i(x', w', t') - \widehat{g}_i(x', w'', t')| \lesssim (d_x + d_y)\big(\gamma_w + N_1 \|g_i\|_{L^\infty(\Omega)} 2^{-l+d_x+d_y}\big) \|w' - w''\|_\infty. \tag{C.20}$$

Let $l = \mathcal{O}\left(d_x + d_y + \log \frac{\gamma_w(\|g\|_{L^\infty(\Omega)} + 1)}{\varepsilon}\right), N_1 = \mathcal{O}\left(\frac{\gamma_x}{\varepsilon}\right), N_2 = \mathcal{O}\left(\frac{\gamma_t}{\varepsilon}\right)$. Then

$$\|\widehat{g}_i - g_i\|_{L^\infty(\Omega)} \le \varepsilon/2, \ |\widehat{g}_i(x', w', t') - \widehat{g}_i(x', w'', t')| \lesssim \gamma_w(d_x + d_y)\|w' - w''\|_\infty. \tag{C.21}$$

Define $\widehat{g} := [\widehat{g}_1, \cdots, \widehat{g}_{d_x}]$ and $\widehat{f}(x, w, t) := \widehat{g}\left(\dfrac{x}{2R_f} + \dfrac{1}{2}, w, t/T\right)$. Then the approximation error of $\widehat{f}$ in $\Omega_{R_f}$ can be bounded by

$$\|\widehat{f} - f\|_{L^\infty(\Omega_{R_f})} \leq \|\widehat{g} - g\|_{L^\infty(\Omega)} \leq \sqrt{d_x}\,\varepsilon/2, \text{ and } \widehat{f}(x, w, t) = 0, \forall\, \|x\|_\infty > R_f. \qquad \text{(C.22)}$$

Therefore, when $R_f \geq C_R \log^{\frac{1}{2}}\left((M_f^2 + C_L)/\varepsilon\right)$, the overall approximation error is

$$
\begin{aligned}
\mathbb{E}_{(x,y)\sim\mathbb{P}}\mathbb{E}_{t,x_t|x}[\|f(x_t, h_*(y), t) - f_*^{\mathbb{P}}(x_t, h_*(y), t)\|^2] &\leq \mathbb{E}_{t,x_t,y}\|f(x_t, h(y), t) - f(x_t, h_*(y), t)\|^2 \cdot \mathbb{1}_{\|x_t\|\leq R_f} \\
&\quad + 4(M_f^2 + C_L)\exp(-C_1' R_f^2) \\
&\leq \|f - f_*^{\mathbb{P}}\|_{L^\infty(\Omega_{R_f})}^2 + 4(M_f^2 + C_L)\exp(-C_1' R_f^2) \\
&\leq d_x \varepsilon^2.
\end{aligned}
$$
$$\text{(C.23)}$$

Now we characterize the configuration of neural network $\widehat{f}(x, w, t)$. For boundedness, by Lemma B.10,

$$\|\widehat{f}(x, w, t)\| \leq \|f_*\|_{L^\infty(\Omega_{R_f})} + \varepsilon \leq 2C_X'' R_f^6 =: M_f. \qquad \text{(C.24)}$$

Hence we can let $R_f = \mathcal{O}\left(\log^{\frac{1}{2}}\left(\dfrac{nK}{\varepsilon\delta}\right)\right)$ to ensure the lower bound of $R_f$ mentioned above and in Theorem B.8. For Lipschitzness, by (C.21),

$$
\begin{aligned}
\|\widehat{f}(x, w, t) - \widehat{f}(x, \widetilde{w}, t)\| &\lesssim \gamma_w(d_x + d_y)\|w - \widetilde{w}\|_\infty \\
&\lesssim (C_X + C_X' R_f^2)(d_x + d_y)\|w - \widetilde{w}\|_\infty.
\end{aligned}
$$
$$\text{(C.25)}$$

Hence $\gamma_f = \mathcal{O}\left((C_X + C_X' R_f^2)(d_x + d_y)\right) = \mathcal{O}\left(\log\left(\dfrac{nK}{\varepsilon\delta}\right)\right)$.

For the size of neural network, for each coordinate, by the construction in (C.18), the neural network $\widehat{g}_i$ consists of $N_1^{d_x+d_y} N_2$ parallel subnetworks, i.e., $g_i(u/N_1, j/N_2)\widehat{\Psi}_{u,j}(\cdot, \cdot, \cdot)$. By definition in (C.17), the subnetwork consists of $\mathcal{O}\left((d_x + d_y)(d_x + d_y + \log\dfrac{R_f}{\varepsilon})\right)$ layers and the width is bounded by $\mathcal{O}(d_x + d_y)$. Therefore, the whole neural network $\widehat{g}_i$ can be implemented by $\mathcal{O}\left((d_x + d_y)(d_x + d_y + \log(R_f/\varepsilon))\right)$ layers with width $\mathcal{O}\left(N_1^{d_x+d_y} N_2(d_x + d_y)\right) = \mathcal{O}\left(\dfrac{R_f^{3(d_x+d_y)}}{\varepsilon^{d_x+d_y+1}T_0^3}\right)$, and the number of parameter is bounded by $\mathcal{O}\left(\dfrac{R_f^{3(d_x+d_y)}\log(R_f/\varepsilon)}{\varepsilon^{d_x+d_y+1}T_0^3}\right)$. Combine these arguments together, we can claim that the size of neural network $\widehat{f}$ is

$$
\begin{aligned}
L &= \mathcal{O}\left((d_x + d_y)(d_x + d_y + \log(R_f/\varepsilon))\right) = \mathcal{O}\left(\log\left(\dfrac{\log(nK/(\varepsilon\delta))}{\varepsilon}\right)\right), \\
W &= \mathcal{O}\left(\dfrac{R_f^{3(d_x+d_y)}}{\varepsilon^{d_x+d_y+1}T_0^3}\right) = \mathcal{O}\left(\dfrac{\log^{3(d_x+d_y)/2}(nK/(\varepsilon\delta))}{\varepsilon^{d_x+d_y+1}T_0^3}\right), \\
S &= \mathcal{O}\left(\dfrac{(d_x + d_y)R_f^{3(d_x+d_y)}\log(R_f/\varepsilon)}{\varepsilon^{d_x+d_y+1}T_0^3}\right) = \mathcal{O}\left(\dfrac{\log^{3(d_x+d_y)/2+1}(nK/(\varepsilon\delta))}{\varepsilon^{d_x+d_y+1}T_0^3}\right).
\end{aligned}
$$
$$\text{(C.26)}$$

To bound of the neural network parameters, note that the trapezoid function $\psi$ is rescaled by $3N_1$ or $3N_2$ and the weight parameter of $\phi_{\text{mul}}^l$ is bounded by a constant. Moreover, the input of $\widehat{f}$ is first rescaled by $R_f$ or $T$. Hence we have

$$B = \mathcal{O}\left(N_1 R_f + N_2 T\right) = \mathcal{O}\left(\dfrac{R_f^3 T}{\varepsilon}\right) = \mathcal{O}\left(\dfrac{T\log^{\frac{3}{2}}(nK/(\varepsilon\delta))}{\varepsilon}\right), \qquad \text{(C.27)}$$

which concludes the proof. $\qquad\qquad\square$

**Proposition C.3.** *To achieve*

$$\inf_{h \in \mathcal{H}} \|h - h_*\|_{L^\infty([0,1]^{D_y})} \leq \sqrt{d_y}\varepsilon, \tag{C.28}$$

*the configuration of* $\mathcal{H} = NN_h(L_h, W_h, S_h, B_h)$ *should satisfy*

$$\begin{aligned} L_h &= \mathcal{O}\left(\log(1/\varepsilon)\right), W_h = \mathcal{O}\left(\varepsilon^{-D_y}\log(1/\varepsilon)\right), \\ S_h &= \mathcal{O}\left(\varepsilon^{-D_y}\log^2(1/\varepsilon)\right), B_h = \mathcal{O}(1). \end{aligned} \tag{C.29}$$

*Here $\mathcal{O}(\cdot)$ hides all the polynomial factors of $d_x, d_y, L$.*

*Proof.* The main idea replicates Yarotsky [2017, Theorem 1]. We approximate each coordinate of $h_* = [h_{*1}, \cdots, h_{*d_y}]$ respectively and then concatenate all them together. By Yarotsky [2017, Theorem 1], $h_{*i}$ can be approximated up to $\varepsilon$ by a network $\widehat{h}_i$ with $\mathcal{O}\left(\log(1/\varepsilon)\right)$ layers and $\mathcal{O}\left(\varepsilon^{-D_y}\log(1/\varepsilon)\right)$ width. Besides, the range of all the parameters are bounded by some constant, and the number of parameters is $\mathcal{O}\left(\varepsilon^{-D_y}\log^2(1/\varepsilon)\right)$. Then we concatenate all the subnetworks to get $\widehat{h} = [\widehat{h}_1, \cdots, \widehat{h}_{d_y}]$ and $\|\widehat{h} - h_*\|_{L^\infty([0,1]^{D_y})} \leq \sqrt{d_y}\varepsilon$. $\qquad\square$

## C.2   Proofs of Distribution Estimation

**Theorem C.4** (Thm. 4.2). *Suppose Assumption 3.1, 3.2, 3.3 hold. For sufficiently large integers $n, K, m$ and $\delta > 0$, further suppose that $\mathbb{P}^1, \cdots, \mathbb{P}^K$ are $(\nu, \Delta)$-diverse over target distribution $\mathbb{P}^0$ with proper configuration of neural network family and $T, T_0$. It holds that with probability no less than $1 - \delta$,*

$$\mathbb{E}_{\{(x_i,y_i)\}_{i=1}^m}\mathbb{E}_{y \sim \mathbb{P}_y^0}[\mathrm{TV}(\widehat{\mathbb{P}}_{x|y}^0, \mathbb{P}_{x|y}^0)] \lesssim \frac{\log^{\frac{5}{2}}(nK/\delta)\log^3((m/\nu) \wedge n)}{\nu^{\frac{1}{2}}((m/\nu) \wedge n)^{\frac{1}{d_x+d_y+9}}} + \frac{\log^2(nK/\delta)}{\nu^{\frac{1}{2}}(nK)^{\frac{1}{D_y+2}}} + \sqrt{\Delta}. \tag{C.30}$$

*Proof.* Combine Theorem C.1 and Theorem B.6 and plug in the configuration of $\mathcal{F}, \mathcal{H}$, we have with probability no less than $1 - \delta$

$$\mathbb{E}_{\{(x_i,y_i)\}_{i=1}^m}\mathbb{E}_{(x,y) \sim \mathbb{P}^0}[\ell^{\mathbb{P}^0}(x, y, s_{\widehat{f}^{\mathbb{P}^0},\widehat{h}})]$$

$$\lesssim \frac{1}{\nu}\log^2(nK/(\varepsilon\delta))\varepsilon^2 + \Delta + \frac{\log^{\frac{3(d_x+d_y)+15}{2}}(nK/\varepsilon\delta)\log(T/T_0)}{(m \wedge (\nu n))\varepsilon^{d_x+d_y+1}T_0^3} + \frac{\log^4(1/\varepsilon)\log(nK/(\varepsilon\delta))}{\nu nK\varepsilon^{D_y}} \tag{C.31}$$

By Lemma C.7,

$$\mathrm{TV}(\widehat{\mathbb{P}}_{x|y}^0, \mathbb{P}_{x|y}^0) \lesssim \sqrt{T_0}\log^{\frac{d_x+1}{2}}(1/T_0) + e^{-T} + \sqrt{\mathbb{E}_{\mathbb{P}_{x|y}^0}[\ell^{\mathbb{P}^0}(x, y, s_{\widehat{f}^{\mathbb{P}^0},\widehat{h}})]} \tag{C.32}$$

Taking expectation of $y, \widehat{f}^{\mathbb{P}}, \mathbb{P}$, we have

$$\mathbb{E}_{\{(x_i,y_i)\}_{i=1}^m}\mathbb{E}_{y \sim \mathbb{P}_y^0}[\mathrm{TV}(\widehat{\mathbb{P}}_{x|y}^0, \mathbb{P}_{x|y}^0)] \lesssim \sqrt{T_0}\log^{\frac{d_x+1}{2}}(1/T_0) + e^{-T} + \nu^{-\frac{1}{2}}\log(nK/(\varepsilon\delta))\varepsilon + \sqrt{\Delta}$$

$$+ \frac{\log^{\frac{3(d_x+d_y)+15}{4}}(\frac{nK}{\varepsilon\delta})\log^{\frac{1}{2}}(\frac{T}{T_0})}{(m \wedge (\nu n))^{\frac{1}{2}}\varepsilon^{\frac{d_x+d_y+1}{2}}T_0^{\frac{3}{2}}} + \frac{\log^2(\frac{1}{\varepsilon})\log^{\frac{1}{2}}(\frac{nK}{\varepsilon\delta})}{(\nu nK)^{\frac{1}{2}}\varepsilon^{\frac{D_y}{2}}}. \tag{C.33}$$

Let $T_0 = \mathcal{O}\left(\varepsilon_0^2/\log^{d_x+1}(1/\varepsilon_0)\right), T = \mathcal{O}(\log(1/\varepsilon_0)), \varepsilon = \mathcal{O}(\varepsilon_0/\log(nK/(\varepsilon_0\delta_0)))$ for some small $\varepsilon_0 > 0$ defined later. Then it reduces to

$$\mathbb{E}_{\{(x_i,y_i)\}_{i=1}^m}\mathbb{E}_{y \sim \mathbb{P}_y^0}[\mathrm{TV}(\widehat{\mathbb{P}}_{x|y}^0, \mathbb{P}_{x|y}^0)] \lesssim \frac{\varepsilon_0}{\nu^{\frac{1}{2}}} + \sqrt{\Delta} + \frac{\log^{\frac{5(d_x+d_y)+17}{4}}(\frac{nK}{\varepsilon_0\delta})\log^{\frac{3d_x+5}{2}}(\frac{1}{\varepsilon_0})}{(m \wedge (\nu n))^{\frac{1}{2}}\varepsilon_0^{\frac{d_x+d_y+7}{2}}} + \frac{\log^2(\frac{1}{\varepsilon_0})\log^{D_y+\frac{1}{2}}(\frac{nK}{\varepsilon_0\delta})}{(\nu nK)^{\frac{1}{2}}\varepsilon_0^{\frac{D_y}{2}}}. \tag{C.34}$$

Let $\varepsilon_0 = C\max\left\{\dfrac{\log^{\frac{5}{2}}(nK/\delta)\log^3((m/\nu)\wedge n)}{((m/\nu)\wedge n)^{\frac{1}{d_x+d_y+9}}}, \dfrac{\log^2(nK/\delta)}{(nK)^{\frac{1}{D_y+2}}}\right\}$, and we can conclude that

$$\mathbb{E}_{\{(x_i,y_i)\}_{i=1}^m}\mathbb{E}_{y\sim\mathbb{P}_y^0}[\text{TV}(\widehat{\mathbb{P}}_{x|y}^0,\mathbb{P}_{x|y}^0)]\lesssim\frac{\log^{\frac{5}{2}}(nK/\delta)\log^3((m/\nu)\wedge n)}{\nu^{\frac{1}{2}}((m/\nu)\wedge n)^{\frac{1}{d_x+d_y+9}}}+\frac{\log^2(nK/\delta)}{\nu^{\frac{1}{2}}(nK)^{\frac{1}{D_y+2}}}+\sqrt{\Delta}.$$
(C.35)
$\square$

**Theorem C.5** (Thm. 4.3). *Suppose Assumption 3.1, 3.2, 3.3 hold. For sufficiently large integers $n, K, m$ and $\delta > 0$, with proper configuration of neural network family and $T, T_0$, it holds that with probability no less than $1 - \delta$,*

$$\mathbb{E}_{\mathbb{P}\sim\mathbb{P}_{meta}}\mathbb{E}_{\{(x_i,y_i)\}_{i=1}^m\sim\mathbb{P}}\mathbb{E}_{y\sim\mathbb{P}_y}[\text{TV}(\widehat{\mathbb{P}}_{x|y},\mathbb{P}_{x|y})]\lesssim\frac{\log^{\frac{5}{2}}(nK/\delta)\log^3(m\wedge n)}{(m\wedge n)^{\frac{1}{d_x+d_y+9}}}+\frac{\log^2(nK/\delta)}{K^{\frac{1}{D_y+2}}}.$$
(C.36)

*Proof.* Combine Theorem C.1 and Theorem B.8 and plug in the configuration of $\mathcal{F}, \mathcal{H}$, we have with probability no less than $1 - \delta$

$$\mathbb{E}_{\mathbb{P}\sim\mathbb{P}_{meta}}\mathbb{E}_{\{(x_i,y_i)\}_{i=1}^m\sim\mathbb{P}}\mathbb{E}_{(x,y)\sim\mathbb{P}}[\ell^{\mathbb{P}}(x,y,s_{\widehat{f}^{\mathbb{P}},\widehat{h}})]$$
$$\lesssim\log^2(nK/(\varepsilon\delta))\varepsilon^2+\frac{\log^{\frac{3(d_x+d_y)+15}{2}}(nK/\varepsilon\delta)\log(T/T_0)}{(m\wedge n)\varepsilon^{d_x+d_y+1}T_0^3}+\frac{\log^4(1/\varepsilon)\log(nK/(\varepsilon\delta))}{K\varepsilon^{D_y}}$$
(C.37)

By Lemma C.7,

$$\text{TV}(\widehat{\mathbb{P}}_{x|y},\mathbb{P}_{x|y})\lesssim\sqrt{T_0}\log^{\frac{d_x+1}{2}}(1/T_0)+e^{-T}+\sqrt{\mathbb{E}_{\mathbb{P}_{x|y}}[\ell^{\mathbb{P}}(x,y,s_{\widehat{f}^{\mathbb{P}},\widehat{h}})]}$$
(C.38)

Taking expectation of $y, \widehat{f}^{\mathbb{P}}, \mathbb{P}$, we have

$$\mathbb{E}_{\mathbb{P}\sim\mathbb{P}_{meta}}\mathbb{E}_{\{(x_i,y_i)\}_{i=1}^m\sim\mathbb{P}}\mathbb{E}_{y\sim\mathbb{P}_y}[\text{TV}(\widehat{\mathbb{P}}_{x|y},\mathbb{P}_{x|y})]\lesssim\sqrt{T_0}\log^{\frac{d_x+1}{2}}(1/T_0)+e^{-T}+\log(nK/(\varepsilon\delta))\varepsilon$$
$$+\frac{\log^{\frac{3(d_x+d_y)+15}{4}}\left(\frac{nK}{\varepsilon\delta}\right)\log^{\frac{1}{2}}\left(\frac{T}{T_0}\right)}{(m\wedge n)^{\frac{1}{2}}\varepsilon^{\frac{d_x+d_y+1}{2}}T_0^{\frac{3}{2}}}+\frac{\log^2\left(\frac{1}{\varepsilon}\right)\log^{\frac{1}{2}}\left(\frac{nK}{\varepsilon\delta}\right)}{K^{\frac{1}{2}}\varepsilon^{\frac{D_y}{2}}}.$$
(C.39)

Let $T_0 = \mathcal{O}\left(\varepsilon_0^2/\log^{d_x+1}(1/\varepsilon_0)\right), T = \mathcal{O}(\log(1/\varepsilon_0)), \varepsilon = \mathcal{O}(\varepsilon_0/\log(nK/(\varepsilon_0\delta_0)))$ for some small $\varepsilon_0 > 0$ defined later. Then it reduces to

$$\mathbb{E}_{\mathbb{P}\sim\mathbb{P}_{meta}}\mathbb{E}_{\{(x_i,y_i)\}_{i=1}^m\sim\mathbb{P}}\mathbb{E}_{y\sim\mathbb{P}_y}[\text{TV}(\widehat{\mathbb{P}}_{x|y},\mathbb{P}_{x|y})]\lesssim\varepsilon_0+\frac{\log^{\frac{5(d_x+d_y)+17}{4}}\left(\frac{nK}{\varepsilon_0\delta}\right)\log^{\frac{3d_x+5}{2}}\left(\frac{1}{\varepsilon_0}\right)}{(m\wedge n)^{\frac{1}{2}}\varepsilon_0^{\frac{d_x+d_y+7}{2}}}$$
$$+\frac{\log^2\left(\frac{1}{\varepsilon_0}\right)\log^{D_y+\frac{1}{2}}\left(\frac{nK}{\varepsilon_0\delta}\right)}{K^{\frac{1}{2}}\varepsilon_0^{\frac{D_y}{2}}}.$$
(C.40)

Let $\varepsilon_0 = C\max\left\{\dfrac{\log^{\frac{5}{2}}(nK/\delta)\log^3(m\wedge n)}{(m\wedge n)^{\frac{1}{d_x+d_y+9}}}, \dfrac{\log^2(nK/\delta)}{K^{\frac{1}{D_y+2}}}\right\}$, and we can conclude that

$$\mathbb{E}_{\mathbb{P}\sim\mathbb{P}_{meta}}\mathbb{E}_{\{(x_i,y_i)\}_{i=1}^m\sim\mathbb{P}}\mathbb{E}_{y\sim\mathbb{P}_y}[\text{TV}(\widehat{\mathbb{P}}_{x|y},\mathbb{P}_{x|y})]\lesssim\frac{\log^{\frac{5}{2}}(nK/\delta)\log^3(m\wedge n)}{(m\wedge n)^{\frac{1}{d_x+d_y+9}}}+\frac{\log^2(nK/\delta)}{K^{\frac{1}{D_y+2}}}.$$
(C.41)
$\square$

## C.3 Auxiliary Lemmas

**Lemma C.6.** *Let $\Omega_{R_f} = [-R_f, R_f]^{d_x} \times [0,1]^{d_y} \times [T_0, T]$ for some $R_f \geq 1$. Then there exists some constant $C_s$, such that the score function $f_*^{\mathbb{P}}(x, w, t)$ is $\dfrac{C_s R_f^3}{T_0^3}$-Lipschitz with respect to $t$ in $\Omega_{R_f}$.*

*Proof.* According to (B.2),

$$f_*^{\mathbb{P}}(x, w, t) = -\frac{x}{\sigma_t^2} + \frac{\alpha_t}{\sigma_t^2} \int x_0 \frac{\phi_t(x|x_0)p(x_0; w)}{\int \phi_t(x|z)p(z; w)\mathrm{d}z}\mathrm{d}x_0. \tag{C.42}$$

Define density function $q_t(x_0|x, w) \propto \phi_t(x|x_0)p(x_0; w)$. Then

$$\frac{\partial}{\partial t} f_*^{\mathbb{P}}(x, w, t) = -\frac{2\alpha_t^2 x}{\sigma_t^2} + \frac{\alpha_t}{\sigma_t^2}\mathrm{Cov}_{q_t(x_0|x,w)}\left(x_0, \frac{\partial}{\partial t}\log\phi_t(x|x_0)\right) - \frac{\alpha_t(1 + \alpha_t^2)}{\sigma_t^4}\mathbb{E}_{q_t(x_0|x,w)}[x_0]. \tag{C.43}$$

Note that

$$\mathrm{Cov}_{q_t(x_0|x,w)}\left(x_0, \frac{\partial}{\partial t}\log\phi_t(x|x_0)\right) = -\mathrm{Cov}_{q_t(x_0|x,w)}\left(x_0, \frac{\partial}{\partial t}\frac{\|x - \alpha_t x_0\|^2}{2\sigma_t^2}\right)$$

$$= \mathrm{Cov}_{q_t(x_0|x,w)}\left(x_0, \frac{\alpha_t(x - \alpha_t x_0)^\top \mathbf{1}}{\sigma_t^2} - \frac{2\alpha_t^2\|x - \alpha_t x_0\|^2}{\sigma_t^4}\right) \tag{C.44}$$

Hence for any $x \in [-R_f, R_f]^{d_x}, w \in [0,1]^{d_y}$,

$$\left\|\frac{\partial}{\partial t}f_*^{\mathbb{P}}(x, w, t)\right\|_\infty \lesssim \frac{\alpha_t^2 R_f}{\sigma_t^2} + \mathbb{E}_{q_t(x_0|x,w)}\left\|\frac{x - \alpha_t x_0}{\sigma_t^2}\right\|^3 + \frac{\alpha_t(1 + \alpha_t^2)}{\sigma_t^4}\mathbb{E}_{q_t(x_0|x,w)}[\|x_0\|_\infty] \tag{C.45}$$

Let $R = \dfrac{2R_f + 2C_0}{\sigma_t}$. We have

$$\mathbb{E}_{q_t(x_0|x,w)}\left\|\frac{\alpha_t x_0 - x}{\sigma_t^2}\right\|^3 \preceq \frac{1}{\sigma_t^3}\int\left\|\frac{\alpha_t x_0 - x}{\sigma_t}\right\|^3 \frac{\phi_t(x|x_0)p(x_0|y)}{\int\phi_t(x|z)p(z|y)\mathrm{d}z}\mathrm{d}x_0$$

$$\leq \frac{R^3}{\sigma_t^3} + \frac{\int_{\|\frac{\alpha_t x_0 - x}{\sigma_t}\| \geq R}\|\frac{\alpha_t x_0 - x}{\sigma_t}\|^2\exp\left(-\frac{\|\alpha_t x_0 - x\|^2}{2\sigma_t^2}\right)p(x_0; w)\mathrm{d}x_0}{\sigma_t^3\int\exp\left(-\frac{\|\alpha_t x_0 - x\|^2}{2\sigma_t^2}\right)p(x_0; w)\mathrm{d}x_0}$$

$$\leq \frac{R^3}{\sigma_t^3} + \frac{\int_{\|\frac{\alpha_t x_0 - x}{\sigma_t}\| \geq R}\exp(-\frac{R^2}{4})p(x_0; w)\mathrm{d}x_0}{\sigma_t^3\int_{\|\frac{\alpha_t x_0 - x}{\sigma_t}\| \leq R/2}\exp(-\frac{R^2}{8})p(x_0; w)\mathrm{d}x_0}. \tag{C.46}$$

The domain $\left\{x_0 : \|\frac{\alpha_t x_0 - x}{\sigma_t}\| \leq R/2\right\}$ includes $\left\{x_0 : \|x_0\| \leq C_0\right\}$, indicating

$$\int_{\|\frac{\alpha_t x_0 - x}{\sigma_t}\| \leq R/2} p(x_0; w)\mathrm{d}x_0 \geq \int_{\|x_0\| \leq C_0} p(x_0; w)\mathrm{d}x_0 \geq 1 - 2\exp(-C_1'C_0^2) \geq \frac{1}{2},$$

$$\int_{\|\frac{\alpha_t x_0 - x}{\sigma_t}\| \geq R} p(x_0; w)\mathrm{d}x_0 \leq \int_{\|x_0\| \geq C_0} p(x_0; w)\mathrm{d}x_0 \leq \frac{1}{2}. \tag{C.47}$$

Therefore, for any $(x, w, t) \in \Omega_{R_f}$,

$$\left\|\frac{\partial}{\partial t}f_*^{\mathbb{P}}(x, w, t)\right\|_\infty \lesssim \frac{R_f^2}{\sigma_t^2} + \frac{R_f^3 + C_0^3}{\sigma_t^6} + \frac{R_f + C_0}{\sigma_t^3} \lesssim \frac{R_f^3}{T_0^3}. \tag{C.48}$$

$\square$

**Lemma C.7.** *Suppose $\mathrm{KL}(\mathbb{P}_{x|y}^0\|\mathcal{N}(0, I)) \leq C_{\mathrm{KL}}$ for some constant $C_{\mathrm{KL}}$. Then*

$$\mathrm{TV}(\widehat{\mathbb{P}}_{x|y}^0, \mathbb{P}_{x|y}^0) \lesssim \sqrt{T_0}\log^{\frac{d_x+1}{2}}(1/T_0) + e^{-T} + \sqrt{\mathbb{E}_{\mathbb{P}_{x|y}^0}[\ell^{\mathbb{P}^0}(x, y, s_{\widehat{f},\widehat{h}})]}. \tag{C.49}$$

*Proof.* With a little abuse of notation, we will use $p_t(x_t|y)$ to denote the conditional density of $x_t|y$ under $\mathbb{P}^0_{x|y}$. Consider the following two backward processes

$$d\widetilde{x}_t = (\widetilde{x}_t + 2\nabla \log p_{T-t}(\widetilde{x}_t|y))dt + \sqrt{2}dW_t, \ \widetilde{x}_0 \sim \mathcal{N}(0,I), 0 \leq t \leq T - T_0, \quad (C.50)$$

$$d\bar{x}_t = (\bar{x}_t + 2\nabla \log p_{T-t}(\widetilde{x}_t|y))dt + \sqrt{2}dW_t, \ \bar{x}_0 \sim p_T, 0 \leq t \leq T - T_0. \quad (C.51)$$

Denote the distribution of $\widetilde{x}_t$ as $\widetilde{\mathbb{P}}_{T-t}$. And note that $\bar{x}_t \sim p_{T-t}$ by classic reverse-time SDE results [Anderson, 1982]. Then by Fu et al. [2024, Lemma D.5],

$$\mathrm{TV}(\mathbb{P}_{T_0}, \mathbb{P}_0) \lesssim \sqrt{T_0} \log^{\frac{d_x+1}{2}}(1/T_0). \quad (C.52)$$

At the same time, we apply Data Processing inequality and Pinsker's inequality to get

$$\mathrm{TV}(\mathbb{P}_{T_0}, \widetilde{\mathbb{P}}_{T_0}) \leq \mathrm{TV}(\mathbb{P}_T, \mathcal{N}(0,I)) \lesssim \sqrt{\mathrm{KL}(\mathbb{P}_T\|\mathcal{N}(0,I))} \lesssim \sqrt{\mathrm{KL}(\mathbb{P}_0\|\mathcal{N}(0,I))}e^{-T}. \quad (C.53)$$

Again according to Pinsker's inequality and Oko et al. [2023, Proposition D.1],

$$\mathrm{TV}(\widehat{\mathbb{P}}, \widetilde{\mathbb{P}}_{T_0}) \lesssim \sqrt{\mathrm{KL}(\widetilde{\mathbb{P}}_{T_0}\|\widehat{\mathbb{P}})} \lesssim \sqrt{\mathbb{E}_{x|y}[\ell^{\mathbb{P}}(x, y, s_{\widehat{f},\widehat{h}})]}. \quad (C.54)$$

Combine three inequalities above and we complete the proof. $\qquad\square$

## D  Proofs in Section A

### D.1  Proof of Theorem A.1

*Proof.* Due to the structure of exponential family, Assumption 3.2 holds obviously. To apply previous results, we only need to verify Assumption 3.1 and 3.3. Recall that a basic property of exponential family is

$$\nabla_x A_\psi(x) = \mathbb{E}_{p_\psi(y|x)}[h_*(y)] \in [0,1]^d, \quad (D.1)$$

$$0 \preceq \nabla_x^2 A_\psi(x) = \mathrm{Var}_{p_\psi(y|x)}(h_*(y)) \preceq I. \quad (D.2)$$

Hence by Assumption A.1, $A_\psi(x) \leq A_\psi(0) + \|x\|_1 \leq \log\left(\int \psi(y)\mathrm{d}y\right) + \|x\|_1 \leq \log C + \|x\|_1$. And $A_\psi(x) \geq A_\psi(0) - \|x\|_1 \geq -\log C - \|x\|_1$. Further note that the posterior density $p_\theta(x|y) = \dfrac{p_\phi(x)\exp(\langle x, h_*(y)\rangle - A_\psi(x))}{Z_\theta}$, where the normalizing constant $Z_\theta(y)$ is lower bounded by

$$
\begin{aligned}
Z_\theta(y) &= \int p_\phi(x)\exp(\langle x, h_*(y)\rangle - A_\psi(x))\mathrm{d}x \\
&\geq \int p_\phi(x)\exp(-2\|x\|_1)/C\,\mathrm{d}x \\
&\geq \exp(-2\sqrt{d}R)(1 - 2\exp(-C_1'R^2))/C =: C_0.
\end{aligned}
\quad (D.3)
$$

where in the second inequality we apply $\mathbb{P}_\phi(\|x\| \geq R) \leq 2\exp(-C_1'R^2)$ and let $R = 1/\sqrt{C_1'}$ to get $C_0$. Therefore, by Assumption A.1,

$$p_\theta(x|y) \leq C_1 \exp(-C_2\|x\|^2 + 2\|x\|_1 + \log C)/C_0 \leq C_1'\exp(-C_2'\|x\|^2), \quad (D.4)$$

and thus Assumption 3.1 holds. At the same time, ley $w = h_*(y)$, then the score function is

$$\nabla_x \log p_\theta(x|y) = \nabla_x \log p_\theta(x,w) = \nabla_x \log p_\phi(x) + w - \nabla_x A_\psi(x). \quad (D.5)$$

Since $\nabla_x \log p_\phi(x)$ is $L$-Lipschitz, $\nabla A_\psi(x)$ is also 1-Lipschitz, the score function $\nabla_x \log p_\theta(x,w)$ is $(L+1)$-Lipschitz in $x$ and 1-Lipschitz in $w$. And $\|\nabla_x \log p_\theta(0,w)\| \leq \|\nabla_x \log p_\phi(0)\| + 2\sqrt{d} = B + 2\sqrt{d}$, indicating that Assumption 3.3 holds with $L' = L + 1, B' = B + 2\sqrt{d}$.

We conclude the proof by applying Theorem 4.3 under meta-learning setting or Theorem 4.2 under $(\nu, \Delta)$-diversity. $\qquad\square$

## D.2 Proof of Theorem A.2

*Proof.* Let $A_M^\pi(s,a) = Q_M^\pi(s,a) - V_M(\pi,s)$ be the advantage function of policy $\pi$. Note that the reward function $r_M \in [0,1]$, we have $|A_M^\pi(s,a)| \leq \dfrac{2}{1-\gamma}$ for any $M, \pi$. According to performance difference lemma,

$$
\begin{aligned}
V_{M^0}(\pi_*^0) - V_{M^0}(\widehat{\pi}^0) &= \frac{1}{1-\gamma} \mathbb{E}_{(s,a)\sim d_*^0}[A_{M^0}^{\widehat{\pi}^0}(s,a)] \\
&= \frac{1}{1-\gamma} \mathbb{E}_{s\sim d_*^0}\left[\mathbb{E}_{a\sim\pi_*^0(\cdot|s)}[A_{M^0}^{\widehat{\pi}^0}(s,a)] - \mathbb{E}_{a\sim\widehat{\pi}^0(\cdot|s)}[A_{M^0}^{\widehat{\pi}^0}(s,a)]\right] \quad \text{(D.6)} \\
&\leq \frac{2}{(1-\gamma)^2} \mathbb{E}_{s\sim d_*^0}[\text{TV}(\pi_*^0(\cdot|s),\widehat{\pi}^0(\cdot|s))].
\end{aligned}
$$

Hence in meta-learning setting, we plug in Theorem 4.3 to obtain

$$
\mathbb{E}_{M^0}\mathbb{E}_{\{(s_i^0,a_i^0)\}_{i=1}^m\sim d_*^0}[V_{M^0}(\pi_*^0)-V_{M^0}(\widehat{\pi}^0)] \lesssim \frac{1}{(1-\gamma)^2}\left[\frac{\log^{\frac{5}{2}}(nK/\delta)\log^3(m\wedge n)}{(m\wedge n)^{\frac{1}{d_a+d_s+9}}} + \frac{\log^2(nK/\delta)}{K^{\frac{1}{\mathcal{D}_s+2}}}\right].
$$
(D.7)

If we further assume $(\nu, \Delta)$-diversity holds, then we plug in Theorem 4.2,

$$
\mathbb{E}_{\{(s_i^0,a_i^0)\}_{i=1}^m\sim d_*^0}[V_{M^0}(\pi_*^0)-V_{M^0}(\widehat{\pi}^0)] \lesssim \frac{1}{(1-\gamma)^2}\left[\frac{\log^{\frac{5}{2}}(nK/\delta)\log^3((m/\nu)\wedge n)}{\nu^{\frac{1}{2}}((m/\nu)\wedge n)^{\frac{1}{d_a+d_s+9}}} + \frac{\log^2(nK/\delta)}{\nu^{\frac{1}{2}}(nK)^{\frac{1}{\mathcal{D}_s+2}}} + \sqrt{\Delta}\right].
$$
(D.8)

$\square$

# E  Experiment Details

## E.1  Conditioned Diffusion

Each $f^k$ and $f^0$ are implemented as a 2-layer MLP with 128 internal channels and 60 input channels. The representation map $h$ is implemented as a 5-layer MLP with 512 internal channels and 10 output channels. We have $n = 1000$ pre-training samples from each source distribution $\mathbb{P}^k$, $m \in \{10, 20, 30, 40, 50, 100\}$ fine-tuning samples from the target distribution $\mathbb{P}^0$. We run Langevin Monte Carlo for sufficiently long time to obtain 100 test samples from the target distribution $\mathbb{P}^0$ for evaluating the test error of different models. In the pre-training phase, the $\{\widehat{f}^k; 1 \leq k \leq K\}$ and $\hat{h}$ are trained on the $K = 10$ source distributions with 400K iterations and a batch size of 512. In the fine-tuning phase, the pre-trained representation map $\widehat{h}$ is fixed, and the $\widehat{f}^0$ is trained on the target distribution with 200K iterations and a batch size of $m$. As an important baseline, we also consider jointly training $h$ and $f^0$ on the target distribution from scratch, using the same fine-tuning samples.

## E.2  Image Restoration on MNIST

Each $f^k$ and $f^0$ are implemented as a 3-layer MLP with 512 internal channels and 784 input channels. The representation map $h$ is implemented as a 5-layer MLP with 512 internal channels and 64 output channels. We have $n = 5000$ pre-training samples from each source distribution $\mathbb{P}^k$, and $m \in \{10, 20, 30, 40, 50, 100\}$ fine-tuning samples from the target distribution $\mathbb{P}^0$. For evaluation, we directly compute the mean squared error between the posterior samples and the ground truth images, based on 100 test samples from $\mathbb{P}^0$. In the pre-training phase, the the $\{\widehat{f}^k; 1 \leq k \leq K = 9\}$ and $\hat{h}$ are 2K epochs and a batch size of 512. The initial learning rate is 0.0003 and is annealed according to a cosine annealing schedule. In the fine-tuning phase, the pre-trained representation map $\widehat{h}$ is fixed, and the $\widehat{f}^0$ is trained on the target distribution with 20K iterations and a batch size of $m$. As an important baseline, we also consider jointly training $h$ and $f^0$ on the target distribution from scratch, using the same fine-tuning samples.

