# OpenReview forum: "Provable Sample-Efficient Transfer Learning Conditional Diffusion Models via Representation Learning"
_NeurIPS.cc/2025/Conference — NeurIPS 2025 poster_

### Official Review · Reviewer_8krS · 2025-06-18

**Clarity:** 3
**Significance:** 2
**Originality:** 3
**Rating:** 4
**Confidence:** 4

**Summary:**

This paper provides rigorous statistical analysis on how transfer learning reduces the sample complexity for training diffusion models. The main idea is to assume distributions of different conditions share certain common information, modeled as $h(y)$. Intuitively, finetuning phase saves the computational cost of learning $h(\cdot)$, as it is assumed such optimal latent representation is already learnt during pretraining.

**Questions:**

1. How tight is the bound in (3.7) and (3.8)? How large is $C_P$?

2. Can you verify the task diversity assumption in practice? What is the relationship between $N_{F}$ and $N_{H}$? What if $N_{F}$ dominates?

**Ethical Concerns:**

["NO or VERY MINOR ethics concerns only"]

**Final Justification:**

This is a theoretical paper and the contents are rigorous, having no obvious flaw. Hence I think it could be accepted. The limitation is that some conclusions are hard to be validated on real world practical models.

**Limitations:**

1. The core assumption, task diversity is strong and not verified. The relationship between $N_{F}$ and $N_{H}$ is unclear.

2. The task-diversity assumption already bears most of the theoretical burden; although the subsequent theorems supply extra bounds on sample diversity, the insights they yield are, in my view, predictable and add little novelty.

3. The inequalities in (3.7) and (3.8) need further discussion. It is not clear how tight they are.

**Quality:**

3

**Strengths And Weaknesses:**

Strength:

1. The paper is well written and easy to follow.

2. The study is comprehensive, considered several settings, such as with or without task diversity assumption. The approximation error brought by the deep network is also analyzed.

Weakness:

1. The core assumption, i.e., task diversity, assumption is quite strong and is not empirically verified. Moreover, the theorem in section 3.2 seems a little bit trivial given the task diversity assumption, as this assumption alone already implies pretraining bounds the error of finetuning phase.

2. The bound in (3.7) and (3.8) needs further discussion. How tight are they? How large is $C_P$?

3. The relationship between $N_{F}$ and $N_{H}$ is not clear. Currently the improved sample complexity comes from $N_{H}$. But what if $N_{F}$>>$N_{H}$, hence dominate the loss? Can you somehow quantify their relationship empirically on real-world dataset such as natural images?

---

> ### Author Rebuttal · Authors · 2025-07-31
>
> Thanks for your careful review and constructive feedback! We address your concerns as follows.
>
> **W1**: The task diversity, assumption is quite strong and is not empirically verified. Moreover, the theorem in section 3.2 seems a little bit trivial given the task diversity assumption, as this assumption alone already implies pretraining bounds the error of finetuning phase.
>
> **A1**: The task diversity is a standard assumption in theoretical studies of transfer learning [1,2,3], which establishes the connection between source tasks and target tasks. Thm 3.4 is indeed a straightforward corollary based on Prop 3.2-3.3 and task diversity. However, Prop 3.2-3.3 is definitely non-trivial and requires novel analysis methods for pretraining and finetuning in CDMs. In addition, we provide some naive analysis to verify the task diversity assumption in Appendix B.5. Conducting a more fine-grained analysis is an interesting future work.
>
> **W2**: The bound in (3.7) and (3.8) needs further discussion. How tight are they? How large is $C_P$?
>
> **A2**: The bound in Eq (3.7) and (3.8) can be interpreted as generalization bounds for *empirical risk minimization over samples of tasks*. Intuitively, standard generalization results based on (local) Rademacher complexity [4] should apply. In fact, one of our technical contributions is to extend the framework in [4] to general *samples* as formalized in Lemma B.11. In addition, note that Eq (3.7), (3.8) are key lemmas for Thm 3.6, which provides the SOTA sample complexity for meta-learning as $\mathcal{O}(\frac{1}{m}+\frac{1}{K})$ dependence when $n$ is large. In contrast, existing work [3; Thm 2] can only achieve $\mathcal{O}(\frac{1}{\sqrt{m}}+\frac{1}{\sqrt{K}})$ for supervised meta-learning due to less sharp techniques.
>
> Regarding $C_P$, it is a constant with polynomial dependence on the parameters in Assumptions 3.1-3.3, and its definition can be found in the proofs in Appendix B.3.
>
> **W3**: Currently the improved sample complexity comes from $N_\mathcal{H}$. But what if $N_\mathcal{F}\gg N_\mathcal{H}$, hence dominate the loss? Can you quantify their relationship empirically on real-world dataset?
>
> **A3**: In this work, we focus on the setting when the conditions are of high dimension and the condition encoder in CDM is harder to learn than the backbone score network, i.e.,  $N_\mathcal{H}>N_\mathcal{F}$. In Table 3 in Appendix A, we list some real world applications of CDMs including text-to-image, reinforcement learning, etc, where the number of parameters of condition encoder is larger than or at least comparable to that of backbone score network, indicating that our setting is aligned with many real world cases. Of course there are also examples such that the class of condition encoder is very small and $N_\mathcal{F}\gg N_\mathcal{H}$. To understand sample efficiency of transfer learning in such settings is beyond the scope of this paper.
>
> Thanks for your valuable feedback and we will add a section for limitation and discussion in the final version.
>
>
> [1] Tripuraneni, Nilesh, Michael Jordan, and Chi Jin. "On the theory of transfer learning: The importance of task diversity." Advances in neural information processing systems 33 (2020): 7852-7862.
>
> [2] Chua, Kurtland, Qi Lei, and Jason D. Lee. "How fine-tuning allows for effective meta-learning." Advances in Neural Information Processing Systems 34 (2021): 8871-8884.
>
> [3] Maurer, Andreas, Massimiliano Pontil, and Bernardino Romera-Paredes. "The benefit of multitask representation learning." Journal of Machine Learning Research 17.81 (2016): 1-32.
>
> [4] Bartlett, Peter L., Olivier Bousquet, and Shahar Mendelson. "Local rademacher complexities." (2005): 1497-1537.

---

> > ### Comment · Reviewer_8krS · 2025-08-02
> >
> > Thanks for the extra information. Most of my concerns have been addressed, and I am now in favor of accepting this paper. But I also want to point out a minor issue regarding A.3: I don't think the number of parameters of the conditional encoder in real-world model is equivalent to $N_H$, unless you can somehow calculate the least number of parameters necessary for the encoder to learn conditional information.

---

> > > ### Author Response · Authors · 2025-08-03
> > >
> > > Thanks for your feedback and we are glad to address most concerns. As for A.3, we agree that the number of parameters is not equivalent to the covering number (complexity) $N_H$ but it has some positive relations. In fact, for a fixed type of model families like MLPs or transformers, the complexity typically increases as the number of parameters increases.
> > >
> > > Please let us know if there is any further question.

---

### Official Review · Reviewer_Qqzq · 2025-07-02

**Clarity:** 2
**Significance:** 2
**Originality:** 2
**Rating:** 4
**Confidence:** 2

**Summary:**

This paper investigates the theoretical underpinnings of transfer learning for Conditional Diffusion Models (CDMs). The core setting involves a source and target domain that share a common, low-dimensional representation of the condition, captured by a function $h(\cdot)$. The proposed transfer learning scheme first learns both a score model and the condition representation on the source tasks. Subsequently, for a new target task with limited data, the condition encoder is reused (frozen), and only the score model is fine-tuned.

The main theoretical contributions provide generalization bounds for this process. The authors first analyze the Lipschitz properties of the score function with respect to the data and the condition representation, establishing bounds that depend only on the initial data distribution. They then connect the source and target tasks through two established theoretical frameworks: task diversity (Theorem 3.4) and meta-learning (Theorem 3.6). In both settings, the analysis demonstrates that transfer learning can yield a tighter generalization bound compared to training from scratch. This improvement stems from leveraging the nK source samples to reduce the error associated with learning the ground-truth representation $h_*$ from its hypothesis class $\mathcal{H}$.

Finally, the authors present auxiliary results on the approximation error of the score function using deep ReLU networks and compare their bounds to prior work, arguing for improvements in terms of tightness, generality, and the mildness of the required assumptions. The central thesis is that learning a shared, low-dimensional condition representation is a key mechanism that enables the success of transfer learning with CDMs.

**Questions:**

In Thm 3.4, the error reduces in terms of $\log N_\mathcal{H}$ but can increase in terms of $\log N_\mathcal{F}$, how to interpret this ?

**Ethical Concerns:**

["NO or VERY MINOR ethics concerns only"]

**Final Justification:**

This paper has solid theoretical derivations, and it does offer justifications for the success of PEFT on diffusion models. Although I prefer theoretical studies with tighter connections and more insights into real-world applications, this bias should not be a major factor in a fair judgment of this paper. Also, the extra verification on the MNIST restoration serves as a good example, So I am increasing my final score to 4.

**Limitations:**

Please see the weaknesses section.

**Quality:**

3

**Strengths And Weaknesses:**

Strengths:
* The paper introduces a clear and tractable framework for analyzing transfer learning in CDMs. The central assumption of a shared, low-dimensional representation (Assumption 3.2) is intuitive and provides a solid foundation for the theoretical development.
* The analysis provides a novel approach to bounding the Lipschitz continuity of the score function (Lemma 3.1) under weaker assumptions than some prior works.

Weaknesses:
* A primary concern is the gap between the theoretical setup and common practices like Parameter-Efficient Fine-Tuning (PEFT). As noted in Remark 1, the analysis assumes the target score network $f^0$ is trained from scratch, independent of the pre-trained source models $f^K$.
* While Assumption 3.2 is critical for the analysis, it is a strong idealization. The existence of a single, perfectly shared representation $h_*$ across diverse tasks is a significant simplification.
* The experiments serve as a proof of concept but are somewhat limited. The core claim is only validated on a synthetic dataset.

---

> ### Author Rebuttal · Authors · 2025-07-31
>
> Thanks for your careful review and constructive feedback! We address your concerns as follows.
>
> W1. A primary concern is the gap between the theoretical setup and common practices like Parameter-Efficient Fine-Tuning (PEFT). As noted in Remark 1, the analysis assumes the target score network is trained from scratch, independent of the pre-trained source models.
>
> **A1**: The main contribution of our work is to take the first step towards understanding the sample efficiency of transfer learning CDMs and show that practical training procedures (pretraining and fine-tuning) is able to reduce sample complexity on target task. In practice, full-finetuning is widely used (e.g. T2I models, see more applications in Table 3 in Appendix A) and thus our theoretical setting is very close to practice. Although PEFT is also a popular training paradigm, its theoretical analysis is much more challenging and no existing theoretical paper has tackled with PEFT for neural networks to our knowledge. It is beyond the core idea of this paper and can be an interesting future work.
>
> **W2**. While Assumption 3.2 is critical for the analysis, it is a strong idealization. The existence of a single, perfectly shared representation across diverse tasks is a significant simplification.
>
> **A2**: Assumption 3.2 is a standard assumption in theoretical analysis of transfer learning [1,2,3], without which one cannot establish connection between source tasks and target tasks. In addition, in many applications such as the setting in our experiments and in Appendix A (e.g. T2I generation), there indeed exists a shared representation across tasks. Therefore, Assumption 3.2, although as an idealization, is very close to practice.
>
> **W3**. The experiments serve as a proof of concept but are somewhat limited. The core claim is only validated on a synthetic dataset.
>
> **A3**: For real data experiments, we consider the image restoration task on MNIST, where we have $K=9$ source tasks with $P^k(x,y)=p(y|x)p_k(x)$. Here the prior $p_k(x)$ is the data distribution of the digit $k$ in the MNIST dataset and $p(y|x)=\mathcal{N}(x, I_{784}/4)$.
> The target task is $P^0(x,y)=p(y|x)p_0(x)$ where $p_0$ is the data distribution of the digit 0.
> We use the complete MNIST 1-9 data for pre-training, which corresponds to $n=5000$.
> For the finetuning phase, we consider $m=10,20,30,40,50,100$ training samples from $P^0(x,y)$, and 100 test samples from $P^0(x,y)$.
> Below are the MSEs of fine-tuning models and train-from-scratch models and it's obvious fine-tuning is consistently better than train-from-scratch, indicating the benefits of transfer learning.
>
> |m|10|20|30|40|50|100|
> |---|---|---|---|---|---|---|
> |fine-tuning|0.3799|0.2846|0.2544|0.2406|0.2404|0.2268|
> |train-from-scratch|0.4409|0.3180|0.2746|0.2551|0.2501|0.2344|
>
> **Q1**: In Thm 3.4, the error reduces in terms of $\log N_{\mathcal{H}}$ but can increase in terms of $\log N_{\mathcal{F}}$, how to interpret this?
>
> **A4**: In Eq (3.4), the RHS increases in terms of both $\log N_{\mathcal{H}}$ and $\log N_{\mathcal{F}}$.
>
> [1] Tripuraneni, Nilesh, Michael Jordan, and Chi Jin. "On the theory of transfer learning: The importance of task diversity." Advances in neural information processing systems 33 (2020): 7852-7862.
>
> [2] Chua, Kurtland, Qi Lei, and Jason D. Lee. "How fine-tuning allows for effective meta-learning." Advances in Neural Information Processing Systems 34 (2021): 8871-8884.
>
> [3] Du, Simon S., et al. "Few-shot learning via learning the representation, provably." arXiv preprint arXiv:2002.09434 (2020).

---

> > ### Comment · Reviewer_Qqzq · 2025-08-05
> >
> > Regarding Q1: I was referring to the sample efficiency, because compared with training from scratch, the bound for transfer learning has a decrease in $\log \mathcal{N_H}$ but an increase in $\log \mathcal{N_F}$. And we should need $\frac{1}{n}\log \mathcal{N_F}\leq (\frac{1}{m}-\frac{1}{nK})\log \mathcal{N_H}$ for improved sample efficiency, which also means learning $h$ from $\mathcal{H}$ should at least not be much easier than learning $f$ from $\mathcal{F}$, is that correct?
> >
> > In general, this paper has solid theoretical derivations, and it does offer justifications for the success of PEFT on diffusion models. Although I prefer theoretical studies with tighter connections and more insights into real-world applications, this bias should not be a major factor in a fair judgment of this paper. Also, the extra verification on the MNIST restoration serves as a good example, so I will increase my score to 4.

---

> ### Author Response · Authors · 2025-08-05
>
> Thanks for your appreciation of our work. In transfer learning, the bound is $\frac{\log N_F}{m}+\frac{\log N_H}{nK}$ since the conditional encoder $h$ is fixed in the fine-tuning stage. In contrast, train-from-scratch also has to train a conditional encoder in addition to the backbone score network, leading to a bound of $\frac{\log N_F+\log N_H}{m}$. Therefore, transfer learning is always better in our setting unless $\log N_F\gg \log N_H$ in which case the leading term of both bounds are $\frac{\log N_F}{m}$. Please let us know if there is any further question.

---

### Official Review · Reviewer_CoEJ · 2025-07-03

**Clarity:** 3
**Significance:** 3
**Originality:** 3
**Rating:** 5
**Confidence:** 4

**Summary:**

As in other areas of machine learning, transfer learning has proven useful in the context of diffusion modeling: one can train on a large data set, and then fine-tune the model on some other data set, instead of training a model from scratch. On the other hand, it is not completely clear why one should expect transfer learning to work, and some theoretical guidance or guarantees would be helpful here. The authors endeavor to find such guarantees, essentially assuming models learn low-dimensional representations which are useful for multiple tasks. They present a large number of formal results on this topic, mainly on how well one expects a model to generalize on task X, given that it was trained on tasks A, B, C, D, ... from the same task distribution.

**Questions:**

1. How tight are the generalization error bounds? How tight are they in practice, when the assumptions may not exactly apply?
2. Can the authors provide explicit examples of the kinds of transfer learning tasks to which their theory applies? Are there examples of situations commonly viewed as 'diffusion model transfer learning' that, for one reason or another, the theory doesn't really apply to? Or is it hard to say? Lines 232-233 ("hard to verify in practice") sort of speak to this.

**Ethical Concerns:**

["NO or VERY MINOR ethics concerns only"]

**Final Justification:**

I maintain my original positive score. The paper has some limitations, mostly with relating its theoretical claims to more realistic settings, and the authors acknowledge this. This didn't change during the rebuttal period, but in fairness to the authors, it is generally hard to cleanly link this kind of theory to empirical experiments. Still, I think the math here is valuable for the community, and was very carefully done. I think this paper should be accepted.

**Limitations:**

There is no "limitations" section. I think it may be worth at least gesturing at the question of whether these bounds approximately hold in real settings, which is not addressed by the experiments currently in the paper.

**Paper Formatting Concerns:**

No concerns.

**Quality:**

4

**Strengths And Weaknesses:**

**Strengths.**
- The motivation is clear, and although the paper is highly technical, the writing is also fairly clear. I especially liked how the authors compared their work to related formal work on this topic, and explicitly related the bounds they derive to other bounds in the literature (see, e.g., the paragraph starting on line 285). This is helpful for someone not directly working on this topic.
- The math is very high quality and appears carefully done.
- The results suggest particular scalings of generalization error with various parameters (e.g., the number of previous tasks), including in the transfer learning setting.

**Weaknesses.**
- The "experiments" section is kind of underwhelming. One thing I'd be interested in is whether the scalings that follow from the math can be shown empirically, or if they are somewhat different in practice (see also the aforementioned discussion around line 285). But this may require more/different experiments than the authors have an appetite for.
- The way the authors assume different tasks are similar is a bit specific and abstract. It may be helpful to provide some examples of the kind of transfer learning tasks that their task similarity notion applies to, even just for the sake of exposition.

---

> ### Author Rebuttal · Authors · 2025-07-31
>
> Thanks for your careful review and constructive feedback! We address your concerns as follows.
>
> **W1**: The "experiments" section is kind of underwhelming. Can the scalings that follow from the math be shown empirically, or if they are somewhat different in practice (see also the aforementioned discussion around line 285). But this may require more/different experiments than the authors have an appetite for.
>
> **A1**: We have conducted additional experiments on real image data (A3 for Reviewer Qqzq) and ablations on non-identical representations (A2 for Reviewer JJrR).
> In practice, it is difficult in general to show the scaling shown in theoretical results. First, the theory only provides an upper bound for TV distance instead of an exact characterization. Besides, the TV distance between two distributions is typically hard to compute especially in high dimensional cases. Moreover, the theory is only about statistical rates while in practice, optimization error is non-negligible. These factors together make it difficult to empirically validate the theoretical scaling.
>
> **W2**: The way the authors assume different tasks are similar is a bit specific and abstract. It may be helpful to provide some examples of the kind of transfer learning tasks that their task similarity notion applies to, even just for the sake of exposition.
>
> **A2**: In Appendix B.5, we verify the task similarity assumptions and provide some naive bounds. A straightforward example is our experiment in Section 6. The conditional distribution $p_{k}(x|y)\propto p_{\beta_k}(x)p(y|x)$, where $p_{\beta_k}(x)$ is the prior distribution of the dynamics determined by a parameter $\beta_k$. The trajectory Eq (5.1) is very similar across all tasks except that the drift term involves different parameters. In this case, we believe that all tasks have similar structures and thus the notion in Eq (3.1) is reasonable.
>
> **Q1**: How tight are the generalization error bounds? How tight are they in practice, when the assumptions may not exactly apply?
>
> **A3**: To the best of our knowledge, there is no result on the lower bound of sample complexity in transfer learning setting, but our results match the state of the arts in existing literature. See more discussion in lines 285-292. In practice when the assumptions may not exactly apply (e.g., real images), our empirical results on image restoration for MNIST (A3 for Reviewer Qqzq) still indicate the sample efficiency of transfer learning. This is aligned with our theories.
>
> **Q2**: Can the authors provide explicit examples of the kinds of transfer learning tasks to which their theory applies? Are there examples of situations commonly viewed as 'diffusion model transfer learning' that, for one reason or another, the theory doesn't really apply to? Or is it hard to say?
>
> **A4**: We provide concrete examples in Appendix A. Specifically, Table 3 lists some real-world application of CDMs and the number of parameters (complexity) of backbone score networks and conditional encoders. In these applications, the conditional encoders are much more complex than the backbone score networks, indicating that the assumption $N_\mathcal{H}\gg N_{\mathcal{F}}$ in our theoretical setting is very common in practice. In Section A.1 and A.2, we discuss two specific applications of transfer learning CDMs, including amortized variational inference and behavior cloning via diffusion policy, which basically satisfy our assumptions. We acknowledge that it is indeed hard to quantify if representation is shared or not in general cases. However, our framework is motivated by the prevalent intuition that shared representations naturally emerge in standard pretraining–finetuning paradigms, which are widespread in diffusion model transfer learning.
>
> Limitations:
>
> Thanks for your valuable suggestions and we will add limitation and discussion section in the final version.

---

> > ### Comment · Reviewer_CoEJ · 2025-08-07
> >
> > I thank the authors for their detailed response.
> >
> > > These factors together make it difficult to empirically validate the theoretical scaling.
> >
> > This seems like a significant limitation, and should be highlighted explicitly in the revised version. But I am highly sympathetic to it being difficult to explicitly verify certain theoretical claims, so I don't think it's an *unreasonable* limitation.
> >
> > > We have conducted additional experiments on real image data (A3 for Reviewer Qqzq) and ablations on non-identical representations (A2 for Reviewer JJrR).
> >
> > I also think the notion of transfer learning considered in these experiments to be a bit narrow. Usually transfer learning means something much broader than (for example) pretraining on different digits (as you acknowledge in Appendix A). Potential difficulties linking the theoretical formulation to more realistic settings should be explicitly mentioned. In any case, I am still content with this limitation, since significantly expanding the experiments section would make the paper very different in aims.
> >
> > Overall, I continue to think that the authors have done a nice and detailed theoretical study of transfer learning in diffusion models, and thank them for their efforts. I maintain my positive score.

---

### Official Review · Reviewer_JJrR · 2025-07-22

**Clarity:** 4
**Significance:** 4
**Originality:** 3
**Rating:** 4
**Confidence:** 3

**Summary:**

This paper examines the generalization error of conditional diffusion models obtained through transfer learning with a smaller dataset than the source training sample. Under the assumption that a low-dimensional representation exists that is shared across all tasks, the generalization guarantee of conditional diffusion models trained using the score matching error is provided in Section 3 for settings where task diversity is both known and unknown. In Section 4, the authors apply the result to the estimation error bound of neural networks by evaluating their model complexity. Their results show that, under the assumption of low-dimensional representation, the curse of dimensionality is mitigated in lower dimensions.

**Questions:**

- While this paper focuses on the Lipschitz score, can the results be extended to other function spaces, such as the Holder space?
- Could the authors provide the lower bound on the estimation error trained through this setting? Specifically, I am interested in how the dependency on $m$ can be theoretically improved in transfer learning.

**Ethical Concerns:**

["NO or VERY MINOR ethics concerns only"]

**Final Justification:**

I would like to maintain my original score.

**Limitations:**

The authors properly address the limitations.

**Quality:**

3

**Strengths And Weaknesses:**

**Strength**
- Exploring the theoretical understanding of conditional diffusion models is a significant topic in the deep learning literature, and this paper addresses this issue.
- This paper provides a general framework for evaluating the generalization guarantee of the conditional diffusion model. The results in Section 3 are model-agnostic and provide substantial theoretical insights into the conditional diffusion model under the transfer learning procedure.
- This paper is clearly written. All theoretical results seem solid.

**Weakness**
- My primary concern is that the upper bound derived in Section 4 consists of $O(m^{-\frac{1}{d_x+d_y+9}})$, which is much worse than $O(m^{-\frac14})$ exhibited in [Yang et al., 2024], as the authors also refer to in the paper. Since $m\ll n$ typically holds in transfer learning, the exponent of $m$ substantially affects the convergence performance of the trained models.
- Additional numerical experiments will support the theoretical statements. For example, I would like to know he shared low-dimensional representation actually improves the generalization performance of trained models in artificial datasets.
- The definition of $R_f$ (in Sections 3 and 4) is lacking in the paper.

---

> ### Author Rebuttal · Authors · 2025-07-31
>
> Thanks for your careful review and constructive feedback! We address your concerns as follows.
>
> **W1**: The primary concern is that the upper bound derived in Section 4 consists of $O(m^{-\frac{1}{d_x+d_y+9}})$, which is much worse than $O(m^{-\frac{1}{4}})$ exhibited in [1].
>
> **A1**: We would like to point out that [1] considers a completely different setting from ours. In fact, [1] considers unconditional diffusion models and the unconditional distribution is assumed to be supported in a low-dimensional linear subspace, where the source task and the target task have the same latent variable distribution. Hence, only a linear encoder is trained for fine-tuning instead of the full score network. The complexity of the linear encoder function class is much lower than that of the neural network family in our setting and hence [1] is able to achieve better rates. We believe that our methods for CDMs can be extended to this setting and get similar results as well.
>
> **W2**: Does the shared low-dimensional representation actually improve the generalization performance of trained models in artificial datasets?
>
> **A2**: We conduct an additional experiment following the setting in Section 5 except that we use distinct operators $M_k$ for $p_k(y|x)=\mathcal{N}(M_kx, I_{100}/4)$ and $M_k$ are independently generated for different tasks. In this case, source and target tasks will not share identical representation and the MSEs on target task are shown below. The performance of fine-tuning is even worse than train-from-scratch, indicating that transfer learning fails without shared representation. We remark that $M_k$ varies from each other substantially in this setting leading to highly variable low-dimensional representations for each task. However, if $M_k$'s are similar (not necessarily identical), fine-tuning can still outperform train-from-scratch.
>
> |m|10|20|30|40|50|100|
> |---|---|---|---|---|---|---|
> |fine-tuning|943.08|215.6|184.94|81.17|69.1|48.38|
> |train-from-scratch|21.99|10.61|5.71|2.38|1.77|1.04|
>
>
> **W3**: The definition of $R_f$ (in Sections 3 and 4) is lacking in the paper.
>
> **A3**: $R_f$ is defined in Section 2.3 (see line 161), which quantifies the region in which the score network $f$ is Lipschitz. For generalization bounds in Section 3, we assume this parameter has some lower bound. In Section 4, we explicitly present the value of $R_f$ in Eq (4.3). We'll make cleaner statements in the revised version.
>
> **Q1**: While this paper focuses on the Lipschitz score, can the results be extended to other function spaces, such as the Holder space?
>
> **A4**: Both Lipschitz score [1,2,3] and Holder density [4,5] are commonly used assumptions in the literatures of diffusion models. And either of two assumptions are not able to indicate the other. In our case, the Lipschitz continuity of score function is crucial since it enables a Lipschitz neural network estimator, which is essential in transfer learning theories. Expansion of our methods to the Holder density space is an interesting future work.
>
> **Q2**: Could the authors provide the lower bound on the estimation error trained through this setting, especially on the dependence of $m$?
>
> **A5**: [2] studies the lower bound of score matching loss with Lipschitz score and achieves the min-max optimal rate of $O(m^{-\frac{2}{d+4}})$ (Thm. 3). If we apply this result to diffusion models, the min-max optimal rate of TV distance should be $O(m^{-\frac{1}{d+4}})$ as indicated in [2] (Coro. 1). This is aligned with our results when reduced to unconditional DMs, i.e. $d_y=D_y=0$. To rigorously establish a lower bound in CDMs and our settings is also an interesting future work.
>
> [1] Yang, Ruofeng, et al. "Few-shot diffusion models escape the curse of dimensionality." Advances in Neural Information Processing Systems 37 (2024): 68528-68558.
>
> [2] Wibisono, Andre, Yihong Wu, and Kaylee Yingxi Yang. "Optimal score estimation via empirical bayes smoothing." The Thirty Seventh Annual Conference on Learning Theory. PMLR, 2024.
>
> [3] Chen, Minshuo, et al. "Score approximation, estimation and distribution recovery of diffusion models on low-dimensional data." International Conference on Machine Learning. PMLR, 2023.
>
> [4] Oko, Kazusato, Shunta Akiyama, and Taiji Suzuki. "Diffusion models are minimax optimal distribution estimators." International Conference on Machine Learning. PMLR, 2023.
>
> [5] Fu, Hengyu, et al. "Unveil conditional diffusion models with classifier-free guidance: A sharp statistical theory." arXiv preprint arXiv:2403.11968 (2024).

---

> > ### Comment · Reviewer_JJrR · 2025-08-04
> >
> > I sincerely appreciate the authors' response.
> > The responses adequately address my concern, and then I would like to maintain my current score.

---

### Note · Authors · 2025-08-15

We thank all reviewers for their valuable feedback. Following the rebuttal, all reviewers expressed positive opinions about the paper, and we are encouraged by their recognition of several key strengths in our work:

1. **Clear motivation and significance** – The motivation is well articulated, and exploring the theoretical foundations of conditional diffusion models (CDMs) is an important and timely topic (Reviewers JJrR, CoEJ).

2. **Clarity and accessibility** – The paper is well written and easy to follow, offering crucial insights that help even non-specialists develop a deeper understanding of the topic (all reviewers).

3. **Mathematical rigor and comprehensive framework** – The mathematical analysis is carefully executed, and the proposed framework provides substantial theoretical insights into the sample efficiency of transfer learning in CDMs (Reviewers 8krS, CoEJ, JJrR).

4. **Novel technical contributions** – The paper introduces new techniques for handling Lipschitz continuity under significantly weaker assumptions (Reviewer Qqzq).

We also address the main concerns raised by reviewers:

1. **Practical applicability of the theoretical framework** (Reviewers CoEJ, 8krS, Qqzq): Appendix A presents concrete real-world examples of transfer learning in CDMs where our theory applies. In Table 3, we highlight tasks where the conditional encoder contains substantially more parameters than the backbone score network, making it harder to learn—consistent with our theoretical setting. We further discuss two concrete applications—amortized variational inference and behavior cloning via diffusion policy—demonstrating the general applicability of our results.

2. **Limited experimental scope (synthetic datasets)** (Reviewers JJrR, Qqzq): We have added new experiments on image restoration task using CDMs on MNIST. The results align with our theoretical predictions, showing that transfer learning substantially improves the sample efficiency of CDMs compared with training from scratch.

We will incorprate all addition experiment results, discussions and limitations into the final version. We thank the reviewers for a productive discussion that addressed their initial concerns. We are especially encouraged by the positive acknowledgment and subsequent score increase from Reviewer Qqzq, and note that no outstanding issues were raised by any other reviewer.

---

### Decision · Program_Chairs · 2025-09-17

**Decision:**

Accept (poster)

**Comment:**

This paper studies transfer learning for training conditional diffusion models. Assuming that the distributions of different conditions share certain common information, the authors derive generalization error bounds for conditional diffusion models, showing how transfer learning can reduce sample complexity.

This is a theoretical paper. Overall, the reviewers agreed that the paper is well written, the analysis is rigorous, and the technical contribution is clear. Reviewers raise concerns regarding the connection of the theoretical claims to realistic settings, which are partly addressed in the rebuttal by some additional experimetns. The authors are encouraged to address this by including additional experiments and further discussion in the final version.